# Spotiphy enables single-cell spatial whole transcriptomics across an entire section

Jiyuan Yang [1,5], Ziqian Zheng[2,5], Yun Jiao[3], Kaiwen Yu [4], Sheetal Bhatara[1], Xu Yang [1], Sivaraman Natarajan [1], Jiahui Zhang[2], Qingfei Pan[1], John Easton [1], Koon-Kiu Yan [1], Junmin Peng [3,6] ✉, Kaibo Liu[2,6] ✉ & Jiyang Yu [1,6] ✉

Spatial transcriptomics (ST) has advanced our understanding of tissue regionalization by enabling the visualization of gene expression within whole-tissue sections, but current approaches remain plagued by the challenge of achieving single-cell resolution without sacrificing whole-genome coverage. Here we present Spotiphy (spot imager with pseudo-single-cell-resolution histology), a computational toolkit that transforms sequencing-based ST data into single-cell-resolved whole-transcriptome images. Spotiphy delivers the most precise cellular proportions in extensive benchmarking evaluations. Spotiphy-derived inferred single-cell profiles reveal astrocyte and disease-associated microglia regional specifications in Alzheimer's disease and healthy mouse brains. Spotiphy identifies multiple spatial domains and alterations in tumor–tumor microenvironment interactions in human breast ST data. Spotiphy bridges the information gap and enables visualization of cell localization and transcriptomic profiles throughout entire sections, offering highly informative outputs and an innovative spatial analysis pipeline for exploring complex biological systems.

The location of a cell within its microenvironment is a critical determinant of its function and interactions with neighboring cells[1]. Cells possess unique characteristics and functions, which are finely tuned to meet the specific needs of their environments through a process known as cellular regional specification[2]. Analyzing local cell behavior involves identifying spatial domains, regions characterized by cellular composition and their transcriptomic profiles. A cell's response to external signals is shaped by its spatial domain, affecting tissue development, homeostasis and disease progression[3]. Current spatial transcriptomics (ST) platforms[4–6] leave an unmet need for single-cell-resolved transcriptomic profiles with genome-wide coverage. ST technologies can generally be grouped into two categories: sequencing-based approaches (for

example, Visium[7], DBiT-Seq[8], Slide-Seq[9,10], Slide-tags[11] and Stereo-seq[12]) and image-based approaches (for example, SeqFISH[13,14], smFISH[15], RNAScope[16], MERFISH[17,18], Xenium[19] and CosMx[20]). Predefining a capture area, or 'spot', allows sequencing-based approaches to generate unbiased genome-wide transcriptomic profiles while minimizing lateral leaking contamination. Yet, limiting the reads to predetermined spots comes with two major drawbacks, specifically information loss in the areas between the spots (noncapture areas) and low resolution (as each spot consists of multiple, often heterogeneous, cells). Image-based ST approaches using high-resolution fluorescent images achieve single-cell resolution but are limited to preselected gene panels consisting of 300–1,000 targets. Image-based ST approaches usually capture

[1]Department of Computational Biology, St. Jude Children's Research Hospital, Memphis, TN, USA. [2]Department of Industrial & Systems Engineering, University of Wisconsin–Madison, Madison, WI, USA. [3]Department of Structural Biology, St. Jude Children's Research Hospital, Memphis, TN, USA. [4]Center of Proteomics and Metabolomics, St. Jude Children's Research Hospital, Memphis, TN, USA. [5]These authors contributed equally: Jiyuan Yang, Ziqian Zheng. [6]These authors jointly supervised this work: Junmin Peng, Kaibo Liu, Jiyang Yu. ✉e-mail: junmin.peng@stjude.org; kliu8@wisc.edu; jiyang.yu@stjude.org

limited counts per gene, making them better suited for validation than data-driven discovery. Initial attempts have focused on enhancing the resolution of sequencing-based ST data, with various deconvolution methods for spot-level ST data already established. These methods integrate single-cell RNA-sequencing (scRNA-seq) and ST data to parse the cellular composition of individual transcriptomic spots[21–33]. Yet, none of the current methods are able to achieve single-cell resolution while maintaining the spatially varying gene expression present in the original sequencing-based ST data.

We present Spotiphy, a toolkit that resolves the gene-coverage-resolution tradeoff. Spotiphy generates inferred scRNA expression profiles (iscRNA data) from all cells (located in both capture and non-capture areas), achieving spatially resolved whole-slide transcriptomic profiling. This provides substantial benefits for downstream analyses and opportunities for biological insight. Spotiphy-derived iscRNA data reveal regional specification of astrocytes and microglia in healthy and Alzheimer's disease (AD) mouse brains, offering a level of detail not detectable by existing scRNA-seq and ST technologies. Spotiphy identifies changes in the patterns of tumor–tumor microenvironment (TME) interactions across multiple spatial domains and delivers single-cell-resolved whole-transcriptome images of the entire section, greatly expanding the information intensity of ST data and providing a more comprehensive and detailed understanding of spatially resolved cellular states and regulatory mechanisms.

## Results

### Spotiphy achieves single-cell spatial whole transcriptomics

Spotiphy delivers single-cell spatial whole-transcriptome profiling via generative modeling of sequencing-based ST, scRNA-seq and histological imaging data (Fig. 1a and Methods). Specifically, Spotiphy selects the most informative genes for each cell type to generate a signature reference from scRNA-seq data. It presets five customizable hyperparameters to ensure scRNA-seq reference accuracy and robustness (Supplementary Fig. 1). Spotiphy determines the locations of nuclei by segmenting the high-resolution histological images. It integrates the above information into a probabilistic model that considers the distribution of gene contributions from each cell type. This unique feature enables the simultaneous deconvolution and decomposition of ST data, thereby generating both cell-type proportion and iscRNA data. Additionally, Spotiphy imputes cell-type proportion and iscRNA data for cells located in noncapture areas through Gaussian processes[34]. Consequently, Spotiphy generates single-cell-resolved images equivalent to the output of image-based approaches, with whole-transcriptomic profiling across entire sections.

### Matched mouse brain datasets for biological validity

To evaluate Spotiphy's performance and the biological validity of its results, we enriched rare immune cells and obtained scRNA-seq profiles for 27,836 CD45$^+$CD11b$^+$ cells (microglia, macrophages and neutrophils), 6,085 CD45$^+$CD11b$^-$ cells (T cells and B cells) and 29,563 CD45$^-$CD11b$^-$ cells (neurons, glial cells and so on) from an AD mouse model[35] and wild-type (WT) control mice (Fig. 1b, Supplementary Table 1 and Methods). By supplementing our data with the Allen Brain Map Atlas[36], we assembled a comprehensive mouse brain single-cell reference with 27 cell types, including neurons, glia and immune cells (Fig. 1c, Supplementary Fig. 2a,b and Supplementary Table 2). We then generated datasets of various ST techniques, including immunohistochemistry (IHC), Visium, Xenium and CosMx. Serial sectioning of the same sample produced nearly identical slices, allowing direct comparison between different ST approaches once the histological images from these platforms were aligned (Supplementary Fig. 2c–e). The matched datasets are a valuable resource for the spatial omics community, particularly for the development and evaluation of ST algorithms.

### Spotiphy excels in rare cell type deconvolution accuracy

To evaluate the accuracy of Spotiphy's cellular deconvolution, Xenium data generated from an AD sample were used as the ground truth, and heat maps of cell-type proportions across the section were generated by Spotiphy and 13 other benchmarking methods[21–33] (Fig. 2a, Extended Data Fig. 1a, Supplementary Figs. 3–5 and Methods). For well-organized excitatory glutamatergic neurons, Spotiphy's delineation of the cortical layers was closer to that of Xenium and had clearer boundaries than those of the other methods. Spotiphy's distribution patterns for unevenly distributed inhibitory GABAergic interneurons were likewise closer to the ground truth. Because the Xenium panel did not include markers for immune cells, we used in situ hybridization (ISH) results from the Allen Mouse Brain Atlas (mouse.brain-map.org)[37] as our ground truth for neutrophils and T and B cells. Whereas most other methods being tested failed to predict the distribution of neutrophils and instead produced random results, Spotiphy identified a specific enrichment of neutrophils around the ventricle that is in line with the ISH results for neutrophil markers, including *Itgb2* (Fig. 2a and Supplementary Figs. 4–6). RCTD[22] appeared comparable to Spotiphy, yet it incorrectly classified some neutrophils as macrophages, leading to an apparent increase of macrophages around the ventricle that did not align with the Xenium data (Supplementary Fig. 4e). Similar misidentifications among microglia, macrophages and neutrophils were observed in the CIBERSORTx results[26] (Supplementary Fig. 5e). No obvious ISH signal was observed for *Cd3e* (T cell marker) or *Cd19* (B cell marker), consistent with the fact that T and B cells are rarely found in the mouse brain. Due to the low numbers of immune cells, and varying cell counts per spot, using proportions for rare cells can lead to inaccuracies. Thus, we translated these proportions into absolute cell counts (multiplying by the total cells per spot segment) for further comparison (Supplementary Fig. 7). For B cells, most methods performed well except CytoSPACE[23], StereoScope[28], iStar[32] and Cell2location[21]. A low rate of false-positive detection of T cells in the striatum (STR) region has been observed in the predictions of Spotiphy, CARD[24] and Tangram[25]. Spotiphy's predictions for macrophages and microglia aligned well with the ground truth. Multiple metrics were used to benchmark Spotiphy's cellular deconvolution against other methods (Fig. 2b–d, Extended Data Fig. 1b–e, Supplementary Fig. 3 and Supplementary Table 3). Spotiphy consistently produced the highest overall Pearson's correlation coefficient with the ground truth. Spotiphy's cell-type proportions for each transcriptomic spot aligned more closely with the ground truth, as evidenced by higher values of correlation, the fraction of cells correctly mapped and the cosine similarity, along with lower values for the absolute error, mean square error and Jensen–Shannon divergence (JSD).

Although the matched Xenium data were useful for evaluating biological validity, it was not an ideal choice of ground truth for benchmarking the method's performance. This is because (1) Xenium's predesigned gene panel has limited plexity, and (2) the data were generated from serial sections instead of an identical section. To obtain a comprehensive benchmarking analysis, we therefore constructed a series of simulated Visium datasets from scRNA-seq data with three noise levels. Additionally, we included eight paired 'scRNA-Visium' datasets from various human tissues[38] to evaluate the deconvolution performance of Spotiphy against the other 13 methods (Extended Data Fig. 2a–e, Supplementary Figs. 8–11, Supplementary Table 4 and Methods). The quantitative metrics further demonstrated that Spotiphy consistently ranked among the top in performance, if not the best. Moreover, Spotiphy's computational time was substantially lower than that of the other methods (Extended Data Fig. 2f). In summary, with a total of 13 methods included for deconvolution benchmarking, our evaluations demonstrate Spotiphy's superiority across different tissue types.

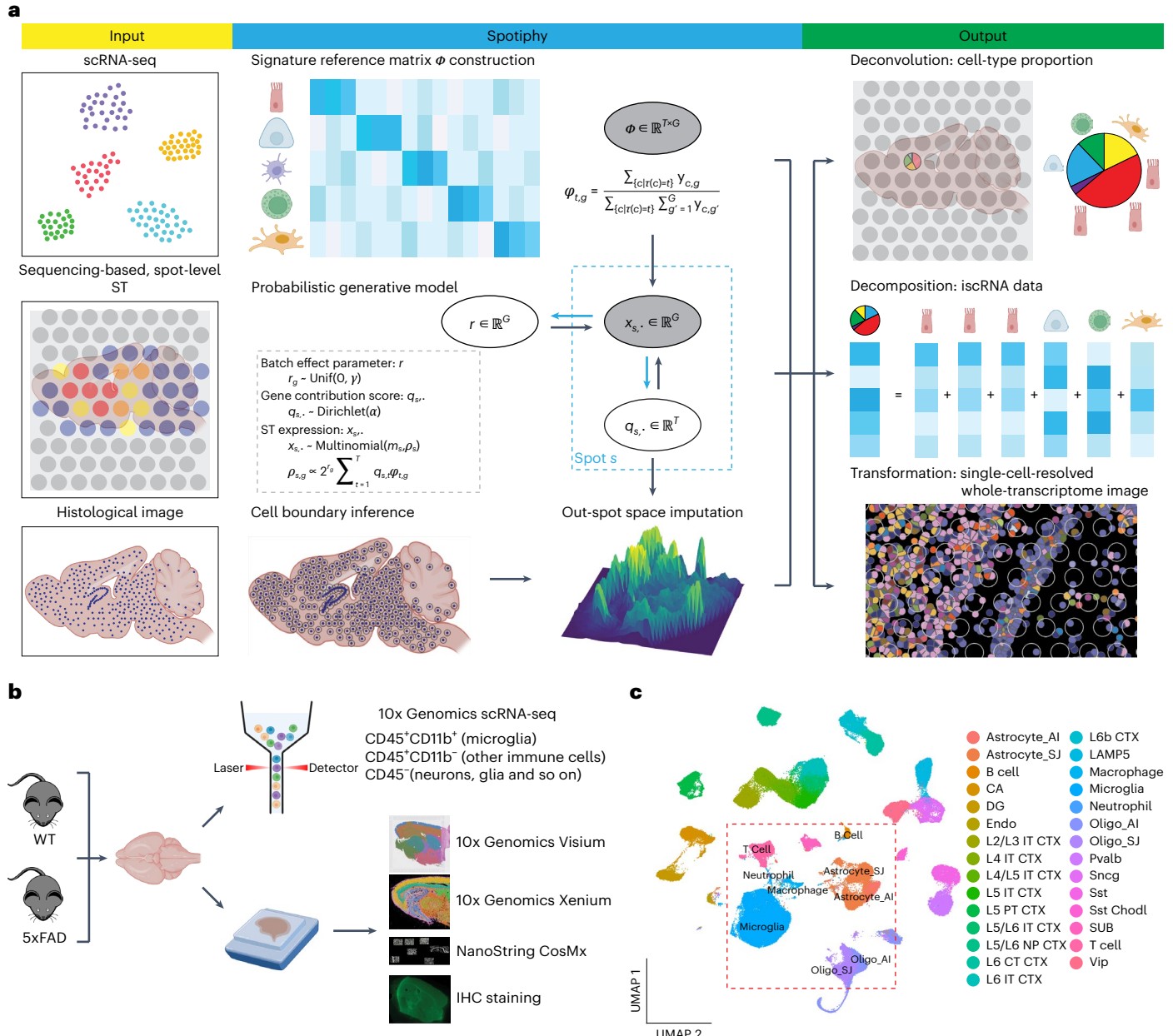

**Fig. 1 | Schematic overview of Spotiphy and the matched datasets used for validation. a**, Spotiphy workflow. Spotiphy requires three types of input data: scRNA-seq data, spot-level sequencing-based ST data and high-resolution histological images. The workflow consists of five major steps. (1) Reference construction. scRNA-seq data are used to select top marker genes and generate a signature reference for all cell types. (2) Segmentation. The histological image is used to count the number of nuclei in each spot. (3) Generative modeling. ST data are used to generate cell-type proportions and iscRNA data. (4) Imputation. Gaussian processes are used to impute cell-type proportions and iscRNA data for cells located outside the transcriptomic spots. (5) Transformation. The results of steps 3 and 4 are merged to produce a pseudo-single-cell-resolved whole-transcriptome image. **b**, Process of generating matched datasets for WT and AD mouse brain tissues. Some icons are from BioRender. **c**, Uniform

manifold approximation and projection (UMAP) of scRNA-seq reference data for the mouse brain. Twenty-two cell types from the Allen Brain Map Atlas (https://portal.brain-map.org/atlases-and-data/rnaseq) and seven cell types from our original data (red box) are included. Astrocytes and oligodendrocytes from both datasets are included for batch correction. Astrocyte_AI, astrocytes from the Allen Brain Map Atlas; Astrocyte_SJ, astrocytes from the original data; IT, interneuron; Oligo_AI, oligodendrocytes from the Allen Brain Map Atlas; Oligo_SJ, oligodendrocytes from the original data; CA, cornu ammonis; CT, corticothalamic; DG, dentate gyrus; Endo, endothelial; L, layer; Pvalb, parvalbumin; PT, pyramidal tract; SNCG, synuclein-γ; SST, somatostatin; SST Chodl, somatostatin and chondrolectin coexpressing; VIP, vasoactive intestinal polypeptide; NP, near-projecting; SUB, subiculum.

## Spotiphy captures astrocyte regional specification

Besides surpassing its peers in deconvolution, Spotiphy is unique in its ability to decompose the transcriptomic profiles of spots to the single-cell level (iscRNA data; Extended Data Fig. 3a and Methods). When we applied unsupervised clustering[39] to the iscRNA data from 33,819 cells generated from mouse brain Visium data, individual

astrocytes did not cluster together; instead, they formed multiple sub-clusters, indicating intraheterogeneity not apparent in the single-cell reference (Figs. 1c and 3a, Extended Data Fig. 3b–e and Supplementary Fig. 12). Similar phenotypes were observed in oligodendrocytes and multiple types of neurons. We subsequently color coded the astrocytes based on their respective subcluster and overlaid the same colors onto

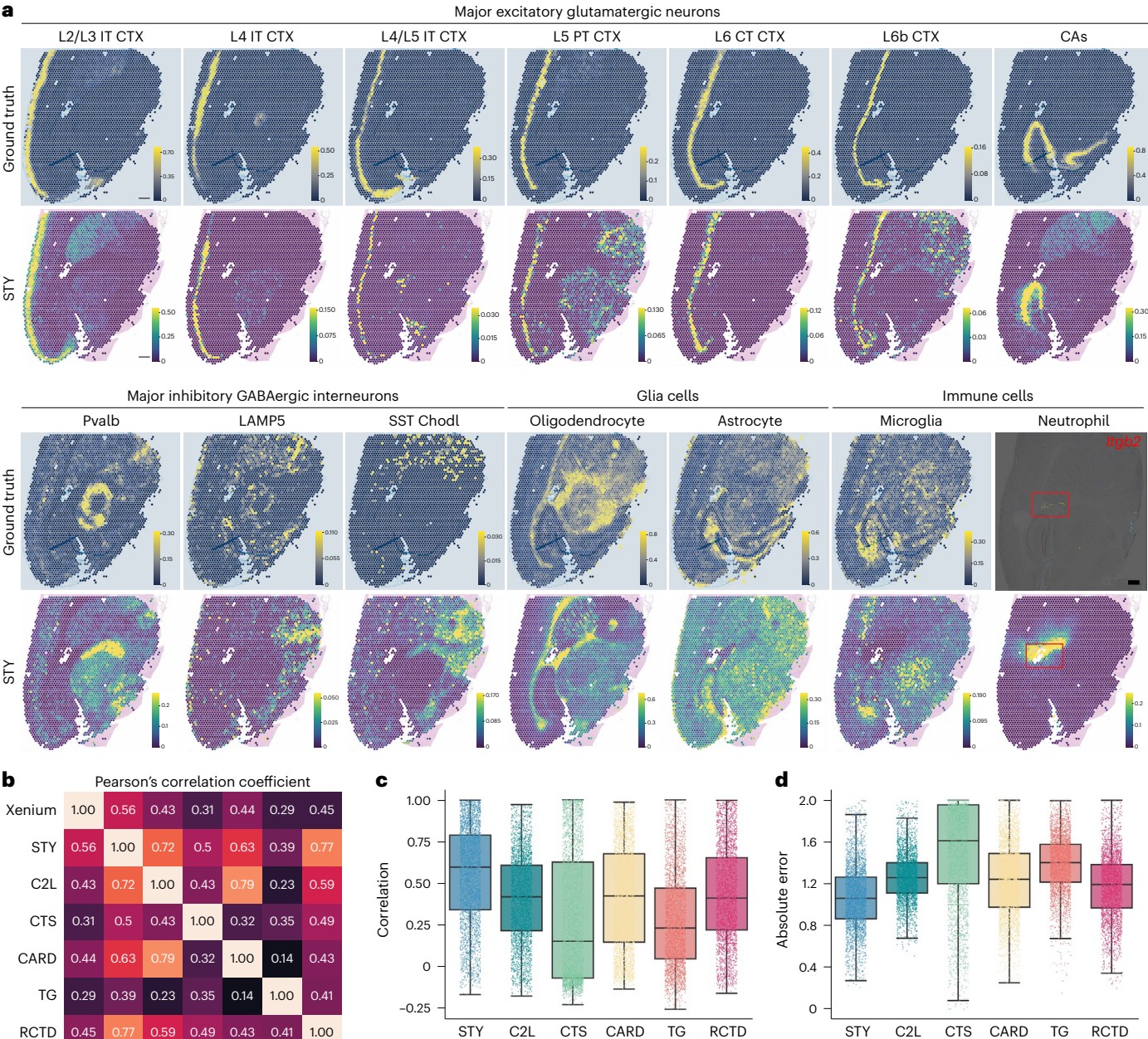

**Fig. 2 | Benchmarking Spotiphy's cellular deconvolution using matched Xenium data. a**, Heat maps depicting the proportion of selected cell types (as determined by Spotiphy and Xenium, the ground truth) across the histological image of the AD sample. The red box in the image showing neutrophils indicates the ventricle region. Expression of *Itgb2* is used as the ground truth for neutrophils in the mouse brain (Allen Mouse Brain Atlas, mouse. brain-map.org/experiment/show/77464984); scale bar, 500 μm. **b**, Pearson correlation coefficient heat map of cell-type proportions generated by Xenium, Spotiphy and other methods selected for benchmarking; STY, Spotiphy; C2L, Cell2location; CTS, CytoSPACE; TG, Tangram. **c,d**, Box plots illustrate the

correlation (**c**; higher is better) and absolute error (**d**; lower is better) for cell-type proportions at each transcriptomic spot generated by each method. The box represents the interquartile range (IQR), with the lower bound indicating the 25th percentile, the middle line indicating the median (50th percentile) and the upper bound indicating the 75th percentile. Whiskers extend from the box to show the data range, with the lower whisker extending to the minimum value (or the smallest data point within 1.5 times the IQR) and the upper whisker extending to the maximum value (or the largest data point within 1.5 times the IQR). Each platform includes 3,476 spots from the AD sample.

the histological images at their spots of origin (Fig. 3b). Surprisingly, the distinct spatial distribution of these subclusters corresponded perfectly with the classical histological regions of the mouse brain, in which the sagittal section is divided into six major topographic regions: cerebral cortex (CTX), hippocampus (HPF), fiber tracts (FT), thalamus (TH), hypothalamus (HY) and stratum (STR) (Extended Data Fig. 3f). Notably, the spatial distribution of astrocyte subclusters was nearly identical in the WT and AD samples; the proportions of each astrocyte

subtype likewise did not differ substantially between the two samples (Fig. 3b and Extended Data Fig. 3e).

Further downstream analysis of the iscRNA data generated by Spotiphy revealed regional specification of astrocytes. To explore potential functional differences in the astrocytes of the telencephalon and diencephalon, we selected a region from each (CTX and TH, respectively) for comparison. Using NetBID2 (ref. 40), we identified differentially expressed genes (DEGs) between CTX and TH astrocytes

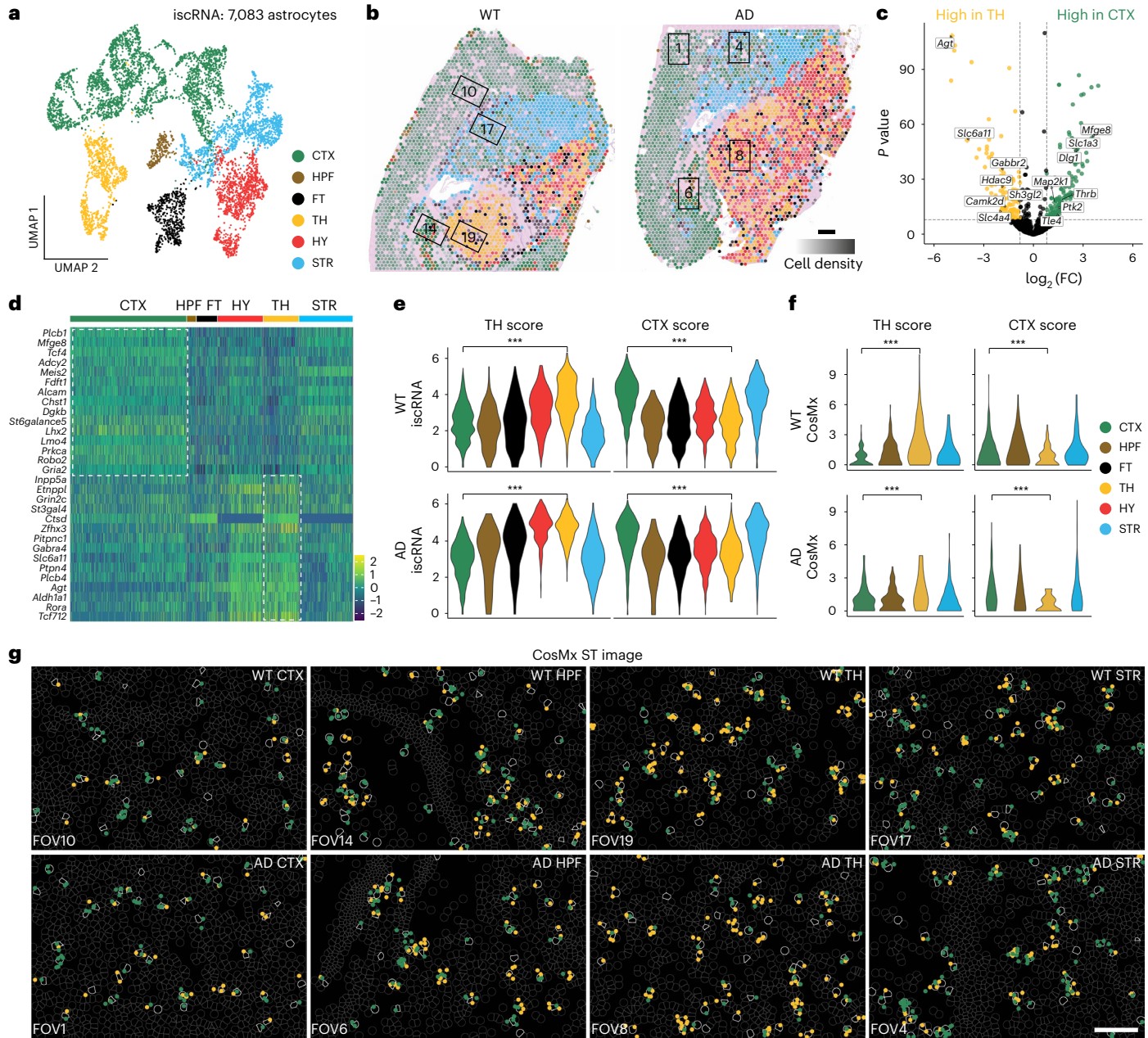

**Fig. 3 | Spotiphy captures astrocyte regional specification in mouse brain tissue. a**, UMAP projection of 7,083 astrocytes extracted from iscRNA data produced by applying Spotiphy to mouse brain Visium data. Clusters are labeled according to their corresponding topographic regions. **b**, Transcriptomic spots from the Visium data. Spots are color coded according to their constituent astrocytes' subclusters annotated in **a**. The opacity of the spots is representative of cell density. Spots not containing astrocytes were removed for clarity. Black boxes represent the corresponding FOVs of the CosMx data (FOV1: AD CTX; FOV6: AD HPF; FOV8: AD TH; FOV4: AD STR; FOV10: WT CTX; FOV14: WT HPF; FOV19: WT TH; FOV17: WT STR); scale bar, 500 μm; FC, fold change. **c**, Volcano plot of DEGs between astrocytes in the CTX and TH. The $x$ axis represents the $\log_2$ (fold change), and the $y$ axis represents the $-\log_{10}$ (adjusted $P$ value) computed using a two-sided $t$-test. DEGs were determined using the iscRNA data.

The highlighted genes are available in the CosMx panel and have been used to generate the signature scores in **e–g**. **d**, Heat map showing the expression of the top 15 DEGs among astrocytes in the CTX region and the TH region, respectively. **e,f**, Violin plots of the CTX and TH scores for each astrocyte subcluster identified from the iscRNA (**e**; all $P$ values are $< 2.2 \times 10^{-16}$) and CosMx (**f**; $P$ values: top left, $2.3 \times 10^{-13}$; top right, $5.7 \times 10^{-4}$; bottom left, $2.2 \times 10^{-3}$; bottom right, $3.1 \times 10^{-10}$) data. Two-sided $t$-tests were conducted to determine the statistical significance of the CTX and TH scores of the astrocytes in each subcluster; ***$P < 0.01$. **g**, Visualization of signature genes in selected FOVs of the CosMx data. Yellow and green dots represent the signature genes used to calculate the TH and CTX scores, respectively. Cell boundaries are depicted with gray lines; astrocytes are outlined in white. CosMx data include one biological replicate each for WT and AD samples (Supplementary Fig. 2d); scale bar, 100 μm.

(Fig. 3c,d and Supplementary Table 5) and then used the top DEGs to define signature scores for each region (CTX score and TH score; Methods). As expected, each astrocyte scored high for its region of origin and low for the comparison region in both WT and AD mouse samples

(Fig. 3e). To validate the accuracy of the iscRNA data and subsequent analyses, we used the CosMx data in the CTX and TH regions in the WT and AD mouse samples as the ground truth. The CosMx data also showed that astrocytes in the CTX region had a higher CTX score, and

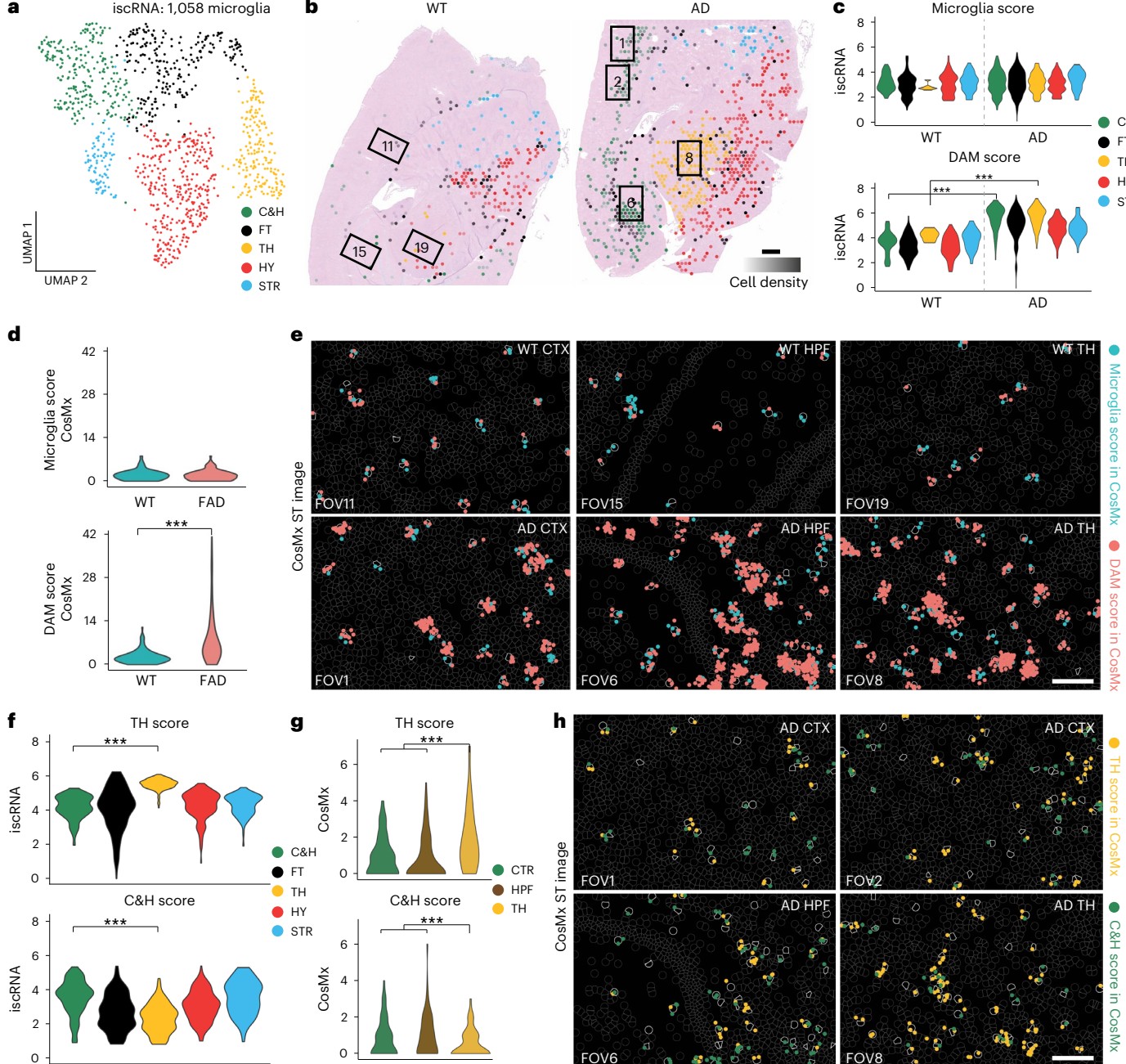

**Fig. 4 | Spotiphy reveals microglia regional specification in AD mouse brain tissue. a,** UMAP of 1,058 microglia extracted from iscRNA data generated by Spotiphy to mouse brain Visium data. Clusters are labeled according to their corresponding topographic regions. **b,** Transcriptomic spots from the Visium data. Spots are color coded according to their constituent microglia's subclusters, annotated in **a**. The opacity of the spots is representative of cell density. Spots without microglia were removed for clarity. Black boxes represent the corresponding FOVs of the CosMx data (FOV1: AD CTX; FOV 2: AD CTX; FOV6: AD HPF; FOV8: AD TH; FOV11: WT CTX; FOV15: WT HPF; FOV19: WT TH); scale bar, 500 μm. **c,d,** Violin plots of the microglia and DAM scores for each microglia subcluster identified from the iscRNA (**c**; the *P* value for the C&H region is $< 2.2 \times 10^{-16}$, and the *P* value for the TH region is $1.8 \times 10^{-4}$) and CosMx

(**d**; $P < 2.2 \times 10^{-16}$) data. **e,** Visualization of signature genes in selected FOVs of the CosMx data. Blue and red dots represent the signature genes used to calculate the microglia and DAM scores, respectively. **f,g,** Violin plots of the C&H and TH scores for the microglia subclusters present in the FAD sample iscRNA (**f**; all *P* values are $<2.2 \times 10^{-16}$) and CosMx (**g**; top CTX–TH: $P = 5.5 \times 10^{-4}$, HPF–TH: $P = 7.8 \times 10^{-5}$; bottom CTX–TH: $P = 8.1 \times 10^{-3}$, HPF–TH: $P = 3.7 \times 10^{-3}$) data. **h,** Visualization of signature genes in selected FOVs of the CosMx data. Yellow and green dots represent the signature genes used to calculate the TH and C&H scores, respectively. Cell boundaries are depicted with gray lines; microglia are outlined in white. CosMx data in **e** and **h** include one biological replicate each for WT and AD samples (Supplementary Fig. 2d); scale bars, 100 μm (**e** and **h**). Data in **c**, **d**, **f** and **g** were analyzed by two-sided *t*-tests; ***$P < 0.01$.

astrocytes in the TH region had a higher TH score (Fig. 3f,g). Differences in the actual transcripts (as measured by CosMx) of the astrocytes from these regions further supported this astrocyte regional specification. Gene set enrichment analysis (GSEA) demonstrated

enrichment of CTX astrocytes in pathways associated with neuronal differentiation and cell fate specification; TH astrocytes, meanwhile, were found to show greater expression of genes related to neuronal synaptic plasticity. These data suggest distinct biological roles for

astrocytes in different regions of the mouse brain (Extended Data Fig. 3g and Supplementary Table 6).

A recent study identified two distinct astrocyte clusters in mouse brains using single-nucleus RNA-seq (snRNA-seq) profiles of astrocytes enriched in different regions[21]. These clusters aligned with the telencephalon (cluster 1) and diencephalon (cluster 2), indicating regional expression differences in astrocytes (Extended Data Fig. 4a). When we applied the signature scores to these data, astrocytes in cluster 1 had a high CTX score and low TH score, whereas those in cluster 2 had a low CTX score and a high TH score, consistent with the astrocyte regional specification that we observed in the iscRNA data (Extended Data Fig. 4b,c).

Furthermore, we generated iscRNA data from the simulated Visium datasets discussed earlier. Unsupervised clustering of the simulated iscRNA data showed that astrocytes, oligodendrocytes and neurons were clustered together and did not form subclusters as they did in the real iscRNA data (Extended Data Fig. 4d,e). Nearly all astrocytes grouped into one single cluster, except for two minor subclusters, which exhibited random distribution and no clear regional patterns (Extended Data Fig. 4f,g). As the simulated Visium datasets were generated using scRNA-seq data, the lack of spatial variable genes and identifiable regional specification was anticipated. This result indicates that Spotiphy will not introduce any 'variations' stemming from its modeling process, suggesting that the regional specification observed in the real iscRNA data reflects genuine biological phenomena rather than merely an artifact of Spotiphy.

## Spotiphy reveals disease-associated microglia regional specifications

To confirm that the iscRNA data generated by Spotiphy for rare cell populations are likewise biologically meaningful, we analyzed the microglial populations, which formed five subclusters corresponding to the histological regions of the mouse brain, with a single cluster (C&H) encompassing both the CTX and HPF regions (Fig. 4a,b and Extended Data Fig. 5a). Both the WT and AD brain samples showed similar distribution patterns of microglia in the FT, HY and STR regions; however, microglia in the C&H and TH regions were found to originate mainly from the AD sample (Extended Data Fig. 5b). To characterize these AD-specific microglial populations, we used NetBID2 to identify the top DEGs and found considerable overlap between those genes and disease-associated microglia (DAM) markers[41–43], suggesting that the microglia in these regions of the AD brain may in fact be DAM (Extended Data Fig. 5c and Supplementary Table 7). We then used commonly used microglia and DAM markers to define signature scores and applied them to the iscRNA data for the microglia from both samples (Extended Data Fig. 5d). As expected, the microglia scores for each region did not vary significantly between the WT and AD samples. The DAM score, however, was significantly higher in the AD brain, especially in the C&H and TH subclusters (Fig. 4c). These observations were validated by the CosMx data, in which the signal intensities of the DAM markers were greater for microglia located in the C&H and TH regions of the AD brain (Fig. 4d,e).

Furthermore, GSEA identified a strong enrichment of immune response pathways in AD-specific microglia, consistent with previous scRNA-seq studies of DAM populations[41–43] (Extended Data Fig. 5e and Supplementary Table 8). Given the strong colocalization of DAMs and β-amyloid noted in previous studies[42–45], β-amyloid IHC staining was performed on both samples using the adjacent section (Extended Data Fig. 5f and Methods). β-Amyloid signals were substantially higher in the C&H, FT and TH regions of the AD brain, with the remaining regions showing no significant difference from regions without microglia (Extended Data Fig. 5g and Supplementary Table 9). Together, these results support that the AD-specific microglia in the C&H and TH regions were indeed the DAM population.

Unsupervised clustering of the iscRNA data for the AD sample identified C&H and TH subclusters of DAMs, indicating DAM regional specification (Extended Data Fig. 6a). This regional specification was supported by a further validation that used CosMx data as the ground truth. NetBID2 analysis identified DEGs for the C&H DAM and TH DAM subclusters (Extended Data Fig. 6b,c and Supplementary Table 10); these were then used to define signature C&H and TH scores that were applied to both the CosMx and iscRNA data. As expected, the iscRNA and CosMx data for C&H DAM showed higher C&H scores, and both datasets for TH DAM showed higher TH scores (Fig. 4f–h). GSEA showed that C&H DAM exhibit upregulation of genes related to the immune response, indicating greater immune activation in the CTX and HPF regions of the AD sample (Extended Data Fig. 6d and Supplementary Table 11). This finding was consistent with the greater relative accumulation of β-amyloid that we observed in the CTX and HPF regions of the AD sample[46,47]. Additionally, the microglia in the simulated iscRNA data were clustered together and had a seemingly random distribution, confirming that the results were authentic and not artifact related (Extended Data Fig. 6e,f).

## Spotiphy charts tumor–TME spatial domains in breast cancer

To evaluate Spotiphy's ability to characterize tumors and the TME, publicly available scRNA-seq and Visium datasets[48,49] for breast tissue samples (BCCID4535, BC1160920F and BCCID44971) were applied to Spotiphy and produced iscRNA data for a total of 23,200 cells (Supplementary Fig. 13, Supplementary Table 12 and Methods). Unsupervised clustering analysis of these iscRNA data revealed multiple subclusters for luminal hormone-responsive (LumHR) cells and luminal secretory (LumSec) cells (Fig. 5a,b). To test if the iscRNA data could distinguish tumors from normal cells, we applied inferCNV analysis[50–54] to both Visium and iscRNA data (Fig. 5c and Supplementary Fig. 14). Although the spot-level Visium data can reveal notable copy number variations (CNVs) in both tumor samples, obvious intraheterogeneity in the CNV patterns was also observed in BC1160920F (Supplementary Fig. 14). Due to resolution limits, we cannot determine which cells contribute to this variation. By contrast, the inferCNV results of the iscRNA data showed that LumHR cells from BCCID4535 and LumSec cells from BC1160920F possessed more CNVs, indicating their greater likelihood of being tumor cells (Fig. 5c). Both LumHR and LumSec cells from BCCID44971

**Fig. 5 | Spotiphy reveals tumor–TME changes across spatial domains in human breast tissue. a,b,** UMAP of 23,200 cells from iscRNA data produced by applying Spotiphy to human breast tissue Visium data. Cells are color coded according to their cell type (**a**) or sample of origin (**b**). LumSec and LumHR cells are encircled by dashed black lines in **a**. **c,** Results of applying inferCNV to the iscRNA data in **a**. Tumor cells originating from BC1160920F (LumSec) and BCCID4535 (LumHR) are encircled by dashed black lines. **d,e,** Transcriptomic spots from the Visium data are color coded according to their constituent LumSec cell sample of origin: BCCID44971 (**d**) and BC1160920F (**e**). The opacity of the spots is representative of cell density; scale bars, 500 μm. **f,** Violin plots of the N-Sec and T-Sec scores for the LumSec cells identified in the iscRNA data from normal sample BCCID44971 and tumor sample BC1160920F. All *P* values are $<2.2 \times 10^{-16}$. **g,** Violin plots of the N-Sec and T-Sec scores for the LumSec cells identified in the Resolve smFISH data

of normal sample P69-S3 and tumor sample P35-S1; top, $P = 3.3 \times 10^{-2}$; bottom, $P = 2.3 \times 10^{-5}$. **h,** Visualization of signature genes in selected regions of the Resolve smFISH data. Blue and red represent the signature genes used to calculate the N-Sec and T-Sec scores, respectively. Cell boundaries are depicted with gray lines; LumSec cells are outlined in white. Resolve smFISH data include three biological replicates each for P69 and P35 samples; scale bar, 100 μm. **i,** Transcriptomic spots from the Visium data. Spots are color coded according to their constituent LumSec cell spatial domains; scale bar, 500 μm. **j,** Cell-type proportions of the three spatial domains based on the iscRNA data. **k,l,** Results of applying CellChat to the iscRNA data of the cells in the three LumSec spatial domains. **k,** Number of interactions among cell types across three spatial domains. **l,** Interaction strength among cell types across three spatial domains. Data were analyzed by two-sided *t*-tests (**f** and **g**); \*\*\**P* < 0.01 and \*\**P* < 0.05.

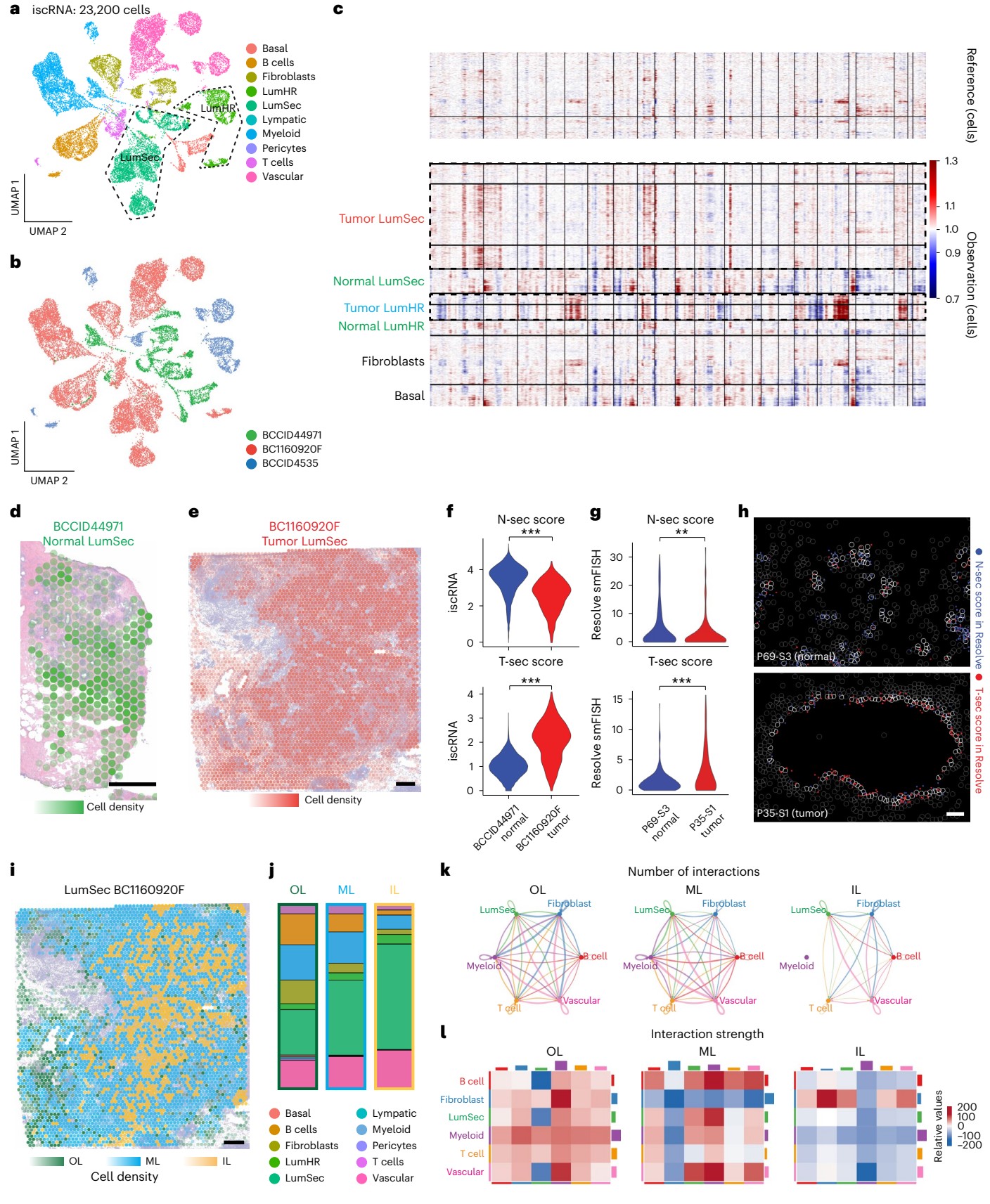

exhibited fewer CNVs, indicating that they are normal cells (Fig. 5d,e and Extended Data Fig. 7a,b). This observation was consistent with the original study's conclusion that BCCID44971 was normal breast tissue[49]. Similarly, iscRNA data successfully identified epithelial cells as the primary tumor cells in the Visium datasets of lung cancer and colorectal cancer (Supplementary Fig. 15).

We subsequently validated the tumor/nontumor identities of luminal cells (as determined by the iscRNA data) using Resolve smFISH data from tumor (P35-S1) and normal (P69-S3) breast tissue samples[48]. We selected the DEGs for tumor (BC1160920F) and normal (BCCID44971) LumSec cells and then used them to define tumor LumSec (T-Sec) and normal LumSec (N-Sec) scores, which we subsequently applied to the iscRNA and smFISH datasets (Extended Data Fig. 7c,d and Supplementary Table 13). Indeed, genes that showed elevated expression in tumor LumSec cells also produced greater signal intensity in the smFISH data for the tumor sample (P35-S1); the same was observed for the N-Sec genes (Fig. 5f–h). Similarly, the top DEGs for tumor (BCCID4535) LumHR cells were also found to show elevated expression and greater signal intensity in both the iscRNA data and the smFISH data for the tumor sample (P35-S1), respectively (Extended Data Fig. 7e–i and Supplementary Table 14). Furthermore, GSEA detected elevated immune activation features in cells originating from the tumor samples (BC1160920F and BCCID4535; Extended Data Fig. 7j,k and Supplementary Tables 15 and 16), supporting Spotiphy's accuracy in differentiating between tumor cells and normal cells.

Three LumSec tumor cell subclusters in BC1160920F exhibited unique localizations, distributed from the center to the periphery of the section (Fig. 5i and Extended Data Fig. 8a). We categorized spots into three 'spatial domains' based on the subcluster of their LumSec cells, termed as outer layer (OL), middle layer (ML) and inner layer (IL; Fig. 5j). We labeled cells by spatial domain and divided the iscRNA data into three domain-specific subsets. Similarly, we defined OL and IL spatial domains in BCCID4535 based on LumHR tumor cell subclusters (Extended Data Fig. 8a–c). The cell-type proportions varied across the spatial domains, with tumor cells (LumSec or LumHR cells) increasing and immune cells (T, B and myeloid cells) decreasing toward the center (Fig. 5j and Extended Data Fig. 8c). To better understand these variations, we used CellChat[55,56] to compare the tumor–TME communication patterns across these spatial domains. Dramatic differences were observed in the tumor–TME interactions in both BC1160920F and BCCID4535 (Fig. 5k,l and Extended Data Fig. 8d,e). Interestingly, the interaction strength (per CellChat) between tumor and immune cells did not always correspond to the cell-type proportions of the spatial domain. For LumSec cells in BC1160920F, the strongest immune reactions were detected in the ML domain, not the OL domain with the most immune cells (Fig. 5l). Interactions between the collagen family and *SDC1/SDC4*, as well as between MDK and *SDC1/ SDC4*, are associated with cell mobility and were notably increased in the ML domains[57,58]. Both pairs were upregulated in the ML domain of BC1160920F and the OL domain of BCCID4535 (Extended Data Fig. 8f,g), indicating that cell mobility may play a role in the infiltration of immune cells into tumors.

To highlight the uniqueness of Spotiphy's decomposition function, we performed a comprehensive decomposition benchmarking of Spotiphy against Tangram[25], SpatialScope[29] and iStar[32] using simulated Visium datasets of the mouse brain (Supplementary Fig. 16). Four methods delivered cell-type-level expression matrices for each spot. Spotiphy consistently achieved the highest correlation and cosine similarity and the lowest absolute error, mean square error and JSD compared to the ground truth. Spotiphy also delivered the highest Matthews correlation coefficient for each spot. In addition, Spotiphy requires significantly less computation time than its competitors. Considering that a typical Visium sample usually contains 3,000 to 4,000 spots[7], this is a major advantage, allowing Spotiphy to deliver reliable results in a shorter time frame.

Another strategy for decomposition benchmarking involves merging the iscRNA data into pseudo-Visium data according to the spot origin of cells and then comparing it with the original inputs (Supplementary Fig. 17). Because Spotiphy's decomposition strategy involves partitioning gene counts into different cells, merging its iscRNA data results in output that is almost identical to the Visium input, achieving the highest overall Pearson's correlation coefficient. iStar also delivers good performance with the second highest Pearson's correlation coefficient. Tangram and SpatialScope, however, demonstrated less than ideal results. We then examined the spatial distribution of six spatial variable genes in the pseudo-Visium data. Spotiphy's output accurately preserved the original distribution patterns. By contrast, SpatialScope, limited to decomposing marker genes from its early phase, only provided the distribution of two genes. Although Tangram's 'iscRNA data' offer whole-genome coverage, it distorted the distribution of two genes and exhibited less overall accuracy. Collectively, these results strongly illustrate Spotiphy's advancement, accuracy and effectiveness and bring single-cell resolution to whole-transcriptomic ST data, preserving the unique distribution patterns of spatial variable genes and creating opportunities for new insights.

### Spotiphy creates pseudo-single-cell whole-slide images

Spotiphy addresses the other major drawback of sequencing-based ST approaches, specifically information loss in noncapture areas (out-spot space). Visium[7] and DBiT-Seq[8] typically sequence about 50% of the entire section. In both mouse brain Visium datasets, only 34% of cells were within the spots (in-spot space), leading to profound information loss (Extended Data Fig. 9a). After analyzing the seamless image-based Xenium data of the mouse brain, human lung cancer and human colorectal cancer, we observed that within small scales (<100 μm), cellular proportion changes are minimal. This allows us to impute the information for noncapture areas based on the nearby regions (Supplementary Fig. 18). In fact, cells of the same type typically exhibit a gradually uniform distribution within a tissue section to maintain tissue function[59–61], a pattern that has also been observed in MERFISH data[62]. Most genes mirror the steady and consistent distribution, particularly those involved in maintaining basic cellular functions and tissue architecture[63,64]. Consequently, Spotiphy imputes cell-type proportions of out-spot space via Gaussian processes (Extended Data Fig. 9b,c).

We then validated the imputation's accuracy using two fields of view (FOVs) located in the CTX and HPF in matched Visium and Xenium data from AD brain samples. CytoSPACE[23] and Tangram[25] were used to benchmark Spotiphy's performance (Fig. 6a and Extended Data Fig. 9d). Coloring by cell types, Spotiphy yields pseudo-single-cell-resolution images that are highly consistent with the ground truth (Fig. 6b and Extended Data Fig. 9e). All methods delivered reasonable cell-type proportions for the in-spot space, with Spotiphy's results being closest to the ground truth (Fig. 6c,d and Supplementary Table 17). Intriguingly, the cell-type proportions for the out-spot space inferred by Spotiphy were closer to the ground truth than the Tangram- and CytoSPACE-generated data for the in-spot space. Furthermore, Spotiphy's merged cell-type proportions for both the in-spot and out-spot spaces were consistent with the accuracy of the in-spot results. These findings support imputation via Gaussian processes as an approach to reconstructing the whole section from sequencing-based ST spot grids. Spotiphy also provides kernel density smoothing as an alternative to estimate the transcriptomic profiles of cells in noncapture areas (Methods).

### Spotiphy overcomes in situ single-nuclei sequencing limits

Slide-tags is another sequencing-based platform that was recently introduced[11]. Slide-tags labels individual nuclei in a whole section with unique barcodes, generating snRNA-seq data with high gene coverage along with spatial information. Although Slide-tags represents an important innovation in the ST field, its practicality is limited by the method's substantial loss of nuclei (approximately 75%)[11]. Similar to how

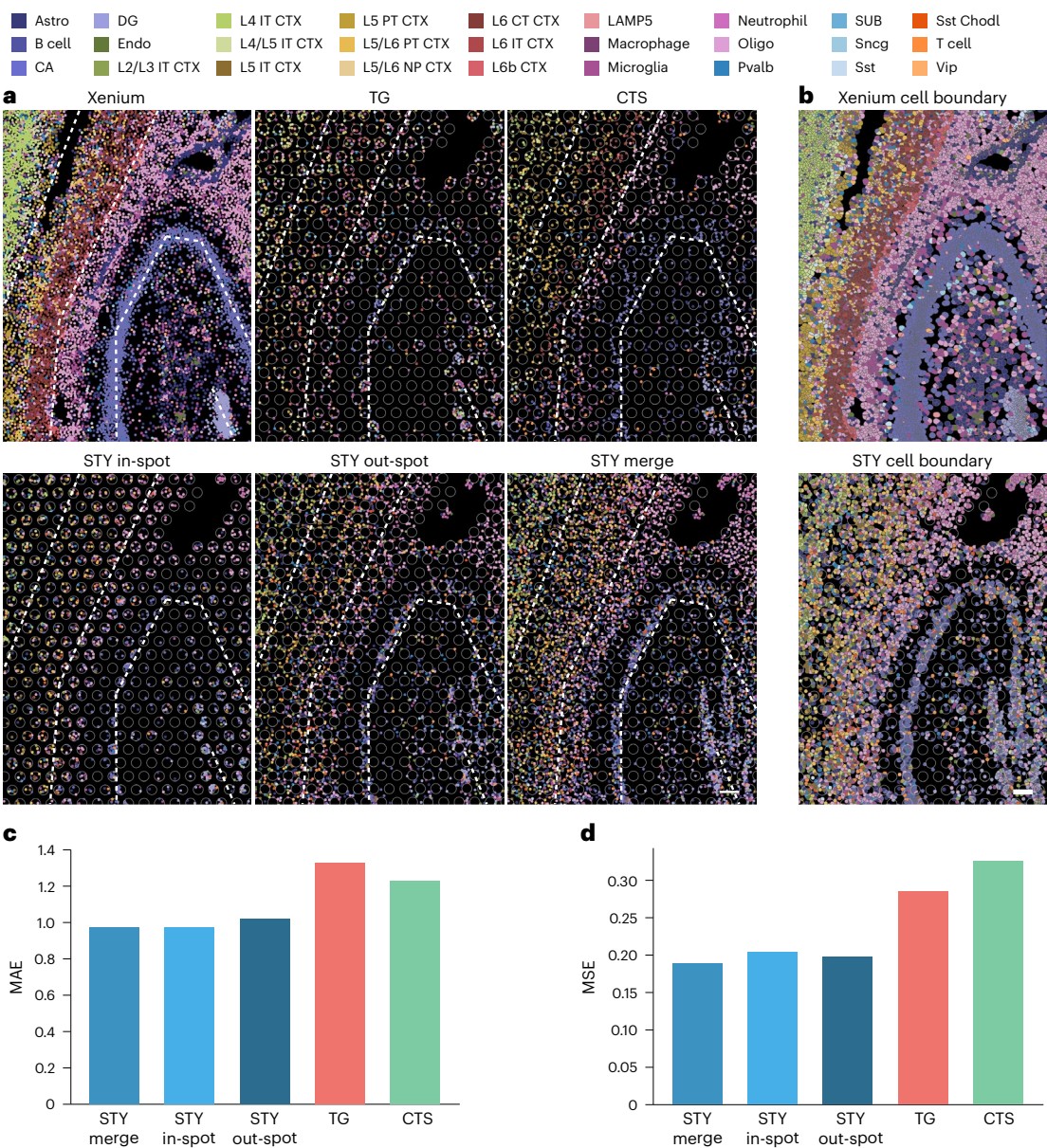

**Fig. 6 | Spotiphy outputs pseudo-single-cell-resolved whole-transcriptome images. a,b,** Cell-type annotation of individual nuclei and cells in selected FOVs in the HPF. Dots in **a** represent all detectable nuclei color coded for cell type. Gray circles represent the transcriptomic spots in the Visium data. Dashed white lines have been added to allow for easier comparison. Cells in **b** are color coded by cell type. The cell boundaries depicted for the Xenium (top) and Spotiphy (bottom) data were inferred through those methods' respective pipelines; scale bars, 100 μm; Astro, astrocytes; Oligo, oligodendrocytes. **c,d,** The mean absolute error (MAE) and mean squared error (MSE) of the cell-type imputation generated by Spotiphy, Tangram and CytoSPACE for each cell. Xenium data were used as the ground truth; STY in-spot, image with only cells inside the spots; STY out-spot, image with only cells outside the spots; STY merge, image with all cells included.

Spotiphy's imputation function recovers data from noncapture areas in spot-based ST, treating each nucleus in Slide-tags data as a 'spot' enables us to retrieve the lost snRNA-seq data. To pinpoint the 'missing' nuclei that Slide-tags has not labeled, we aligned the hematoxylin and eosin (H&E) image of mouse HPF data (839 nuclei) from the original paper with the image of the same brain region in our AD brain, wherein 3,193 nuclei were detected (Extended Data Fig. 10a–c and Methods). Spotiphy generated iscRNA data for 3,193 nuclei using Slide-tags snRNA-seq data via the imputation function. Cells from both datasets mixed well by cell type after unsupervised clustering, indicating that Spotiphy's results were accurate and free of batch effects (Extended Data Fig. 10d,e). iscRNA data displayed a similar expression pattern of marker genes identified by Slide-tags snRNA-seq data (Extended Data Fig. 10f). The cell-type proportions obtained from Slide-tags snRNA-seq data and iscRNA data for the HPF were also concordant, further supporting the accuracy of

Spotiphy's inference (Extended Data Fig. 10g). Although no 'new' data was generated, Spotiphy was able to increase the number of nuclei for each cell type, providing sufficient statistical power for downstream analysis. Whether these additional pseudocells could improve the accuracy of cellular proportion analysis is worth further exploration.

Together, the results reported herein demonstrate that Spotiphy effectively addresses the key challenges of sequencing-based ST approaches, delivering accurate and biologically meaningful single-cell-resolved whole-transcriptomic ST data of all detectable cells. These data are compatible with conventional scRNA-seq analysis algorithms and tools.

## Discussion

Spotiphy is an integrated method that marries the genome-wide coverage of sequencing-based ST approaches with the single-cell resolution

of image-based methods. Spotiphy offers a wider array of features (Supplementary Fig. 19a). Spotiphy's superior inference of cell-type proportions[21–25] largely owes to its improved marker gene selection and generative modeling approach. Rather than simply predicting the cell-type proportions of transcriptomic spots, Spotiphy models the gene counts for each cell in each spot; in doing so, it not only greatly increases the information density but also brings single-cell resolution to sequencing-based ST data. By contrast, Cell2location[21] and RCTD[22] use probabilistic models without explicitly incorporating decomposed expression into their models. CytoSPACE[23], CARD[24] and Tangram[25] approach deconvolution from an optimization perspective, attempting to identify the combination of cells in the scRNA-seq reference whose collective expression most closely resembles that of each spot in ST data. They face two drawbacks. (1) Output accuracy heavily relies on the quality of reference data. If the reference lacks or contains only a limited number of the correct cell types, the algorithm would produce results deviating from the ground truth. This explained their poor performance for immune cell populations. (2) No additional information is gained from this mapping process, rendering them less powerful than Spotiphy. SpatialScope[29] uses a strategy similar to RCTD[22] for cell-type proportion deconvolution and enhances spot expression assembly with pseudocells that closely resemble the scRNA-seq reference, as generated by a deep learning model. However, it confines the gene expression for each cell type to a predetermined range that aligns with the scRNA-seq reference, making it difficult to parse cellular regional specification. Spotiphy uses a Bayesian approach to dissect the expression of individual genes and assign them to specific cell types, producing iscRNA data that more accurately reflect expression fluctuations within transcriptomic spots and reveal cellular regional specifications. Capturing region-specific cell subtypes through conventional single-cell transcriptomics technologies is challenging for several reasons. (1) The single-cell sampling may not cover enough regions of tissue, where ST data usually have full coverage of the whole-tissue block. (2) The population of cells, particularly rare types of cells, is often too small to detect these intravariations for one single-cell profiling. By contrast, Spotiphy can generate an average of about 10,000 cell expression profiles from a single tissue section (whereas one scRNA-seq sample typically has around 5,000 cells on average). (3) It is impossible to determine whether subclusters of certain cell types observed in scRNA-seq data are due to regional variations without the spatial information unique to ST data. Spotiphy can recover the information loss for noncapture areas via Gaussian processes (Supplementary Fig. 20). The insights into the cellular organization, heterogeneity and function within complex biological systems provided by Spotiphy will deepen our understanding of the molecular mechanisms underlying AD pathogenesis and the pathogenesis of solid tumors.

Spotiphy's results are reliant on the quality of input. Because the decomposition applies to every gene, the iscRNA data will retain the gene coverage of the ST data. Therefore, sequencing-based ST datasets are well suited for Spotiphy, mainly because of their whole-transcriptomics coverage. Other approaches have enhanced their resolution but at the expense of gene coverage. For example, Slide-Seq v1/Slide-Seq v2 achieve a resolution of approximately 10 μm but provide gene coverage of $10^1$-$10^3$ (refs. 9,10); Stereo-seq offers subcellular-resolution spot size (0.2 μm) with a gene coverage of $10^2$-$10^3$ (ref. 12). Data generated through these approaches do not stand to greatly benefit from the application of Spotiphy. At present, pairing gene coverage-prioritizing approaches (for example, Visium and DBiT-Seq) with Spotiphy may yield the best outcomes; the data discussed in the Results had a relatively high gene coverage of $10^3$-$10^4$ at single-cell resolution (Supplementary Fig. 19b).

Spotiphy's performance is also contingent on the quality of the scRNA-seq reference for its generative model. An scRNA-seq dataset derived from the same sample as the ST data would be ideal to avoid batch effects. However, acquiring such matched datasets in the real world is both technically challenging and financially cumbersome. To accommodate unmatched datasets, Spotiphy assigns variables (batch prior) to each gene to model the differences between ST and scRNA-seq profiles, effectively mitigating batch effects on deconvolution and decomposition. Spotiphy also presets four additional hyperparameters, including the quantile parameter, fold change threshold, nonzero count cell percentage and number of marker genes, to ensure scRNA-seq reference accuracy and robustness (Supplementary Fig. 1). In fact, the scRNA-seq reference used for mouse brains was primarily sourced from Allen Brain Map Atlas[36], demonstrating that Spotiphy can produce reliable deconvolution results even when the reference originates from nonmatched tissues. This flexibility greatly broadens Spotiphy's potential applications, allowing for more uses in various research contexts without the need for matched reference datasets.

Spotiphy produces single-cell spatial whole-transcriptomic profiles, but it is not the only solution. Emerging methods like iSpatial[65] and Liger[66] aim to generate the same results from image-based ST data, which could be worth exploring, but with some caveats. The major limitation of image-based ST approaches is the low-plexity of their predesigned gene panels[67]. These methods use scRNA-seq data to predict the expression of nontargeted genes, a process complicated by the different measurement techniques used by scRNA-seq and image-based ST to quantify gene expression. For scRNA-seq, the raw count is a measure of the free mRNA molecules within an individual cell[68], whereas the raw counts produced by image-based ST are a measure of the positive fluorescent signals within regions around the nuclei[69]. This fundamental discrepancy greatly diminishes the usefulness of scRNA-seq data as a reference, and minimizing these 'technique-derived batch effects' is challenging. Additional limitations make image-based ST data less suitable for inference. First, the fluorescence signal intensity is highly influenced by probe affinity, which varies across probe sets. Second, different excitation wavelengths will exhibit different signal intensities, complicating signal quantification. Third, image-based ST data tend to be sparser due to differences in the sensitivity of different probes. Together, these issues make it more difficult to accurately compare expression levels between different genes.

Potential future directions for Spotiphy include the implementation of five improvements. First, a deep learning-based vision transformer model that correlates cell-type information with image features would improve Spotiphy's performance by taking advantage of extra information from histological images. Currently, Spotiphy assigns a cell type and corresponding iscRNA profile to a random nucleus within each spot. Although this random assignment is confined to a small region (50-100 μm), it poses limitations in improving the accuracy of cell-type assignment. This solution, which requires the use of massive training datasets, is contingent on the increased availability of sequencing-based ST data in the future[70]. It is also worth noting that for users studying cell–cell interactions at a fine scale, vision transformer modeling has the potential to increase inference accuracy. By leveraging additional image features, cell-type information and iscRNA data can be effectively assigned to the corresponding cells. A second exciting extension for Spotiphy involves reconstructing the three-dimensional (3D) structure of tissue. Recent studies have shown that profiling sequential sections with ST can reveal the real 3D organization of tissues[71,72]. Yet, it is costly and not feasible for broad use, which is where Spotiphy can step in. By leveraging the similarities in image features from adjacent sections, Spotiphy could infer the cellular proportions and expression profiles of neighboring sections, ultimately reconstructing the 3D structure with single-cell-resolved profiles. STalign[73] would be of great assistance for ensuring alignments between adjacent sections. Another opportunity for improvement resides in Spotiphy's inability to identify cell types that are present in the ST data but not in the scRNA-seq reference. We believe that this shortcoming owes primarily to the algorithm's inability to discern whether uncharacterized gene expression is indicative of an as of yet

undefined cell type or simply a biological change in an existing cell type. In other words, the ease and accuracy of predictions could be improved if the gene coverage of sequencing allowed for a clearer distinction between cell subtypes and even substates. We therefore recommend including as many cell types as possible in the construction of the scRNA-seq reference to optimize the precision of Spotiphy's results. Furthermore, with the continuous advancement of single-cell sequencing technologies, we aim to enhance Spotiphy by incorporating inputs from scMultiome (scRNA and scATAC)[74] and Paired-Tag (scRNA and scChIP)[75] profiles with spatial-ATAC-seq[76] and spatial-CUT&Tag[77], which have yielded spatially resolved chromatin modification and epigenome profiles. This expansion would enable Spotiphy to integrate and analyze multimodal data, providing a more comprehensive and detailed understanding of spatially resolved cellular states and regulatory mechanisms. Last, improvement in the resolution of ST platforms holds the potential to enhance existing methods that directly use ST data for principal component analysis and clustering analysis[78,79]. By leveraging the iscRNA data generated by Spotiphy as input, these approaches will achieve more accurate and comprehensive output results at both gene and cellular levels. Their outputs, including DEGs and neighborhood signatures, will provide more realistic guidance for future experimental validations.

## Online content

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

## Methods

### Sample information and processing

**Animals.** Four-month-old 5xFAD and C57BL/6J (all Jackson Laboratory) mice were used in this study. All animals were housed within the vivarium at St. Jude Children's Research Hospital or Thomas Jefferson University and maintained on a 12-h light/12-h dark cycle with ad libitum access to food and water. All experimental procedures using animals were performed in accordance with the National Institutes of Health Guide for the Care and Use of Laboratory Animals, and all protocols were approved by the St. Jude Children's Research Hospital (protocol 542) Institutional Animal Care and Use Committee. Experiments were performed in accordance with the Code of Ethics of the World Medical Association (Declaration of Helsinki) for animal experiments.

**Tissue preparation.** Mice were deeply anesthetized with Avertin and perfused with ice-cold 1× PBS (pH 7.4) to flush circulating blood cells from the brain. The exsanguinated brain was quickly removed, and the left hemisphere was dissected and placed in ice-cold HBSS (Gibco, Life Technologies). The right hemisphere was embedded in OCT and kept at −80 °C until use. Embedded brains were cut from the midline to the lateral side in laterally at the sagittal plane level in 10-µm-thick sections or as required by the cryostat.

### Single-nucleus isolation and multiome profiling

The left hemisphere was used for profiling using a 10x Genomics Chromium Single Cell Multiome ATAC + Gene Expression kit. A small piece of the tissue was flash-frozen and used for direct nuclei isolation by following the 10x Genomics demonstrated protocol Nuclei Isolation from Complex Tissues for Single Cell Multiome ATAC + Gene Expression Sequencing (CG000375 Rev A). Nuclei were counted by staining with Trypan Blue on a hemocytometer, and the concentration was adjusted to 1,000 nuclei per µl. The remaining larger piece of the tissue sample was dissociated to yield single cells using papain-based dissociation solution on a GentleMACS Octodissociator at 37 °C for 20 min. After filtering the dissociated sample, neuronal and immune cells were separated from unwanted myelin using density gradient centrifugation with 22% Percoll. The pellet containing microglia, lymphocytes and nonimmune neuronal cells like astrocytes, oligodendrocytes and neuronal stem cells was washed in HBSS buffer and stained with CD45 (1:100) and CD11b (1:100) to enrich immune cell populations using a FACS Aria III Cell Sorter. After isolating CD45+CD11b+ microglia, CD45+ other immune cells and double-negative neuronal cells were counted using Acridine Orange/propidium iodide staining on a Luna FL Cell Counter. In total, 15,000 neuronal cells, 10,000 microglia and all cells of the immune population were mixed and used for nuclei isolation using 10x Genomics Demonstrated Protocol Nuclei Isolation for Single Cell Multiome ATAC + Gene Expression Sequencing (CG000365 Rev A). Nuclei were counted by staining with Trypan Blue on a hemocytometer and adjusted to a concentration of 2,000 nuclei per µl. Nuclei isolated from both pieces were mixed such that 5,000 nuclei were targeted from nuclei isolated from enriched cell populations and 1,000 nuclei were targeted from nuclei obtained directly from frozen sections. The mixed nuclei population was tagmented and immediately loaded on to Chip J to be partitioned into GEMS using 10x Genomics User Guide Chromium Next GEM Single Cell Multiome ATAC + Gene Expression (CG000338 Rev C). Partitioned nuclei underwent barcoding, reverse transcription, cDNA amplification and sample index ligation, and libraries were made for both ATAC-seq and gene expression sequencing. Libraries were quantified using a D5000 kit on an Agilent Tapestation and Illumina MiSeq platform. Quantified libraries were then sequenced on an Illumina NovaSeq 6000.

### Sequencing-based ST profiling (10x Genomics Visium)

The right hemisphere was embedded in OCT (Tissue-Tek, Sakura) after removing excess PBS. The tissue was cryosectioned as per the tissue preparation guide from Visium Spatial Gene expression (10x Genomics, CG000160). Briefly, OCT-embedded tissue blocks were sectioned (10 µm) and placed within the capture area of the Visium spatial slide (10x Genomics, PN-1000184). The slides were then permeabilized for 18 min at 37 °C as determined with the Visium Tissue optimization procedure. For Visium Spatial Gene expression, brightfield images were acquired using an AxioScan Z.1 Whole-Slide scanner and exported into JPG files. cDNA libraries were generated according to the Visium Spatial Gene Expression User Guide (10x protocol, CG000239). The libraries were loaded and sequenced on an Illumina NovaSeq 6000 following the recommendation's of the 10x Genomics sequencing parameters. Raw data were converted into FastQ and matrices of expression using Space Range software V1.0 provided by 10x Genomics.

### Image-based ST profiling (10x Genomics Xenium and NanoString CosMx)

Serial sections of mouse brain tissue were sent to 10x Genomics and NanoString for processing. A 298-plex mouse neuron panel was used for 10x Genomics Xenium profiles, and a 1,000-plex mouse panel was used for NanoString CosMx profiles (Supplementary Table 18).

### β-Amyloid staining

Ten-micron serial sections of mouse brain tissue were generated for IHC staining using X-34 targeting β-amyloid plaques. High-resolution images of H&E staining were acquired using a Zeiss system. By aligning the IHC images with Visium images based on histological features, we can further quantify the fluorescence intensity in different brain regions.

### Mouse brain scRNA-seq reference

scRNA-seq data from 1,228,636 cells from Yao et al.[36] were used to identify 22 cell types in the mouse brain. The whole dataset was downsampled to 5,000 random picked cells for cell types with numbers over 10,000. In total, 103,281 cells were selected and further merged with 49,406 cells of seven cell types from the original generated dataset.

### Human breast datasets

scRNA-seq data from 126 specimens from Kumar et al.[48] were used to construct the single-cell reference with ten major cell types (Supplementary Fig. 13a–c). Three 10x Genomics Visium profiles from Wu et al.[49] were used as ST inputs. BCCID4535 and BC1160920F are labeled as tumor samples, and BCCID44971 is labeled a normal control in the original paper[49]. Spotiphy was performed on three Visium samples separately (Supplementary Fig. 13d). Resolve smFISH data from samples P35-S1 (tumor) and P69-S3 (normal) from Kumar et al.[48] were used as the ground truth for downstream analysis and validation. The original data links are available in the Data availability section.

### Clustering based on Seurat and scMINER

All clustering analyses of iscRNA data in this study were performed using scMINER[39]. Clustering of the mouse brain scRNA-seq reference was performed using Seurat[80] for batch correction with harmony[81], and clustering of the human breast scRNA-seq reference was performed using Seurat and the same parameters as those used in the original paper[48].

### Downstream analyses and signature score definition

Differential expression analysis was performed using the scMINER getDE() function (https://jyyulab.github.io/scMINER). getDE() uses the 'limma (v-3.60.4) R package' as the default method. The GSEA in this study was conducted using the NetBID2 (ref. 40) funcEnrich.Fisher() function and visualized with the draw.funcEnrich.cluster() function (https://jyyulab.github.io/NetBID). The top DEGs were selected (Supplementary Table 19), and their average values were used as a signature score through the Seurat[80] AddModuleScore function.

## Tumor identification by inferCNV

The tumor–normal distinction in this study was performed using the inferCNV package[50-54] with default settings. It is worth mentioning that the inferCNV package is sensitive to the choice of normal cell reference. Therefore, it is important to select the appropriate normal cell reference in the iscRNA data to ensure proper result interpretation.

## Cell–cell communication

Cell–cell communication analysis and comparisons were performed using the CellChat package[55,56] with default settings.

## Spotiphy pipeline

**Introduction.** For a more detailed description of the Spotiphy pipeline, see the Supplementary Methods. In summary, Spotiphy integrates probabilistic modeling, statistical testing and computer vision techniques to (1) estimate the proportion of cell types at each capture area (for example, circular spots) of the tissue, (2) decompose the spatial expressions of the multicell mixtures into the single-cell level and (3) generate pseudo-single-cell-resolution images. The pipeline takes three inputs. The first input is an untransformed spatial expression count matrix $X \in \mathbb{R}^{S \times G}$ of locations $s \in \{1, 2, \cdots, S\}$ and genes $g \in \{1, 2, \cdots, G\}$, which can be obtained by using sequencing-based ST technologies, such as 10x Genomics Visium and Slide-Seq. The second input is the scRNA count matrix $Y \in \mathbb{R}^{C \times G}$ of cells $c \in \{1, 2, \cdots, C\}$ and genes $g \in \{1, 2, \cdots, G\}$, which is normalized by the counts per million method. All cell types need to be annotated, and we let $\tau(c) \in \{1, 2, \cdots, T\}$ denote the type of cell $c$, where $T$ is the number of cell types. The last input is the high-resolution H&E-stained image, which is used to determine the location of each nucleus and the number of nuclei in each capture area. Note that the H&E-stained image is not needed when estimating cell-type proportions.

**Marker gene selection.** Let $z_{t,g}$ and $\hat{z}_{t,g}$ denote the average expression of gene $g$ in type $t$ cells and the corresponding estimation based on scRNA matrix $Y$. Let $F_{g,t} = \{\hat{z}_{t,g}/\hat{z}_{t',g}, |, t' \neq t\}$ denote the set of fold change values for gene $g$ when comparing gene expression in cell type $t$ with each other cell type. Let $f_{g,t}(\nu)$ denote the $\nu$th sample quantile for the values in set $F_{g,t}$. Additionally, with the scRNA matrix $Y$, we can conduct a $z$ test for the null hypothesis $H_0 : z_{t,g} \leq z_{t',g}$ and alternative hypothesis $H_1 : z_{t,g} > z_{t',g}$, where the $P$ value is denoted as $\lambda_{g,t,t'}$. Let $w_{t,g}$ denote the coverage rate of gene $g$ in type $t$ cells, that is, $w_{t,g}$ is the percentage of cells that have nonzero expression of gene $g$. With the parameter $\nu$ and thresholds $l_{\text{fold}}$, $l_\lambda$ and $l_{\text{cover}}$, gene $g$ is selected as a candidate of type $t$ marker when the following three conditions are satisfied:

- $f_{g,t}(\nu) > l_{\text{fold}}$,
- $\max\{\lambda_{g,t,t'} | t' \neq t\} < l_\lambda$ and
- $w_{t,g} > l_{\text{cover}}$.

These conditions ensure that the candidate genes are effective in distinguishing type $t$ cells from all other cell types. Finally, if cell type $t$ has more than $n_{\text{select}}$ candidate genes, we rank them based on the fold change quantile $f_{g,t}(\nu)$ and only select the top $n_{\text{select}}$ genes. We repeated this process for all cell types and aggregated the selected marker genes. By only keeping the marker genes in matrices $X$ and $Y$, we obtain matrices $X^{(m)} \in \mathbb{R}^{S \times G_m}$ and $Y^{(m)} \in \mathbb{R}^{C \times G_m}$, where $G_m$ is the total number of selected marker genes, and the superscript 'm' means that only the marker genes are considered.

**Construction of the scRNA-seq reference matrix.** We let $\Phi^{(m)} \in \mathbb{R}^{T \times G_m}$ be the scRNA reference matrix, where $\varphi_{t,g}^{(m)}$ represents the average proportion of gene $g$ in expression of type $t$ cells. Thus, we have $\sum_{g=1}^{G_m} \varphi_{t,g}^{(m)} = 1$ for $t = 1, 2, \cdots, T$. Reference matrix $\Phi^{(m)}$ is estimated by maximizing the likelihood function, an approach similar to BayesPrism[82], where the results can be expressed as

$$\hat{\varphi}_{t,g}^{(m)} = \frac{\sum_{\{c|\tau(c)=t\}} y_{c,g}^{(m)}}{\sum_{\{c|\tau(c)=t\}} \sum_{g'=1}^{G_m} y_{c,g'}^{(m)}}.$$

**Probabilistic generative model and estimation of cell-type proportions.** To model the spatial expression matrix $X^{(m)}$, we let $Q \in \mathbb{R}^{S \times T}$ be the score matrix of gene contribution, where the gene contribution score $q_{s,t}$ denotes the proportion of genes at location $s$ that are contributed by type $t$ cells. Thus, $\sum_{t=1}^T q_{s,t} = 1$ for $s = 1, 2, \cdots, S$. The proportion of gene $g$ at location $s$ can be derived as

$$\rho_{s,g} = \sum_{t=1}^T q_{s,t} \cdot \varphi_{t,g}^{(m)}.$$

However, because the scRNA data and ST data are obtained using different technologies, the batch effect between these two data types cannot be ignored. To quantify the batch effect, we introduced batch effect parameters $r = \{r_1, r_2, \cdots, r_G\}$ to the probabilistic generative model and adjust $\rho_{s,g}$ as

$$\tilde{\rho}_{s,g} = \frac{\rho_{s,g} \cdot 2^{r_g}}{\sum_{g'=1}^{G_m} \rho_{s,g'} \cdot 2^{r_{g'}}}.$$

We then assumed that conditioned on score matrix $Q$, batch effect parameters $r$ and the scRNA reference matrix $\Phi^{(m)}$, the spatial expression at location $s$ follows a multinomial distribution,

$$x_s^{(m)} \left| Q, r, \Phi^{(m)} \sim \text{Multinomial}\left(m'_s, [\tilde{\rho}_{s,1}, \tilde{\rho}_{s,2}, \cdots, \tilde{\rho}_{s,G_m}]\right),\right.$$

where $m'_s$ is the total gene count at location $s$. For full details of the generative model, see the Supplementary Methods.

In Spotiphy, we applied variational inference to approximate the posterior distribution of score matrix $Q$ and batch effect parameters $r$. This approach was implemented using the Python package Pyro[83], which supports GPU acceleration. The posterior mean values were then used as the estimation of the unknown parameters.

**Nucleus segmentation and inference of cell boundaries.** In the Spotiphy pipeline, we adopted the pretrained deep learning model from Stardist[84,85] to segment the nuclei in H&E-stained images. With the segmentation results, we assumed that a cell was at location $s$ if the center of the cell's nucleus was inside the capture area. We let $N_s$ denote the number of cells at location $s$.

**Decomposition of spatial expression.** We first update the estimated cell-type proportions. When H&E-stained images are available, we can leverage the nucleus segmentation results. Specifically, for $t = 1, 2, \cdots, T$, we let $n_{s,t}$ denote the number of type $t$ cells at location $s$. These values can be determined by solving the following optimization problem:

$$\min \sum_{t=1}^T \left| \frac{n_{s,t}}{N_s} - p_{s,t} \right|$$

$$\text{s.t.} \sum_{t=1}^T n_{s,t} = N_s$$

Then, the updated cell-type proportions at location $s$ are $\tilde{p}_{s,t} = n_{s,t}/N_s$ for $t = 1, 2, \cdots, T$. When an H&E-stained image is not available, we define the threshold $l_p$ and assume that if $p_{s,t} < l_p$, there is no strong evidence to suggest the existence of cell-type $t$ at location $s$. Consequently, the proportion is updated as follows:

$$\tilde{p}_{s,t} = \begin{cases} 0, & p_{s,t} < l_p \\ \frac{\sum_{t_1=1}^T p_{s,t_1} \cdot I(p_{s,t_1} \geq l_p)}{\sum_{t_2=1}^T I(p_{s,t_2} \geq l_p)} & p_{s,t} \geq l_p \end{cases},$$

where $I(\cdot)$ is the indicator function. After updating the cell-type proportions, we let $\tilde{\boldsymbol{P}} \in \mathbb{R}^{S \times T}$ be the corresponding matrix where the $st$ th entry is $\tilde{p}_{s,t}$.

To make sure that the decomposed spatial expression can facilitate more downstream analysis, we aimed to decompose the spatial expression of all $G$ genes, rather than the $G_m$ marker genes. Thus, we constructed a single-cell reference matrix $\boldsymbol{\Phi} \in \mathbb{R}^{T \times G}$ using the full scRNA count matrix $\boldsymbol{Y}$. In addition, we let $\boldsymbol{U} \in \mathbb{R}^{S \times G \times T}$ denote a 3D tensor, where $u_{s,t,g}$ is the expression of gene $g$ in type $t$ cells at location $s$. As a result, $\boldsymbol{u}_{s,g} = [u_{s,g,1}, u_{s,g,2}, \cdots, u_{s,g,T}]$ is the decomposition of spatial expression $x_{s,g}$. By applying the probabilistic model again, the conditional distribution of $\boldsymbol{u}_{s,t}$ is

$$\boldsymbol{u}_{s,g} \mid \boldsymbol{X}, \boldsymbol{\Phi}, \tilde{\boldsymbol{P}} \sim \text{Multinomial}\left(x_{s,t}, [\omega_{s,g,1}, \omega_{s,g,2}, \cdots, \omega_{s,g,T}]\right),$$

where $\omega_{s,g,t} = \frac{\tilde{p}_{s,t} \cdot \varphi_{t,g}}{\sum_{t'=1}^{T} \tilde{p}_{s,t'} \cdot \varphi_{t',g}}$. Then, $u_{s,t,g}$ can be estimated as the posterior mean: $\hat{u}_{s,t,g} = x_{s,t} \cdot \omega_{s,g,t}$.

**Cell-type annotation in the pseudo-single-cell-resolution images.** Recall that we can estimate the exact numbers of each cell type within each capture area. However, we are not able to identify which specific nuclei belong to each cell type. Thus, given that there are $n_{s,t}$ cells of type $t$ at location $s$, we randomly assign $n_{s,t}$ nuclei as belonging to cell-type $t$ within the capture area. For nuclei outside the capture areas, we used Gaussian processes to estimate the proportion of each cell type in the neighborhood of each nucleus outside the capture areas. In this way, we assign a cell type to each nucleus outside the capture area by randomly sampling from the estimated proportions of different cell types. By annotating the nuclei and inferring the cell boundaries, we obtain a pseudo-single-cell-resolution image that closely resembles the output of image-based ST approaches (Supplementary Fig. 20).

**Evaluation of cell-type proportion estimation using Xenium, CosMx and synthetic data**
**Alignment of 10x Genomics Xenium and NanoString CosMx data to 10x Genomics Visium data.** Alignment is necessary to make the Xenium data, CosMx data and Visium data comparable. To achieve this goal, we first calibrated the 10x Genomics Xenium nuclei (stained with DAPI) image and NanoString CosMx nuclei (stained with DAPI and 18Sr/H3) image to match the scale of the Visium H&E-stained image, ensuring that pixels in both images represent the same physical size. It is worth mentioning that due to the sensitivity differences between DAPI and H&E staining, the absolute number of nuclei on Visium (48,219) is smaller than on Xenium (104,831). We then applied affine transformation to Xenium and CosMx images to align the histological features to Visium images. Taking Xenium as an example, we first scaled the Xenium image to make sure each pixel in the image represents the same physical length as the H&E-stained image from the Visium data (0.753 µm per pixel). We then aligned the Xenium image with the H&E-stained image by rotating and translating the Xenium image to ensure that the brain tissue regions (for example, CTX, HPF, TH, HY and STR) were properly aligned. We then projected the Visium spot array onto the Xenium image. In this way, we calculated the cell-type proportion of each spot using cell annotations identified by Xenium profiles as ground truth and further evaluated the accuracy of Spotiphy outputs (Supplementary Fig. 2c). For NanoString CosMx, we obtained data from a total 20 of restricted FOVs (Supplementary Fig. 2d).

**Generation of simulated ST datasets.** To create synthetic ST datasets that closely resembled the actual dataset, we used the estimated number of each cell type at every location, as determined by Spotiphy, as the ground truth. Specifically, for generating the spatial expression at a given location $s$, we randomly sampled $n_{s,t}$ cells of type $t$ cells from the scRNA-seq data for $t = 1, 2, \cdots, T$. We then merged all scRNA-seq expression data. To enhance the resemblance of the synthetic expression to

real data, we introduced batch effects, artificial zero reads and random noise (Extended Data Fig. 2a). Considering the magnitude of the disturbance, we produced three synthetic datasets characterized by small, medium and large levels of noise (Supplementary Methods).

**Estimation of cell-type proportions in mouse brain and human breast cancer samples using Spotiphy.** When select marker genes for the 27 cell types in mouse brain data and for the 10 cell types in human breast cancer data, the parameters are selected as $l_{\text{fold}} = 1.5$, $l_\lambda = 0.1$, $l_{\text{cover}} = 60\%$, $v = 0.15$ and $n_{\text{select}} = 50$. This is also the default parameter setting in Spotiphy. In total, 1,059 marker genes are selected. To approximate the posterior distribution of the unknown parameters in the probabilistic model, the parameters are updated for 8,000 iterations in Pyro. Plots of the loss function suggest that the estimation of the parameters has already converged.

**Evaluation metrics.** Here, we applied six evaluation metrics to quantify the performance of cell-type proportion estimation in each spot: absolute error, square error, JSD, cosine similarity, Pearson correlation and fraction of cells correctly mapped. For the first three metrics, a lower value indicates better performance, whereas for the remaining three metrics, a higher value signifies improved performance. We let $p_{s,t}$ denote the estimated proportion of type $t$ cells at location $s$ and let $p_{s,t}^*$ denote the ground truth. The computation methods of each metric for proportion estimation at location $s$ are briefly described as follows:

Absolute error: $\sum_{t=1}^{T} |p_{s,t} - p_{s,t}^*|$.

Square error: $\sum_{t=1}^{T} \left(p_{s,t} - p_{s,t}^*\right)^2$.

JSD: $\sqrt{\frac{1}{2}\left[D(\boldsymbol{p}_s \parallel \boldsymbol{m}) + D(\boldsymbol{p}_s^* \parallel \boldsymbol{m}_s)\right]}$, where $D$ is the Kullback–Leibler divergence, $\boldsymbol{p}_s = [p_{s,1}, p_{s,2}, \cdots, p_{s,T}]$, $\boldsymbol{p}_s^* = [p_{s,1}^*, p_{s,2}^*, \cdots, p_{s,T}^*]$, and $\boldsymbol{m}_s = (\boldsymbol{p}_s + \boldsymbol{p}_s^*)/2$.

Pearson correlation: $\frac{\sum_{t=1}^{T}(p_{s,t} - a)(p_{s,t}^* - a)}{\sqrt{\sum_{t=1}^{T}(p_{s,t} - a)^2 \sum_{t=1}^{T}(p_{s,t}^* - a)^2}}$, where $a = \frac{1}{T} = \frac{\sum_{t=1}^{T} p_{s,t}}{T} = \frac{\sum_{t=1}^{T} p_{s,t}^*}{T}$.

Cosine similarity: $\frac{\sum_{t=1}^{T} p_{s,t} p_{s,t}^*}{\sqrt{\sum_{t=1}^{T}(p_{s,t})^2 \sum_{t=1}^{T}(p_{s,t}^*)^2}}$.

Fraction of cells correctly mapped: $\sum_{t=1}^{T} \min(p_{s,t}, p_{s,t}^*)$.

**Reporting summary**
Further information on research design is available in the Nature Portfolio Reporting Summary linked to this article.

## Data availability

The scRNA-seq datasets for mouse brain samples contain two parts. The mouse whole-CTX HPF scRNA-seq dataset is available at Allen Brain Map Atlas (portal.brain-map.org/atlases-and-data/rnaseq/mouse-whole-cortex-and-hippocampus-10x). The immune cell-enriched scRNA-seq dataset generated in this study, final scRNA-seq reference and matched ST datasets for WT and AD mouse brains, including Visium, Xenium and CosMx datasets, are available at Zenodo at https://doi.org/10.5281/zenodo.10520022 (ref. 86). iscRNA datasets and simulated ST datasets used for evaluations are available at https://spotiphy.stjude.org. The astrocyte snRNA-seq dataset was downloaded from ArrayExpress under accession number E-MTAB-11115. The human breast cancer scRNA-seq dataset was downloaded from the Gene Expression Omnibus under accession number GSE195665. The three human breast cancer Visium datasets were downloaded from Zenodo at https://doi.org/10.5281/zenodo.4739739 (ref. 87). The human lung scRNA-seq dataset was downloaded from the Human Lung Cell Atlas (https://fetal-lung.cellgeni.sanger.ac.uk/scRNA.html). The human lung cancer Visium dataset was downloaded from 10x Genomics (https://www.10xgenomics.com/datasets/human-lung-cancer-ffpe-2-standard). The human lung cancer Xenium dataset was

downloaded from 10x Genomics (https://www.10xgenomics.com/datasets/ffpe-human-lung-cancer-data-with-human-immuno-oncology-profiling-panel-and-custom-add-on-1-standard). The human colorectal cancer scRNA-seq dataset was downloaded from Synapse under record number 26844071. The human colorectal cancer Visium dataset was downloaded from 10x Genomics (https://www.10xgenomics.com/datasets/human-colorectal-cancer-11-mm-capture-area-ffpe-2-standard). The human colorectal cancer Xenium dataset was downloaded from 10x Genomics (https://www.10xgenomics.com/datasets/ffpe-human-colorectal-cancer-data-with-human-immuno-oncology-profiling-panel-and-custom-add-on-1-standard). Eight additional datasets for deconvolution benchmarking were downloaded from GitHub at github.com/QuKunLab/SpatialBenchmarking. Source data are provided with this paper.

## Code availability

Spotiphy version 1.0 was coded in Python and used to generate the results in this work. The source code for Spotiphy is freely available online at GitHub at https://github.com/jyyulab/Spotiphy. The documentation and tutorial with test inputs are available online at https://spotiphy.stjude.org.

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

## Acknowledgements

We extend our gratitude to the members of the laboratory of J. Yu for their efforts in testing and improving Spotiphy. We thank Y. Zhou and L. Tang in the Department of Biostatistics at St. Jude for their statistical support. We thank N. Navin, A. Casasent and K. Kessenbrock for sharing the breast cancer spatial data. We also thank S. August for scientific editing. This work is supported, in part, by National Institutes of Health grants R01GM134382 (to J. Yu), U01CA264610 (to J. Yu), U01CA281868 (to J. Yu), R01CA274251 (to J. Yu) and RF1AG068581 (to J.P.) and by the American Lebanese Syrian Associated Charities. The content is solely the responsibility of the authors and does not necessarily represent the official views of the National Institutes of Health.

## Author contributions

J. Yang, Z.Z., J.P., K.L. and J. Yu conceived the study. J. Yang and Z.Z. designed the algorithm and performed evaluations. Z.Z. developed the software packages. J. Yang performed the biology validation analyses. K.Y., Y.J. and J.P. conducted mouse brain sample preparation and validations. S.B. and X.Y. generated the scRNA-seq data. S.N., X.Y. and J.E. generated Visium data. J. Yang generated Xenium and CosMx data, in collaboration with the services of 10x Genomics and NanoString, respectively. J.Z., Q.P. and K.-K.Y. assisted with software benchmarking tests and documentation. J. Yang, Z.Z., K.L. and J. Yu wrote the paper.

## Competing interests

The authors declare no competing interests.

## Additional information

**Extended data** is available for this paper at https://doi.org/10.1038/s41592-025-02622-5.

**Correspondence and requests for materials** should be addressed to Junmin Peng, Kaibo Liu or Jiyang Yu.

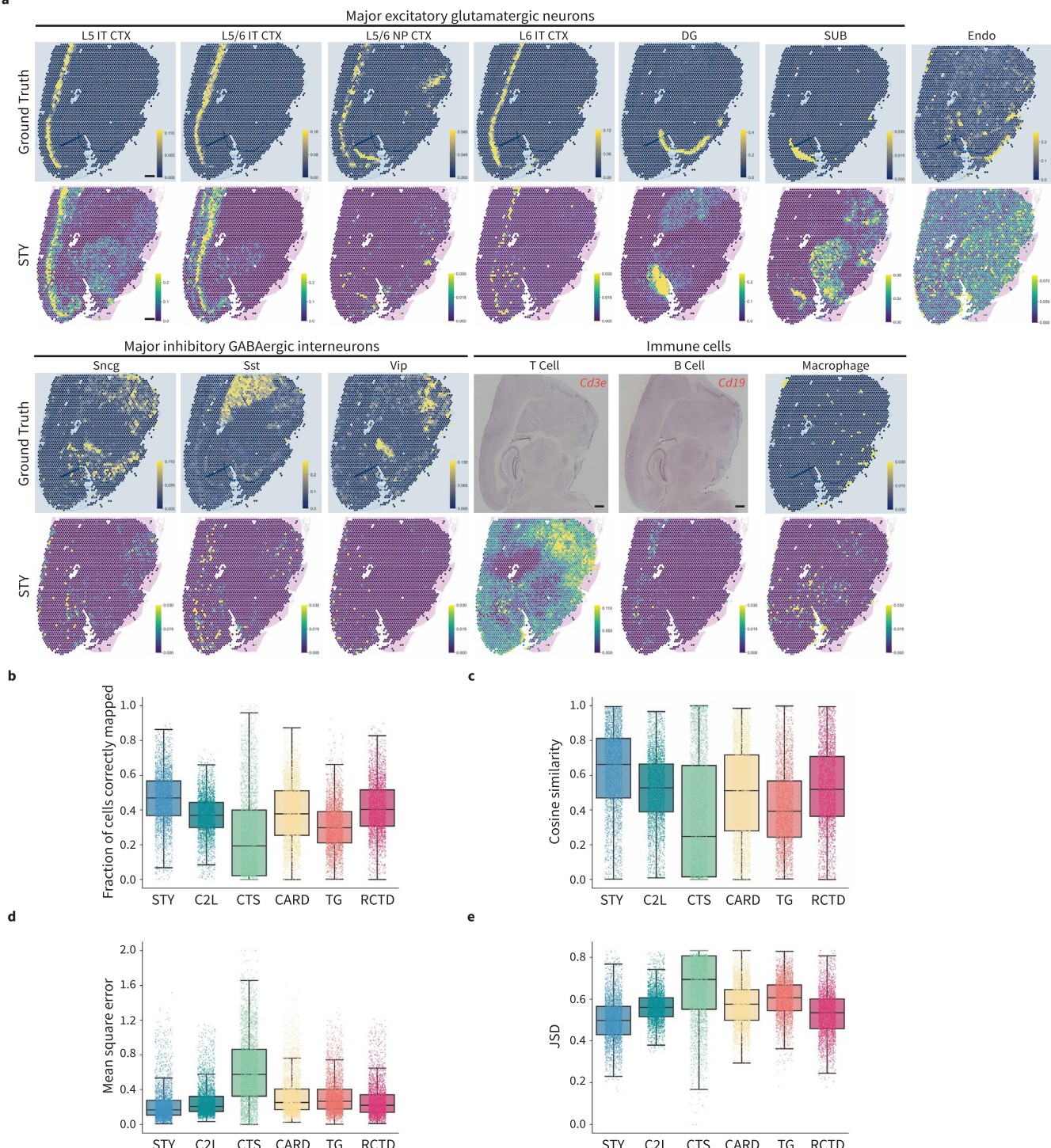

**Extended Data Fig. 1 | Spotiphy provides cell-type proportions that matches ground truth with high confidence. a**, Heatmaps depicting the proportion of the rest cell types (as determined by Spotiphy and Xenium, the ground truth) across the histological image of AD sample. Expression of *Cd3e* and *Cd19* are used as the ground truth of T and B cells in mouse brain. Allen Mouse Brain Atlas, mouse.brain-map.org/experiment/show/75988359, mouse.brain-map.org/ experiment/show/69540555. Scale bar: 500 μm. **b−e**, Box plots for the fraction of cells correctly mapped (**b**, higher is better), cosine similarity (**c**, higher is better), square error (**d**, lower is better), and JSD (**e**, Jensen−Shannon divergence, lower is better) of the cell-type proportions generated by each method for each transcriptomic spot. Boxplots are generated in the same manner as described in Fig. 2. Each platform includes 3476 spots from the AD sample.

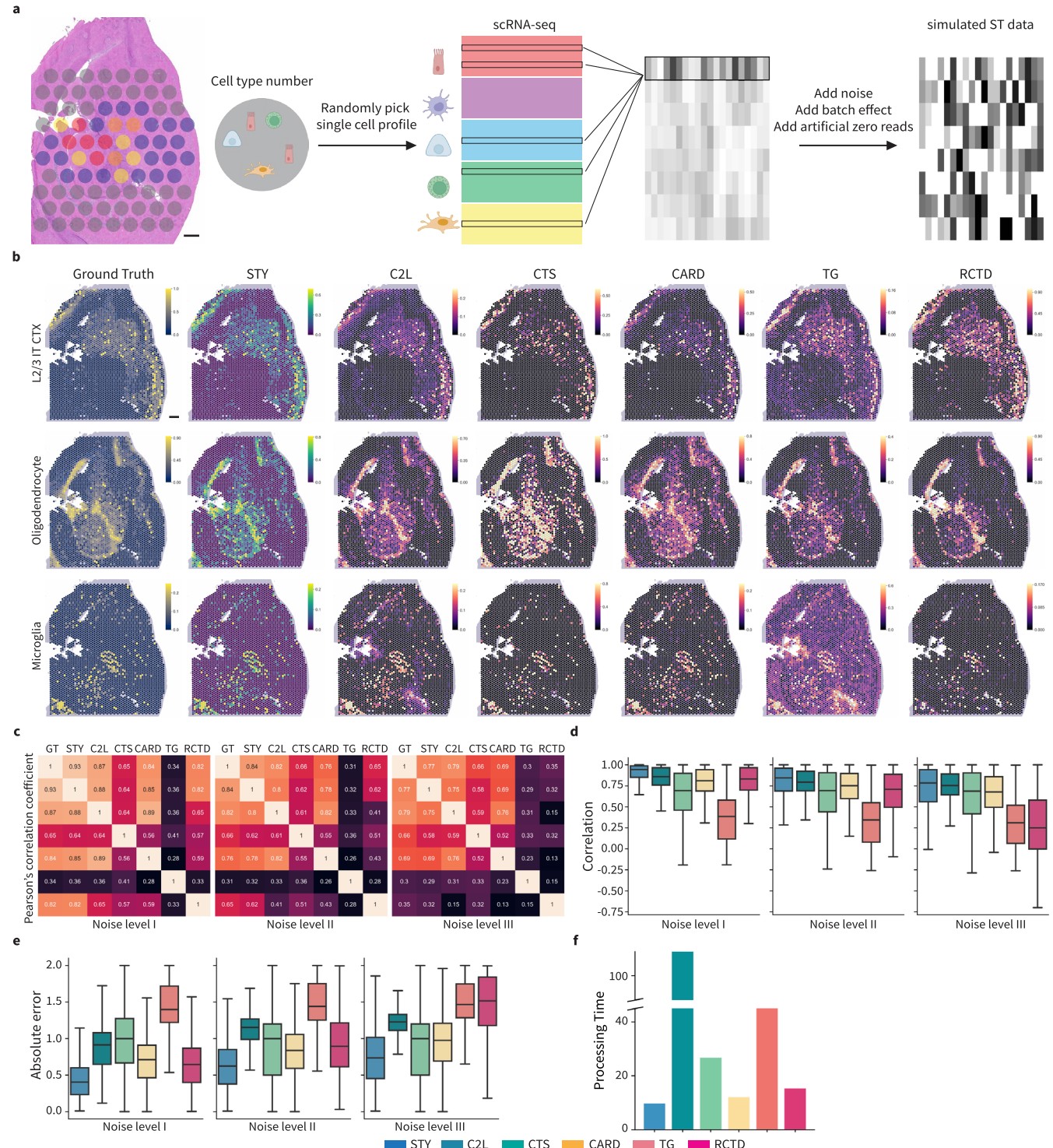

**Extended Data Fig. 2 | Simulated ST dataset further validates the superiority of Spotiphy. a**, Workflow summary for simulated ST data generation based on deconvolution result of another mouse sample. Scale bars: 500 μm. **b**, Heatmaps depicting the proportion of three selected cell types (as determined by Spotiphy and Xenium, the ground truth) across the histological image of this mouse sample. Scale bars: 500 μm. **c**, Pearson correlation coefficient heatmap of cell-type proportions generated by Xenium, Spotiphy, and other methods selected for benchmarking. **d**, **e**, Box plots for the correlation (**d**, higher is better) and absolute error (**e**, lower is better) of the cell-type proportions for each transcriptomic spot generated by each method. Boxplots are generated in the same manner as described in Fig. 2. Each platform includes 3529 spots from the brain sample. **f**, Barplot for processing time of all benchmarking methods. The processing time for each method is calculated as the average of three repeated experiments.

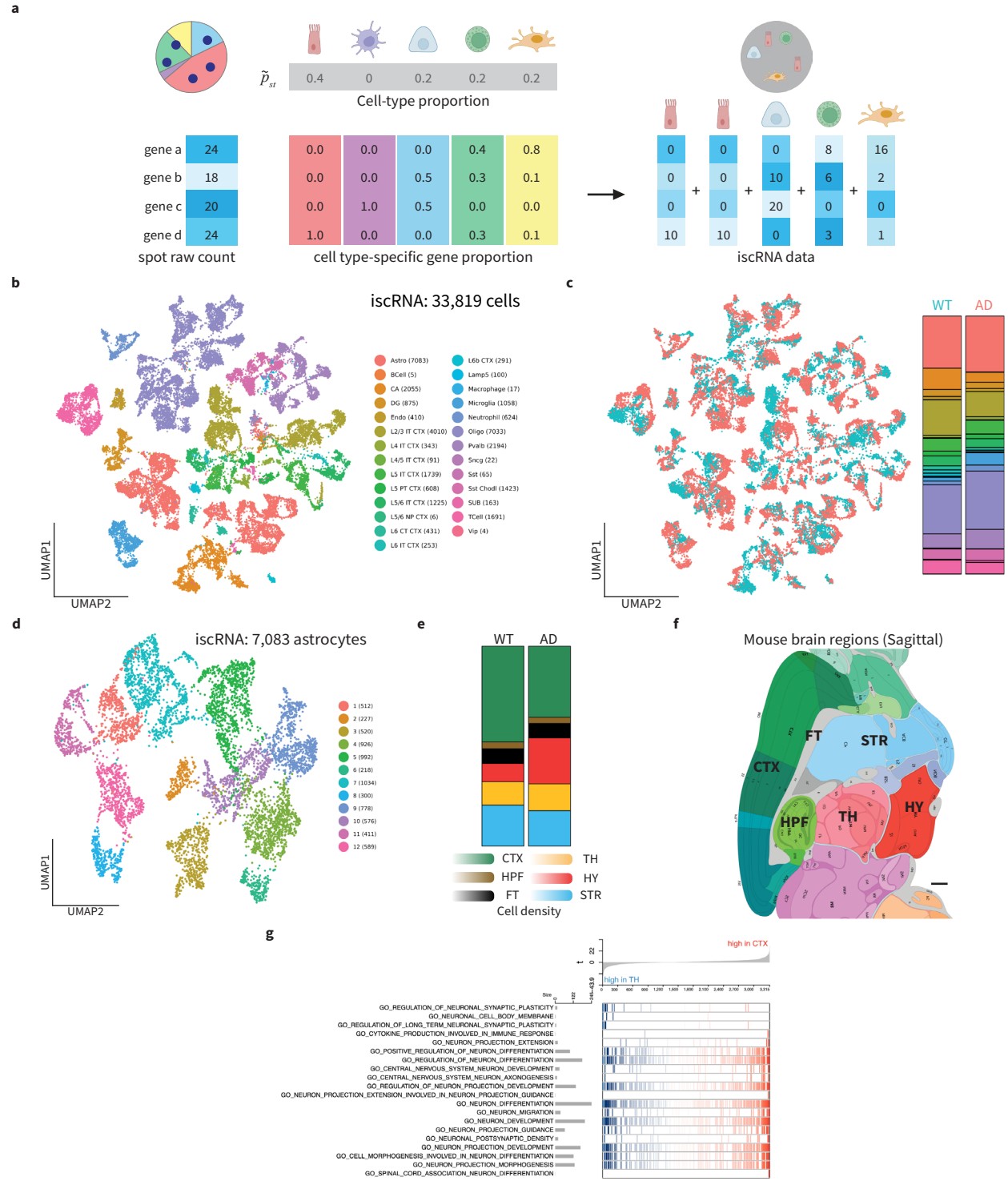

**Extended Data Fig. 3 | Decomposition outputs of Spotiphy and iscRNA data construction. a**, Workflow summary for ST data decomposition and iscRNA data construction. **b**, **c**, UMAP projection of 33,819 cells from iscRNA data produced by applying Spotiphy to mouse brain Visium data. Clusters are labeled according to cell type (**b**) or sample-of-origin (**c**). **d**, UMAP projection of 7,083 astrocytes extracted from iscRNA data produced by applying Spotiphy to mouse brain Visium data. Clusters are labeled according to the original clusters. **e**, Bar plots of astrocyte subtype proportions annotated from Fig. 3a for both WT and AD mouse samples. **f**, Diagram for sagittal section of mouse brain labeled with different topographic regions. CTX: cerebral cortex; HPF: hippocampus; FT: fiber tracts; TH: thalamus; HY: hypothalamus; STR: striatum. Allen Mouse Brain Sagittal Atlas, https://mouse.brain-map.org/static/atlas. Scale bars: 500 μm. **g**, Result of applying NetBID2 GSEA analysis to iscRNA data of astrocytes in CTX vs. TH.

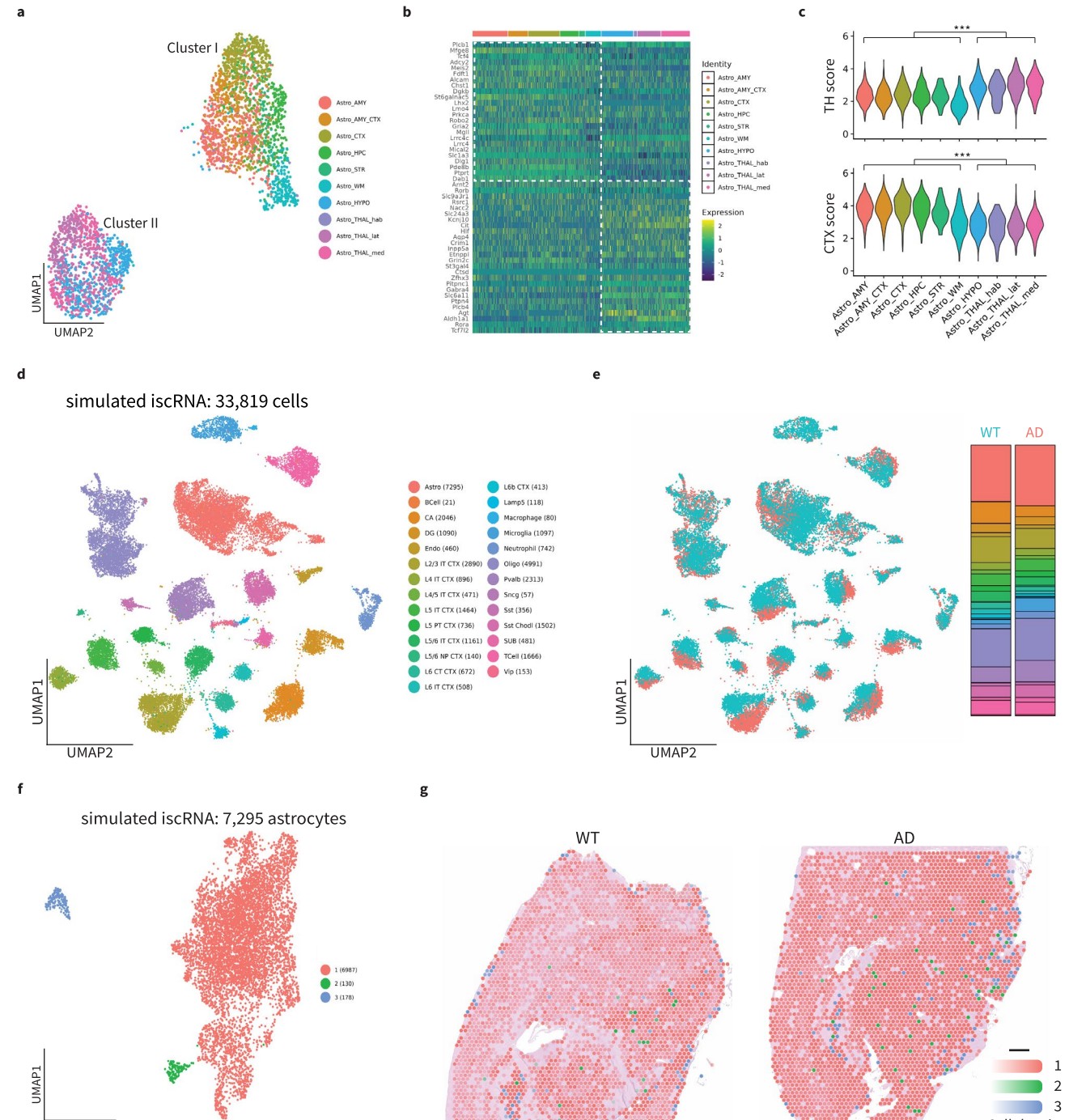

**Extended Data Fig. 4 | Prior research provides evidence validating astrocyte regional specification identified by iscRNA data generated by Spotiphy.** **a**–**c**, Published astrocyte dataset[28] confirmed astrocyte regional specification observed in iscRNA data. **a**, UMAP projection of published astrocyte dataset. **b**, Heatmap of expression of top DEGs identified from Fig. 3d among astrocytes in the published dataset. **c**, Violin plots of the CTX and TH scores for astrocytes in the published dataset. All p-values are $< 2.2 \times 10^{-16}$. **d**–**g**, astrocytes from simulated iscRNA data lose regional specification. **d**, **e**, UMAP projection of 33,819 cells from simulated iscRNA data. Clusters are labeled according to cell type (**d**) or sample-of-origin (**e**). **f**, UMAP projection of 7,295 astrocytes extracted from simulated iscRNA data labeled with default clusters. **g**, Transcriptomic spots from the Visium data. Spots are color-coded according to their default clusters, annotated in **f**. The opacity of the spots is representative of cell density. Scale bars: 500 μm. Two-sided t-tests were conducted in **c**. ***, p < 0.01.

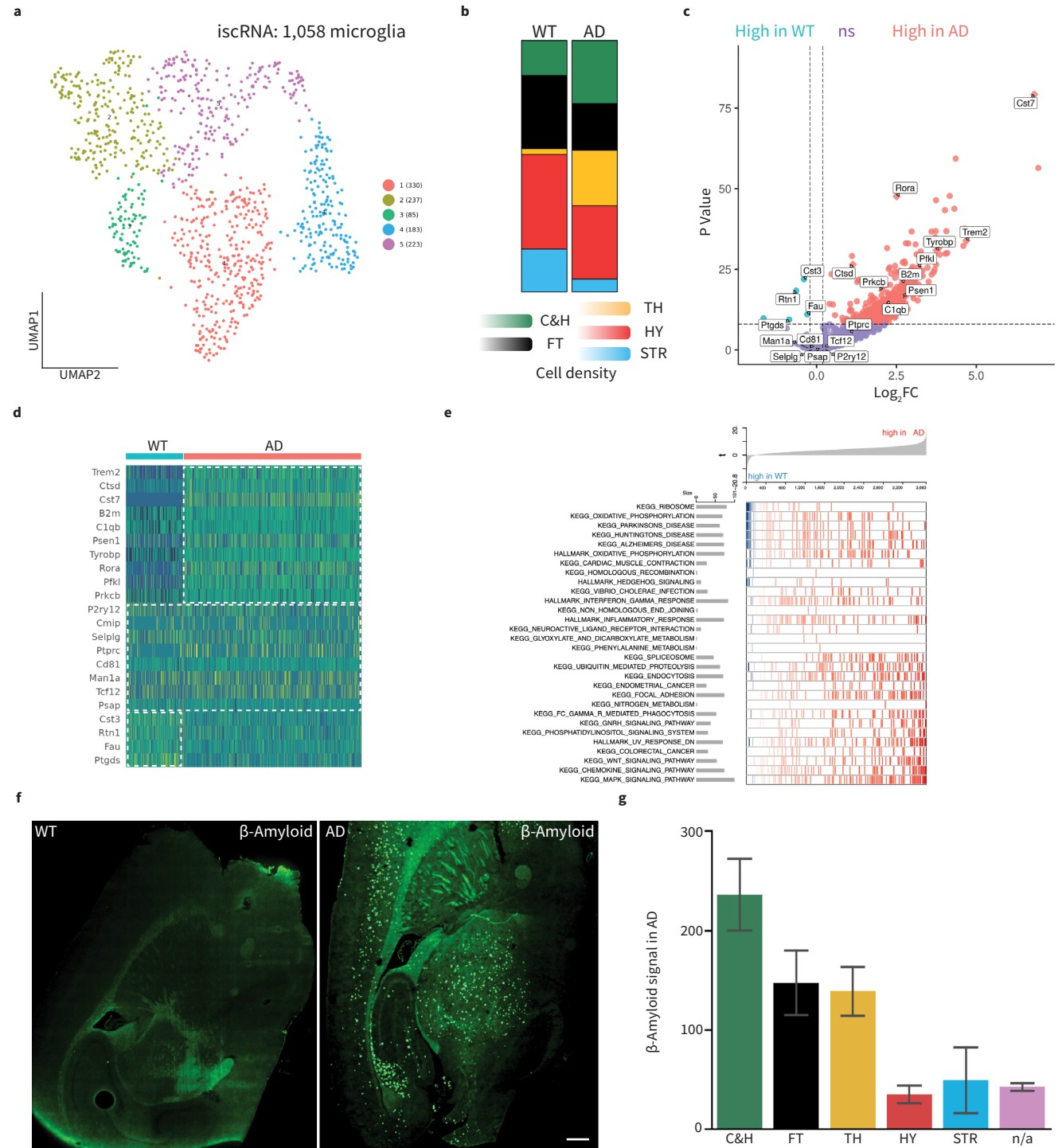

**Extended Data Fig. 5 | Unsupervised clustering analysis of iscRNA data identified DAM sub-populations. a**, UMAP projection of 1,058 microglia extracted from iscRNA data produced by applying Spotiphy to mouse brain Visium data. Clusters are labeled with default clusters. **b**, Bar plots of microglia subtype proportions annotated from Fig. 4a for both WT and AD mouse samples. **c**, Volcano plot of differentially expressed genes (DEGs) among microglia in AD vs. WT. DEGs were determined using the iscRNA data. Volcano plot is generated in the same manner as described in Fig. 3. The genes highlighted are available in the CosMx panel and have been used to generate the signature scores in Fig. 4c–e.

**d**, Heatmap of the expression of the top DEGs among microglia in WT and AD samples, respectively. **e**, Result of applying NetBID2 GSEA analysis to iscRNA data of microglia in AD vs. WT. **f**, beta-amyloid IHC staining of WT and AD samples. Scale bars: 500 μm. **g**, Quantification of beta-amyloid signals within different topographic regions of AD sample. Each barplot represents the average of fluorescence signal within each spot after overlaying the IHC image with Visium spot grid of the AD sample. The error bar represents the standard deviation. AD sample contains the following number of spots for each region: C&H (192), FT (144), TH (160), HY (233), STR (40), and n/a (2707).

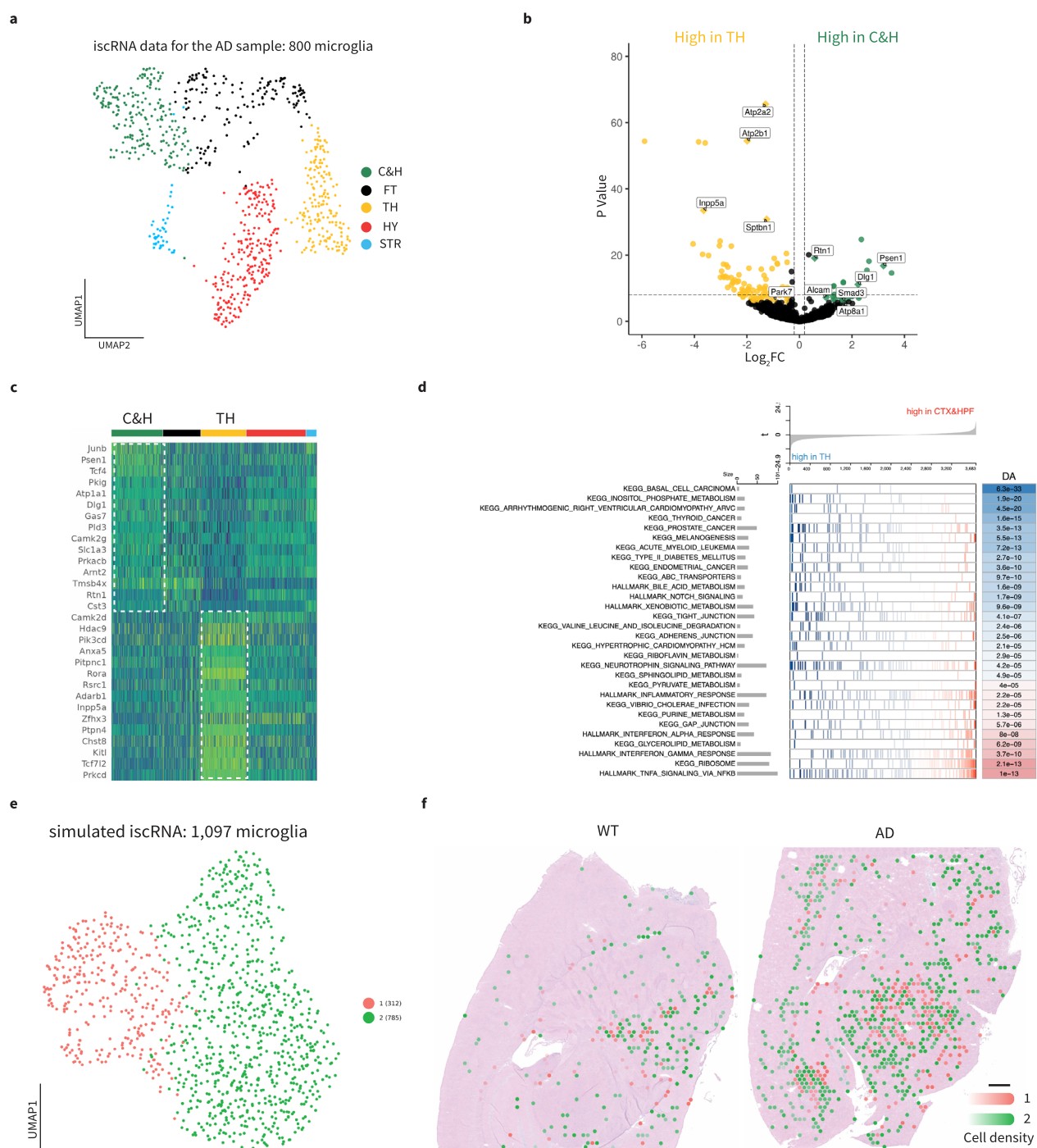

**Extended Data Fig. 6 | Spotiphy captures DAM regional specification in AD mouse brain tissue. a**, UMAP projection of 800 microglia extracted from iscRNA data generated by Spotiphy to AD mouse brain Visium data. Clusters are labeled according to their corresponding topographic regions. **b**, Volcano plot of differentially expressed genes (DEGs) among microglia in C&H vs. TH. Volcano plot is generated in the same manner as described in Fig. 3. DEGs were determined using the iscRNA data. The genes highlighted are available in the CosMx panel and have been used to generate the signature scores in Fig. 4f–h. **c**, Heatmap of expression of top DEGs among microglia in AD sample. **d**, Result of applying NetBID2 GSEA analysis to iscRNA data of AD microglia in C&H vs. TH. **e**, UMAP projection of 1,097 microglia extracted from simulated iscRNA data labeled with default clusters. **f**, Transcriptomic spots from the Visium data. Spots are color-coded according to their default clusters, annotated in **e**. The opacity of the spots is representative of cell density. Scale bars: 500 μm.

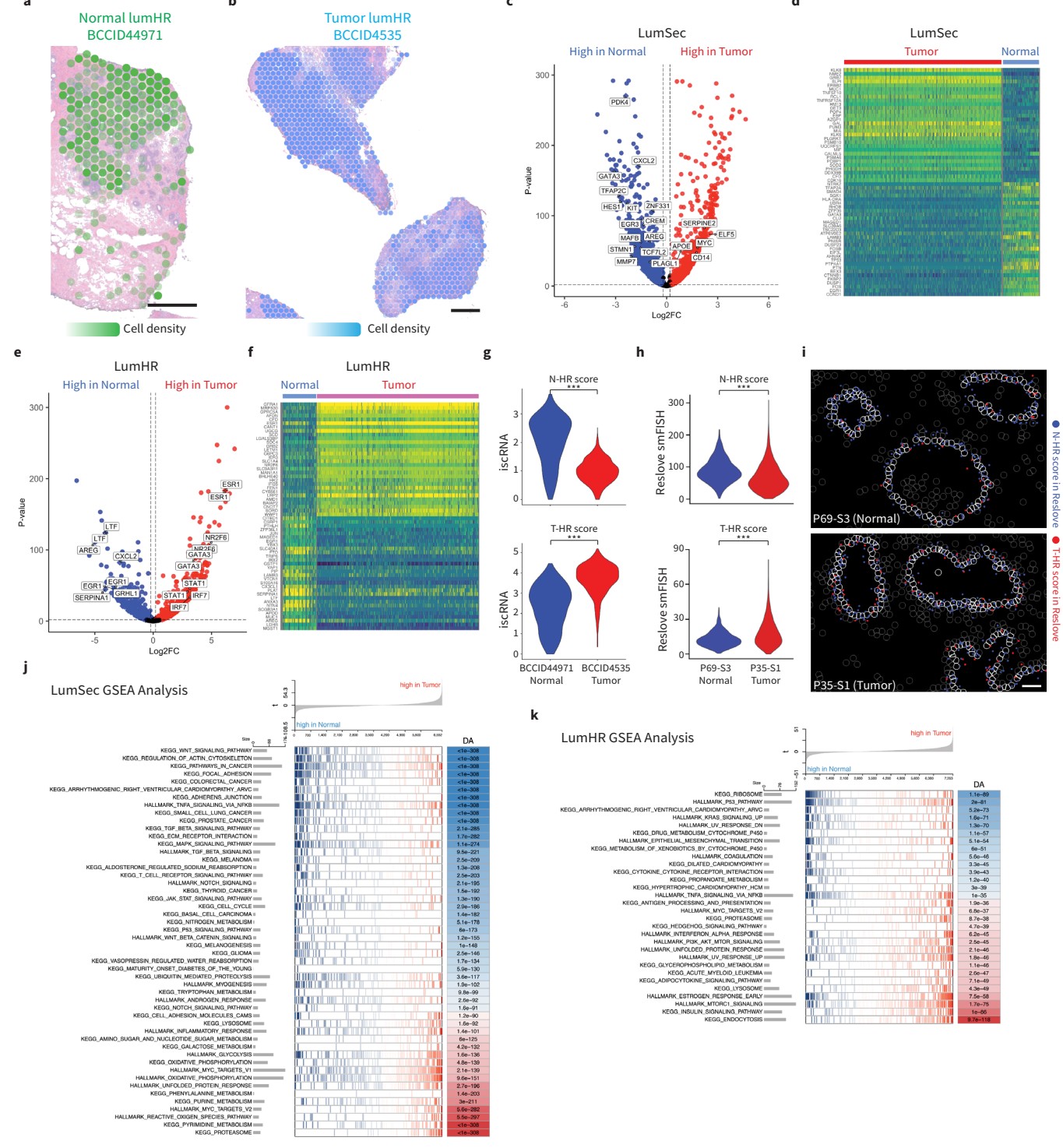

**Extended Data Fig. 7 | See next page for caption.**

**Extended Data Fig. 7 | Spotiphy reveals tumor-microenvironment changes across spatial domains. a**, **b**, Transcriptomic spots from the Visium data. Spots are color-coded according to their constituent LumHR cells' sample-of-origin: BCCID44971 (**a**) and BCCID4535 (**b**) annotated in Fig. 5b. The opacity of the spots is representative of cell density. Scale bars: 500 μm. **c**, Volcano plot of differentially expressed genes (DEGs) among LumSec cells in BC1160920F vs. BCCID44971. Volcano plot is generated in the same manner as described in Fig. 3. DEGs were determined using the iscRNA data. The genes highlighted are available in the Resolve panel and have been used to generate the N-Sec and T-Sec scores in Fig. 5f–h. **d**, Heatmap of expression of top DEGs among LumSec cells. **e**, Volcano plot of DEGs among LumHR cells in BCCID4535 vs. BCCID44971. DEGs were determined using the iscRNA data. The genes highlighted are available in the Resolve panel and have been used to generate the N-HR and T-HR scores in **g**–**i**. **f**, Heatmap of expression of top DEGs among LumHR cells. **g**, Violin plots of

the N-HR and T-HR scores for the LumHR cells identified in the iscRNA data of normal sample BCCID44971 and tumor sample BCCID4535. All p values are $<2.2 \times 10^{-16}$. **h**, Violin plots of the N-HR and T-HR scores for the LumSec cells identified in the Resolve smFISH data of normal sample P69-S3 and tumor sample P35-S1. p value at upper: $1.9 \times 10^{-52}$, lower: $2.2 \times 10^{-42}$. Two-sided t-tests are conducted. ***, p < 0.01. **i**, Visualization of signature genes in selected regions of the Resolve smFISH data. Blue and red represent the signature genes used to calculate the N-HR and T-HR scores, respectively. Cell boundaries are depicted with gray lines; LumHR cells are outlined in white. Resolve smFISH includes three biological replicates each for P69 and P35 samples. Scale bars: 100 μm. **j**, Result of applying NetBID2 GSEA analysis to iscRNA data of LumSec cells in BC1160920F vs. BCCID44971. **k**, Result of applying NetBID2 GSEA analysis to iscRNA data of LumHR cells in BCCID4535 vs. BCCID44971.

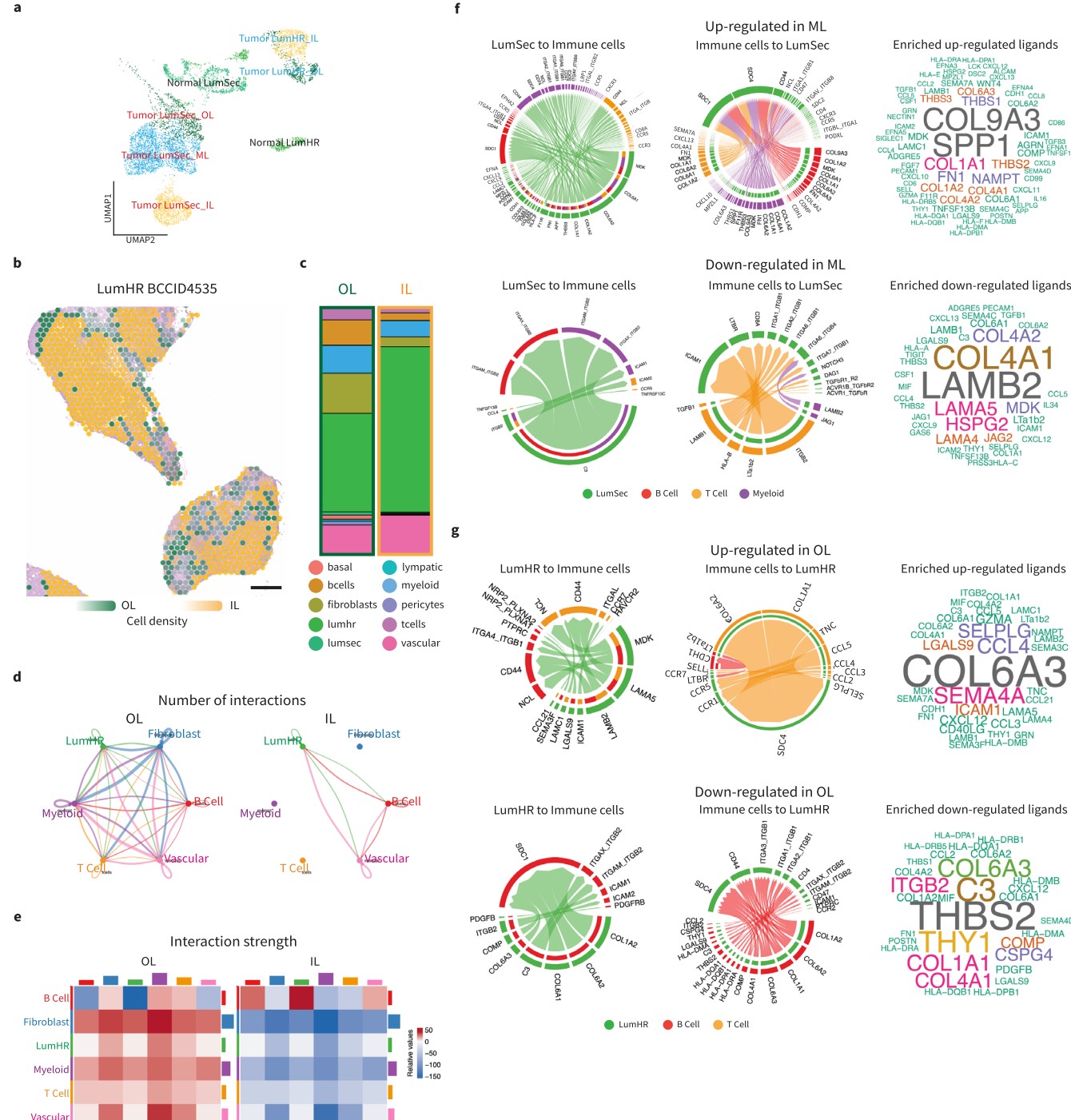

**Extended Data Fig. 8 | Downstream analysis of cell-cell communication patterns in human breast samples. a**, UMAP projection of LumSec and LumHR cells from human breast tissue iscRNA data produced by applying Spotiphy to Visium data. Cells are color-coded according to their constituent LumSec cells' and LumHR cells' spatial domains. **b**, Transcriptomic spots from the Visium data. Spots are color-coded according to their constituent LumHR cells' spatial domains. The opacity of the spots is representative of cell density. Scale bars: 500 μm.

**c**, Cell-type proportions of the two spatial domains based on the iscRNA data shown in Fig. 5a. **d**, **e**, Results of applying CellChat to the iscRNA data of the cells in the two LumHR spatial domains. **d**, Number of interactions among cell types across two spatial domains. **e**, Interaction strength among cell types across two spatial domains. OL, outer layer; IL, inner layer. **f**, **g**, Detailed visualization of the up-regulated and down-regulated signaling ligand-receptor pairs using Chord diagrams of LumSec cells in ML domain (**f**) and LumHR cells in OL domain (**g**).

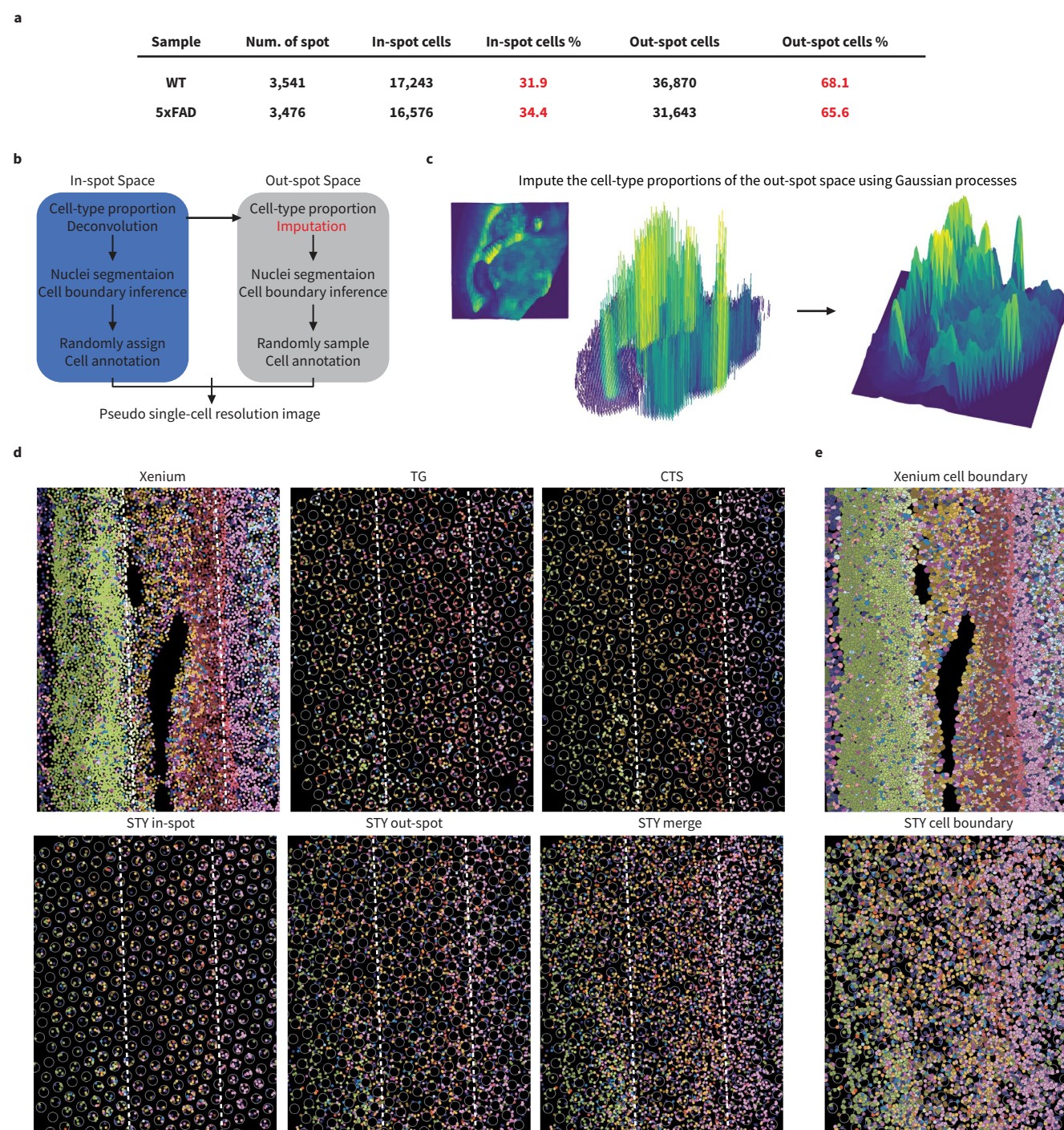

**a**

| Sample | Num. of spot | In-spot cells | In-spot cells % | Out-spot cells | Out-spot cells % |
|--------|-------------|---------------|-----------------|----------------|------------------|
| WT | 3,541 | 17,243 | 31.9 | 36,870 | 68.1 |
| 5xFAD | 3,476 | 16,576 | 34.4 | 31,643 | 65.6 |

**Extended Data Fig. 9 | Spotiphy's imputation to out-spot increases information density of ST data. a**, Quantification of all detectable nuclei/cells including both in-spot and non-capture areas from WT and AD samples. **b**, The workflow of pseudo single-cell resolution image generation. **c**, Schematic overview of Gaussian processes for cell-type proportion imputation. **d**, **e**, Cell type annotation of individual cells in selected FOVs of cerebral cortex. Dots in **d** represent all detectable nuceli, color-coded for cell type. Grey circles represent the transcriptomic spots in the Visium data. Dashed white lines have been added to allow for easier comparison. **e**, Depicts whole cells, color-coded for cell type. The cell boundaries depicted for the Xenium (upper panel) and Spotiphy (lower panel) data were inferred through those methods' respective pipelines. Color legend in Fig. 6. Scale bars in **d**, **e**: 100 μm.

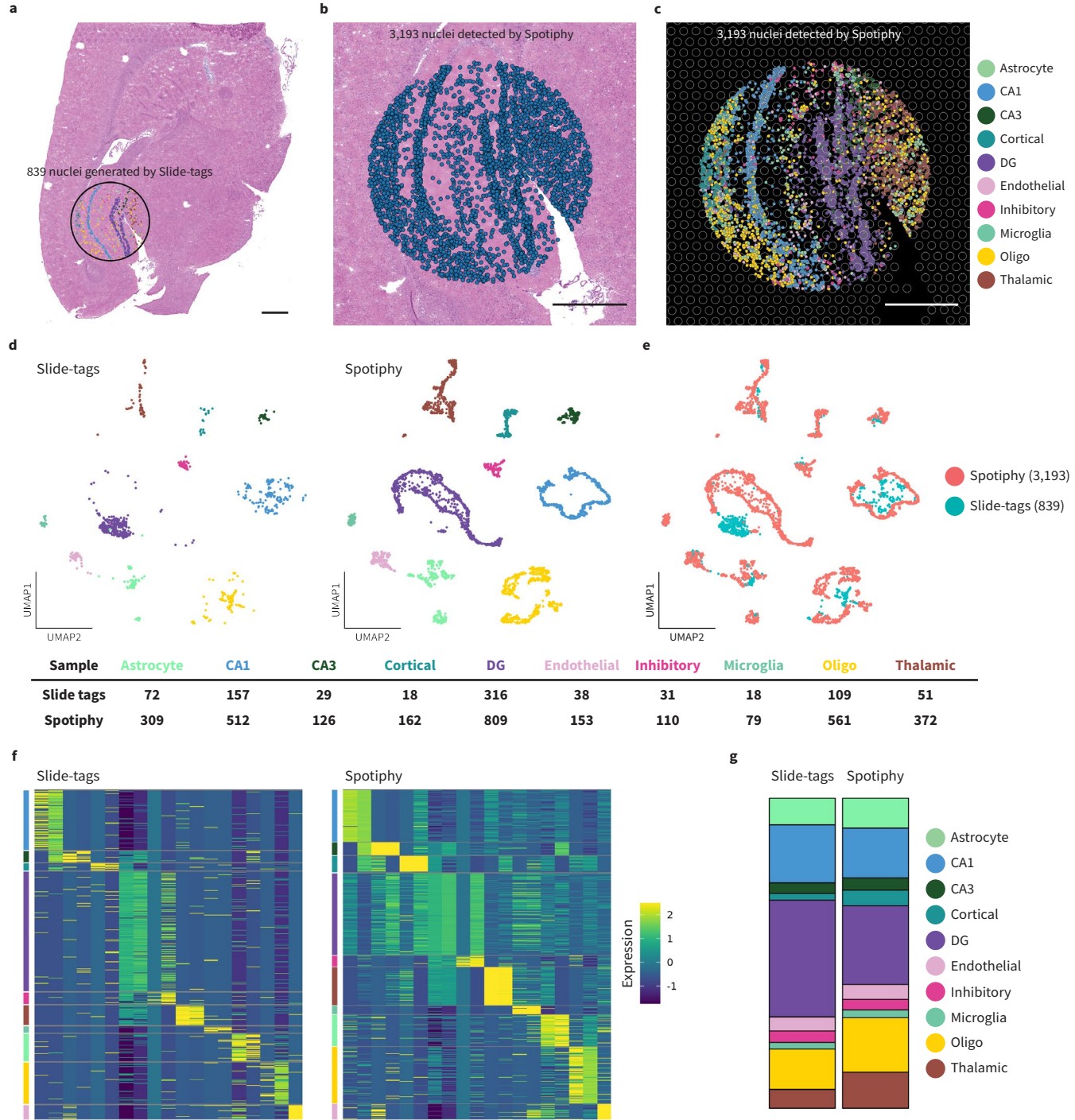

**Extended Data Fig. 10 | Spotiphy's imputation to loss nuclei increases information density of Slide-tags data. a**, Alignment of AD sample Visium H&E image and hippocampus plot from Slide-tags data of. **b**, **c**, Closer look at same hippocampus region of AD sample with 3,193 nuclei detected by Spotiphy segmentation. **b**, nuclei are color-coded with blue to indicate their location on the H&E image. **c**, nuclei are color-coded for cell type annotated by Spotiphy.

Grey circles represent transcriptomic spots from Visium data. Scale bars in **a**–**c**: 500 μm. **d**, **e**, UMAP projection of 4,032 cells including Slide-tags data and iscRNA data. Clusters are labeled according to their cell type (**d**) or data-of-origin (**e**). **f**, Heatmaps displaying the expression patterns of the marker gene sets from the original article in Slide-tags data and iscRNA data. **g**, Cell-type proportions of the Slide-tags data and iscRNA data.

# Reporting Summary

## Statistics

For all statistical analyses, confirm that the following items are present in the figure legend, table legend, main text, or Methods section.

| n/a | Confirmed | |
|---|---|---|
| ☐ | ☒ | The exact sample size (*n*) for each experimental group/condition, given as a discrete number and unit of measurement |
| ☐ | ☒ | A statement on whether measurements were taken from distinct samples or whether the same sample was measured repeatedly |
| ☐ | ☒ | The statistical test(s) used AND whether they are one- or two-sided<br>*Only common tests should be described solely by name; describe more complex techniques in the Methods section.* |
| ☒ | ☐ | A description of all covariates tested |
| ☐ | ☒ | A description of any assumptions or corrections, such as tests of normality and adjustment for multiple comparisons |
| ☐ | ☒ | A full description of the statistical parameters including central tendency (e.g. means) or other basic estimates (e.g. regression coefficient) AND variation (e.g. standard deviation) or associated estimates of uncertainty (e.g. confidence intervals) |
| ☐ | ☒ | For null hypothesis testing, the test statistic (e.g. *F*, *t*, *r*) with confidence intervals, effect sizes, degrees of freedom and *P* value noted<br>*Give P values as exact values whenever suitable.* |
| ☐ | ☒ | For Bayesian analysis, information on the choice of priors and Markov chain Monte Carlo settings |
| ☒ | ☐ | For hierarchical and complex designs, identification of the appropriate level for tests and full reporting of outcomes |
| ☐ | ☒ | Estimates of effect sizes (e.g. Cohen's *d*, Pearson's *r*), indicating how they were calculated |

*Our web collection on statistics for biologists contains articles on many of the points above.*

## Software and code

Policy information about availability of computer code

| Data collection | Single-cell Sequencing alignments were done by Cell Rannger-arc v2.0.0.<br>Visium Sequencing alignments were done by Space Ranger v1.3.0.<br>CosMx data were collected through Services.<br>Xenium data were collected through Services, and analyzed by Xenum Explorer v1.3.0. |
|---|---|

| Data analysis | The Spotiphy package is available at https://github.com/jyyulab/Spotiphy. Usage of the code is introduced at https://jyyulab.github.io/Spotiphy/. Jupyter notebooks covering the analysis in this paper are available upon request. |
|---|---|

Specific package versions used in Python:
Python 3.9.18; numpy 1.22.4; scipy 1.9.1; scikit-learn 1.2.2; Scanpy 1.9.3; anndata 0.10.3; opencv-python 4.8.1.78; torch 2.1.1; pyro-ppl 1.8.4; tensorflow 2.12.0; stardist 0.8.3.

Specific package versions used in R 4.1.3:
Seurat 4.3.0; Harmony 0.1.1; CellChat 1.5.0; InferCNV 1.3.3; NetBID 2.0; scMINNER 1.0.0. Dependencies have not been listed for brevity.

Package versions of benchmark methods:
Cell2location 0.1.3; Tangram 1.0.4; CARD 1.0; RCTD 2.2.1; cytoSPACE 1.0.6; iStar (no version number); Stereoscope 0.2.0; SPOTlight 1.10.0; MuSic, 1.0.0; SpatialScope 0.0.1; SpatialDWLS 4.0.0; Redeconve 1.1.1; CIBERSORTx https://cibersortx.stanford.edu/.

For manuscripts utilizing custom algorithms or software that are central to the research but not yet described in published literature, software must be made available to editors and reviewers. We strongly encourage code deposition in a community repository (e.g. GitHub). See the Nature Portfolio guidelines for submitting code & software for further information.

# Data

Policy information about availability of data

All manuscripts must include a data availability statement. This statement should provide the following information, where applicable:

- Accession codes, unique identifiers, or web links for publicly available datasets
- A description of any restrictions on data availability
- For clinical datasets or third party data, please ensure that the statement adheres to our policy

The scRNA-seq datasets for mouse brains contain two parts: the mouse whole cortex and hippocampus scRNA-seq dataset is available at Allen Brain Map Atlas (portal.brain-map.org/atlases-and-data/rnaseq/mouse-whole-cortex-and-hippocampus-10x). The immune cell-enriched scRNA-seq dataset generated in this study is available at the Zenodo data repository under record number 10520022. The final scRNA-seq reference used in this study can also be found at the Zenodo data repository under record number 10520022. Additionally, the matched spatial transcriptomics (ST) datasets for WT and AD mouse brains, including Visium, Xenium, and CosMx, are available at the Zenodo data repository under record number 10520022. iscRNA datasets and simulated ST datasets used for evaluations are available at https://spotiphy.stjude.org.

The snRNA-seq dataset for astrocyte was downloaded from the ArrayExpress under accession number E-MTAB-11115. The scRNA-seq dataset for human breast cancer is downloaded from the Gene Expression Omnibus under accession number GSE195665. The three Visium datasets for human breast cancer are downloaded from the Zenodo data repository under record number 4739739. The scRNA-seq dataset for human lung is downloaded from Human lung cell Atlas (https://fetal-lung.cellgeni.sanger.ac.uk/scRNA.html). The Visium dataset for human lung cancer is downloaded from 10x Genomics (https://www.10xgenomics.com/datasets/human-lung-cancer-ffpe-2-standard). The Xenium dataset for human lung cancer is downloaded from 10x Genomics (https://www.10xgenomics.com/datasets/ffpe-human-lung-cancer-data-with-human-immuno-oncology-profiling-panel-and-custom-add-on-1-standard). The scRNA-seq dataset for human colorectal cancer is downloaded from Synapse under record number 26844071. The Visium dataset for human colorectal cancer is downloaded from 10x Genomics (https://www.10xgenomics.com/datasets/human-colorectal-cancer-11-mm-capture-area-ffpe-2-standard). The Xenium dataset for human colorectal cancer is downloaded from 10x Genomics (https://www.10xgenomics.com/datasets/ffpe-human-colorectal-cancer-data-with-human-immuno-oncology-profiling-panel-and-custom-add-on-1-standard). Eight additional datasets for deconvolution benchmarking are downloaded from github.com/QuKunLab/SpatialBenchmarking.

# Research involving human participants, their data, or biological material

Policy information about studies with human participants or human data. See also policy information about sex, gender (identity/presentation), and sexual orientation and race, ethnicity and racism.

| Reporting on sex and gender | N/A |
|---|---|
| Reporting on race, ethnicity, or other socially relevant groupings | N/A |
| Population characteristics | N/A |
| Recruitment | N/A |
| Ethics oversight | N/A |

Note that full information on the approval of the study protocol must also be provided in the manuscript.

# Field-specific reporting

Please select the one below that is the best fit for your research. If you are not sure, read the appropriate sections before making your selection.

☒ Life sciences      ☐ Behavioural & social sciences      ☐ Ecological, evolutionary & environmental sciences

For a reference copy of the document with all sections, see nature.com/documents/nr-reporting-summary-flat.pdf

# Life sciences study design

All studies must disclose on these points even when the disclosure is negative.

| | |
|---|---|
| Sample size | 18 mouse samples, comprising 8 WT and 8 FAD models, were used for single-cell data profiling. One pair of mouse samples (WT_1 and FAD_1) was utilized across multiple spatial transcriptomics (ST) platforms. Sample size calculations were not performed. The sample size for scRNA-seq data was determined based on extensive prior experience with these technologies in mouse brains. Clustering analysis demonstrates its robustness to downsampling, indicating that the sample size is sufficient. Regarding ST data, each tissue section is capable of producing only one type of ST data. Because we are utilizing adjacent sections to generate matched datasets, we obtain only one data for every ST platform from each sample. Later clustering analysis also shows the sample size is sufficient. |
| Data exclusions | No data was excluded. |
| Replication | 8 Replicates were gathered from both WT and FAD genotypes for single-nucleus data profiling. All samples were sequenced successfully and included in the final dataset. The final dataset had no batch effects. |
| Randomization | Randomization is not relevant to our study design as this is not a trial. There were no WT vs. FAD group comparison using scRNA-seq data. |
| Blinding | Prior to clustering, the researchers were blind to the mouse sample labels and genotype information. Unsupervised clustering was performed blind to the cell source or any other metadata that could reveal sample identity. Cell annotation was based on previously known marker genes. |

# Reporting for specific materials, systems and methods

We require information from authors about some types of materials, experimental systems and methods used in many studies. Here, indicate whether each material, system or method listed is relevant to your study. If you are not sure if a list item applies to your research, read the appropriate section before selecting a response.

## Materials & experimental systems

| n/a | Involved in the study |
|---|---|
| ☐ | ☒ Antibodies |
| ☒ | ☐ Eukaryotic cell lines |
| ☒ | ☐ Palaeontology and archaeology |
| ☐ | ☒ Animals and other organisms |
| ☒ | ☐ Clinical data |
| ☒ | ☐ Dual use research of concern |
| ☒ | ☐ Plants |

## Methods

| n/a | Involved in the study |
|---|---|
| ☒ | ☐ ChIP-seq |
| ☐ | ☒ Flow cytometry |
| ☒ | ☐ MRI-based neuroimaging |

## Antibodies

| | |
|---|---|
| Antibodies used | CD45 (ab10558, https://www.abcam.com/products/primary-antibodies/cd45-antibody-ab10558.html), CD11b (ab133357, https://www.abcam.com/products/primary-antibodies/cd11b-antibody-epr1344-ab133357.html) |
| Validation | The flow cytometry antibodies used in this study are widely used and have been validated by manufacturers. |

## Animals and other research organisms

Policy information about studies involving animals; ARRIVE guidelines recommended for reporting animal research, and Sex and Gender in Research

| | |
|---|---|
| Laboratory animals | 4-month-old 5XFAD, C57BL/6J, (all Jackson Laboratory, Bar Harbor, ME) mice were used in this study. |
| Wild animals | This study did not involve wild animals. |
| Reporting on sex | Sex was not considered in this study. |
| Field-collected samples | No field-collected samples were used. |
| Ethics oversight | All the experimental procedures in the animals were performed in accordance with the NIH Guide for the Care and Use of Laboratory Animals and all protocols, were approved by the St Jude Children's Research Hospital (protocol 542) IACUCS. Experiments were carried out in accordance with The Code of Ethics of the World Medical Association (Declaration of Helsinki) for animal experiments. |

Note that full information on the approval of the study protocol must also be provided in the manuscript.

# Plants

| | |
|---|---|
| Seed stocks | *Report on the source of all seed stocks or other plant material used. If applicable, state the seed stock centre and catalogue number. If plant specimens were collected from the field, describe the collection location, date and sampling procedures.* |
| Novel plant genotypes | *Describe the methods by which all novel plant genotypes were produced. This includes those generated by transgenic approaches, gene editing, chemical/radiation-based mutagenesis and hybridization. For transgenic lines, describe the transformation method, the number of independent lines analyzed and the generation upon which experiments were performed. For gene-edited lines, describe the editor used, the endogenous sequence targeted for editing, the targeting guide RNA sequence (if applicable) and how the editor was applied.* |
| Authentication | *Describe any authentication procedures for each seed stock used or novel genotype generated. Describe any experiments used to assess the effect of a mutation and, where applicable, how potential secondary effects (e.g. second site T-DNA insertions, mosiacism, off-target gene editing) were examined.* |

# Flow Cytometry

## Plots

Confirm that:

☐ The axis labels state the marker and fluorochrome used (e.g. CD4-FITC).

☐ The axis scales are clearly visible. Include numbers along axes only for bottom left plot of group (a 'group' is an analysis of identical markers).

☐ All plots are contour plots with outliers or pseudocolor plots.

☐ A numerical value for number of cells or percentage (with statistics) is provided.

## Methodology

| | |
|---|---|
| Sample preparation | Mice were deeply anesthetized with an Avertin and perfused with ice-cold 1xPBS (pH 7.4) to flush circulating blood cells from the brain. The exsanguinated brain was quickly removed, left hemisphere was collected by dissecting and placed in ice-cold HBSS (Gibco, Life Technologies NY, USA).The dissecting left hemisphere was used for profiling using 10x Genomics Chromium Single Cell Multiome ATAC + Gene Expression kit. A small piece of the tissue was flash frozen and used for direct nuclei isolation by following the 10X Genomics Demonstrated protocol Nuclei Isolation from Complex Tissues for Single Cell Multiome ATAC + Gene Expression Sequencing; CG000375 Rev A. Nuclei were counted by staining Trypan Blue on a hemocytometer and adjusted to a concentration of 1000 nuclei/uL. The remaining larger piece of the tissue sample was dissociated to yield single cells using papain-based dissociation solution on GentleMACS Octodissociator at 37 oC for 20 min. After filtering the dissociated sample, the neuronal and immune cells were separated from the unwanted myelin using density gradient centrifugation with 22% Percoll. The pellet containing microglia, lymphocytes, non-immune neuronal cells like astrocytes, oligodendrocytes, neuronal stem cells was washed in HBSS buffer and stained with CD45 (1:100) and CD11b (1:100) to enrich immune cell populations using FACS Aria III Cell Sorter. After isolating CD45+ & CD11b+ Microglia, CD45+ other Immune cells, and double negative neuronal cells were counted using AO/PI staining on Luna fl Cell Counter. |
| Instrument | BD FACSAria™ III Cell Sorter |
| Software | BD FACSDiva™ Software |
| Cell population abundance | 15,000 Neuronal cells, 10,000 Microglia and all the cells of the immune population were mixed and used for nuclei isolation using 10X Genomics Demonstrated Protocol Nuclei Isolation for Single Cell Multiome ATAC + Gene Expression Sequencing; CG000365 Rev A. Nuclei were counted by staining Trypan Blue on a hemocytometer and adjusted to a concentration of 2000 nuclei/uL. Nuclei isolated from both pieces were mixed such that 5000 nuclei were targeted from nuclei isolated from enriched cell populations and 1000 nuclei were targeted from nuclei obtained directly from frozen section. |
| Gating strategy | To get the enriched immune cell populations using flow cytometry, start by creating a forward scatter (FSC) vs side scatter (SSC) plot to exclude debris and dead cells. Next, plot CD45 against SSC to identify and gate on CD45+ cells. From this CD45+ population, plot a CD11b vs SSC and gate on CD11b+ cells. |

☐ Tick this box to confirm that a figure exemplifying the gating strategy is provided in the Supplementary Information.

