## [Peer Review File · Nature Methods]

Spotiphy enables single-cell spatial whole transcriptomics across the entire section

Corresponding Author: Dr Jiyang Yu

A version of this paper was originally rejected for publication by Nature Methods, however that decision was reconsidered after appeal by the authors.

Version 0:

Decision Letter:

12th Mar 2024

Dear Dr Yu,

Thank you for your inquiry about submitting your manuscript, "Spotiphy enables single-cell spatial whole transcriptomics via generative modeling" to Nature Methods. The paper sounds like it may be of interest, and should fit the scope of the journal. We would be willing to consider it further for publication in Nature Methods.

Of course, it is very difficult to judge a paper based only on the limited information available in a presubmission inquiry. Therefore I am sure you understand that we cannot promise to send your paper out for peer review and must read it in its entirety before deciding if this would be suitable.

Please keep in mind that the journal is aimed at a large, interdisciplinary audience and places a strong emphasis on the practical value of the work presented for basic research in the life sciences. We strongly encourage you to include data to validate method performance and demonstrate its general applicability.

Manuscripts describing new algorithms and software should describe the principles underlying the algorithms in an accessible style targeted for our broad biological audience. The manuscript should demonstrate the practical relevance of the tool and any advantages and limitations it presents compared to available software. The code, and ideally a user manual and example data for testing the code, must be made available to reviewers as part of the peer review process.

You will find our Guide to Authors at <http://www.nature.com/naturemethods> to assist you in preparing your manuscript. However, it is not necessary at this stage to spend major effort adhering to our detailed formatting instructions.

Thank you for your interest in Nature Methods.

Sincerely,
Rita

Rita Strack, Ph.D.
Senior Editor
Nature Methods

Version 1:

Decision Letter:

13th May 2024

Dear Dr Yu,

Your Article entitled "Spotiphy enables single-cell spatial whole transcriptomics via generative modeling" has now been seen by three reviewers, whose comments are attached. While they find your work of potential interest, they have raised serious concerns which in our view are sufficiently important that they preclude publication of the work in Nature Methods, at least in its present form.

As you will see, the reviewers raise concerns about performance, validation, benchmarking, and demonstrations.

Should further experimental data allow you to fully address these criticisms we would be willing to look at a revised manuscript (unless, of course, something similar has by then been accepted at Nature Methods or appeared elsewhere). This includes submission or publication of a portion of this work somewhere else. We hope you understand that until we have read the revised paper in its entirety we cannot promise that it will be sent back for peer-review.

Although we cannot publish your paper, it may be appropriate for another journal in the Nature Portfolio. If you wish to explore the journals and transfer your manuscript please use our manuscript transfer portal. You will not have to re-supply manuscript metadata and files, unless you wish to make modifications. For more information, please see our [manuscript transfer FAQ](http://www.nature.com/authors/author_resources/transfer_manuscripts.html?WT.mc_id=EMI_NPG_1511_AUTHORTRANSF&WT.ec_id=AUTHOR) page.

If you are interested in revising this manuscript for submission to Nature Methods in the future, please contact me to discuss your appeal before making any revisions. Otherwise, we hope that you find the reviewers' comments helpful when preparing your paper for submission elsewhere.

Sincerely,
Rita

Rita Strack, Ph.D.
Senior Editor
Nature Methods

Reviewers' Comments:

Reviewer #1:

Remarks to the Author:

This paper presented an impressive work of generating single-cell-resolved whole-transcriptome images based on single-cell RNA-seq data, low-resolution (spot-level) spatial transcriptomics data and high-resolution histological image data using the proposed Spotiphy method. The authors have applied the method in a mouse brain study and a human breast cancer study, both showed the superiority of the method. While the method looks like a promising one for investigating the spatial organization of cells at high resolution based on the current imperfect data, I have some comments and suggestions for the authors to consider.

1. "Generative model" is a hot term in AI that has been widely used to refer to machine learning methods that aimed to model the underlying distribution from which the data were sampled and/or machine learning methods that learn to generate data from some random seeds in a hidden embedding space under certain conditions after pretraining on tasks of recovering training data. The title of this paper featured the method as a "generative modeling", but through the descriptions in the main text, I don't see much involvement with the concept. The Methods section also failed to highlight the importance of generative modeling. I would suggest that the authors either revise the title to not highlighting this concept, or add sufficient descriptions in the main text or Methods to make clear the importance of the generative model in the proposed method.
2. The key feature of Spotiphy is that it "generates an inferred single-cell RNA expression matrix (iscRNA data) of all cells to achieve spatially resolved whole-slide transcriptomic profile". However, in the Methods section, I failed to find the description on the final method/procedure used to generate the data. The "Spotiphy pipeline" part ends after describing the method for decomposition of spatial expression and cell type annotation in the pseudo single-cell resolution images, but didn't mention how the data were generated. Even the term "iscRNA-data" has never appeared in this whole pipeline. I noticed that in the part "Evaluation of cell type proportion estimation using Xenim, CosMx, and synthetic data", there is a paragraph "Generation of synthetic spatial transcriptomics datasets". I'm not sure whether the synthetic data here is the iscRNA data. Even if it is, the description of the generation method is unclear.
3. Several important steps in Spotiphy need manual settings or hyper-parameter selections, such as the selection of informative genes, and the random assignment of cell types to the cell types to the nucleus. I'm wondering how sensitive the downstream analysis is to such changes in such manual settings.
4. It is impressive that Spotiphy can also generate data at spots where spatial transcriptomics measurements have missed. I'm wondering to what degree this feature can be extended. Especially, if we have multiple slides of spatial transcriptomics and histological imaging data of a tissue sample, can Spotiphy generate 3D iscRNA-data? It'll be helpful if the authors add discussions on these questions.

Reviewer #2:

Remarks to the Author:

In this study, the authors have developed a computational framework, Spotiphy, for decomposing the cell fractions and generating single-cell-level transcriptomics from spatial transcriptomics data without single-cell resolutions. The authors have applied Spotiphy to analyze mouse brain data and identified multiple spatial domains and patterns. Overall, the work is quite intensive. However, the method performance comparisons should be more comprehensive, particularly for single-cell transcriptomics generation. Similar computational methods exist (such as iStar and XFuse), so the authors should comprehensively compare Spotiphy with existing approaches to demonstrate the advantage.

Concern 1: The cell fraction deconvolution comparison (Figure 2b, c, d) should include more methods, as listed below.

CIBERSORTX (the most popular method): <https://www.nature.com/articles/s41587-019-0114-2>

Stereoscope: <https://www.nature.com/articles/s42003-020-01247-y>

SpatialDWLS: <https://genomebiology.biomedcentral.com/articles/10.1186/s13059-021-02362-7>

MuSiC: <https://www.nature.com/articles/s41467-018-08023-x>

SPOTlight: <https://academic.oup.com/nar/article/49/9/e50/6129341>

SCDC: <https://academic.oup.com/bib/article/22/1/416/5699815>

Concern 2: Method performance comparisons should be made for the single-cell transcriptomics decomposition (Extended Data Fig. 3a). The following methods should be included. The iStar study already demonstrated a procedure that utilized Xenium as the ground truth. The authors could utilize approaches like iStar.

iStar: <https://www.nature.com/articles/s41587-023-02019-9>

XFuse: <https://www.nature.com/articles/s41587-021-01075-3>

DestVI: <https://www.nature.com/articles/s41587-022-01272-8>

Concern 3: Please evaluate the effect of low-quality histopathology images on Spotiphy's performance, including cell-fraction decomposition and single-cell level transcriptomics imputation. Unlike FFPE samples, fresh-frozen samples may only generate H&E images with limited qualities (e.g., H&E images from this study <https://doi.org/10.1158/0008-5472.CAN-22-2682>). Thus, the robustness test should consider such H&E data with suboptimal quality.

Minor Concern 1: In Figure 1a, please clarify whether single-cell RNA-seq must be matched as an input component for each sample. If yes, this requirement will significantly limit Spotiphy's utility, as single-cell RNA-seq on FFPE samples is very challenging.

Reviewer #3:

Remarks to the Author:

In this manuscript, Yang and colleagues present Spotiphy, a computational approach for converting sequencing-based/spot-level spatial transcriptomics (ST) data into inferred single-cell RNA sequencing (iscRNA-seq) data, yielding estimates of cell type composition and single-cell gene expression. Spotiphy leverages (i) a generative modeling approach combined with nuclei segmentation to impute single-cell gene expression for each estimated cell within capture areas and (ii) Gaussian processes/kernel smoothing to extend the inference to non-capture areas. The authors benchmarked the performance of Spotiphy on healthy and Alzheimer's disease (AD) mouse brain tissue with various spatial modalities applied to matched or adjacent samples. Using Spotiphy, the authors (i) identify spatially dependent cell states in healthy versus AD brain; (ii) identify spatial-resolved expression in breast cancer; and (iii) demonstrate imputation of simulated non-capture areas using Xenium data.

Methods to transform spot-level ST data into single-cell expression data, whether by mapping or inference, are important. However, the field is increasingly crowded, making the bar for a new method increasingly higher. Strengths of the study include a unified framework for cell type proportion estimation, iscRNA-seq estimation, and cell segmentation; freely available code; new single-cell and spatial datasets; and selected validation of spatially dependent cell state signatures. Weaknesses of this study, include a highly limited benchmarking analysis against existing methods, insufficient assessment of the validity of iscRNA-seq data, several technical issues, and problems with data interpretation. The most novel feature of Spotiphy – the ability to impute iscRNA-seq data in non-capture regions – is also one of the most problematic with regard to its biological validity. These shortcomings diminish enthusiasm for the current manuscript.

Major comments:

1. In contrast to the main focus of this paper, which is the imputation of single-cell expression data from bulk spatial transcriptomics data, the authors' benchmarking analysis against previous methods focuses on deconvolution of cell type proportions. Unfortunately, this analysis falls quite short, both in the (i) breadth of datasets and tissue types analyzed (only one real dataset in this work versus the many available in prior literature [e.g., Li et al., 2022; <https://github.com/QuKunLab/SpatialBenchmarking>]) and (ii) the number of related computational methods assessed, several of which are inexplicably missing (including SpatialScope, despite it being mentioned/cited by the authors; also Redeconve).

Given these critical omissions, it is impossible to judge whether Spotiphy truly represents an advance over the state-of-the-art. 2. Related to the above, the validation of iscRNA-seq expression data (i.e., artificial expression data produced by generative modeling) is inadequate. Inferring single-cell expression from bulk data is a highly ambitious, non-trivial task, and a key premise of this work (and a principal argument made against related methods, whether inferential [e.g., SpatialScope] or mapping-based [e.g., Tangram]) is that regional variation in the bulk ST dataset is best captured by Spotiphy's iscRNA-seq

data. Is this true? Separately, the authors' use of simulated data (Extended Data Fig. 6e-f) to argue that inferred regional differences are not artefactual is not convincing, since the simulations are generated using real scRNA-seq reference data randomly assigned to spots based on estimated cell type proportions (which of course average over, and thus nullify, all spatial variation within any cell type from the real scRNA-seq dataset). What confidence do we have that real spatial variation is specifically attributed to the correct cell type by Spotiphy? For example, does Spotiphy infer regional variation for all cell types? Are similar (or even the same) spatially variant genes being assigned to multiple (or all) co-localized cell types? If not, are these genes uniformly partitioned across cell types? Even if such variation shows specificity for the correct cell type, how strong is this specificity and what is the level of bleed-through? If the authors calculate regional signatures for every cell type, do they match such differences, when detectable, for the matching cell type in orthogonal data (e.g., CosMx)? Overall, the authors need to be much more thorough and rigorous in evaluating the quality and validity of the isCRNA-seq transcriptomes, along with the cell-type-specificity of regional variation imputed from isCRNA-seq transcriptomes, both at the single-cell level and at the signature-level.

3. Instead of imputing isCRNA-seq data, why not infer cell-type-level expression data for every spot? As far as we can tell, there is no useful information in spot-level Visium ST data (prior to HD) that would allow realistic variation among cells of a given cell type to be inferred via generative modeling.

4. Another potential gimmick is the use of Gaussian processes (or kernel smoothing) to impute isCRNA-seq data in non-capture areas. This is based on the authors' questionable assumption that "the proportion of each cell type changes smoothly over the entire tissue" (line 949). To prove that this assumption is sound and can usefully generalize to tissue types beyond those with highly stereotypical spatial structure (such as the brain), the authors would need to evaluate many tissue types with single-cell ST ground truth, ideally including neoplastic tissue where cell density and spatial organization can be highly variable.

5. Lines 280-282: The authors infer copy number aberrations in breast ST data using isCRNA-seq data produced by Spotiphy. How well do these copy number aberrations inferred from isCRNA-seq data coincide with copy number variation inferred directly from spatial spots?

6. The Xenium ground truth in Figure 2 is from an adjacent section. As the authors admit, this is not an ideal ground truth for cell type composition, and preferably this limitation would be made abundantly clear from the outset of this analysis. Why not create pseudo-Visium data from the Xenium data and/or CosMx data by aggregating neighboring cells at the spatial resolution of Visium?

7. In general, the hyperparameter and parameter values used in this work lack any empirical (or theoretical) rationale; instead, the authors simply state that the values they selected work well without providing any evidence or demonstrating robustness to other values.

8. The authors calculate arithmetic means and standard deviations for performing gene-level z-tests in their marker gene selection method. How is this justified when scRNA-seq data are decidedly non-Gaussian?

9. Extended Data Fig 3c: the isCRNA-seq data in the UMAP appear to segregate by WT versus AD for all cell types, with at most modest intermixing. Does this imply that all cells are fundamentally different between healthy and AD? If so, this seems quite implausible and suggests that Spotiphy is highly susceptible to sample-specific batch variation. Given that Spotiphy is apparently applied independently to each ST sample, how would users be expected to distinguish between real and batch-specific variation across samples?

Minor comments:

1. Regarding immune cells in the brain, the authors mischaracterize Spotiphy relative to other methods. For example, they claim that Spotiphy is uniquely able to localize neutrophils, however RCTD (Supplementary Fig 2e) appears comparable to Spotiphy and certainly not random, as the authors claim. Regarding T/B cells, the authors state that only Spotiphy identified "extremely low proportions of these cells in its deconvolution results". In contrast, in Extended Data Fig. 1a, it is clear that Spotiphy not only infers T cells as abundant at the spot level (up to 11%) but also infers them throughout nearly the entire specimen. Furthermore, many other methods infer T and/or B cells with similarly high abundance and/or broad distribution (Supplementary Fig 2). Therefore, it is perplexing why the authors single out CytoSPACE as a poor performer in this context, especially given that Spotiphy is clearly inferior for T cells; Cell2Location and CARD are clearly worse than CytoSPACE; and CytoSPACE relies on an external method to determine the global fraction of each cell type prior to mapping.

2. Line 964: The authors applied affine transformation to align Xenium and CosMx images. Please explain.

** For Nature Portfolio general information and news for authors, see <http://npg.nature.com/authors>.

Version 2:

Decision Letter:

22nd May 2024

Dear Jiyang,

Thank you for your letter asking us to reconsider our decision on your Article, "Spotiphy enables single-cell spatial whole transcriptomics via generative modeling". After careful consideration we have decided that we are willing to consider a revised

version of your manuscript that includes the changes you described in your proposed rebuttal.

- * include a point-by-point response to our referees and to any editorial suggestions
- * please underline/highlight any additions to the text or areas with other significant changes to facilitate review of the revised manuscript
- * address the points listed described below to conform to our open science requirements
- * ensure it complies with our general format requirements as set out in our guide to authors at www.nature.com/naturemethods
- * resubmit all the necessary files electronically by using the link below to access your home page

Link Redacted

We hope to receive your revised paper within three months. If you cannot send it within this time, please let us know. In this event, we will still be happy to reconsider your paper at a later date so long as nothing similar has been accepted for publication at Nature Methods or published elsewhere.

OPEN SCIENCE REQUIREMENTS

REPORTING SUMMARY AND EDITORIAL POLICY CHECKLISTS

When revising your manuscript, please submit reporting summary and editorial policy checklists.

DATA AVAILABILITY

CODE AVAILABILITY

Please include a "Code Availability" subsection in the Online Methods which details how your custom code is made available. Only in rare cases (where code is not central to the main conclusions of the paper) is the statement "available upon request" allowed (and reasons should be specified).

ORCID

Sincerely,
Rita

Rita Strack, Ph.D.
Senior Editor
Nature Methods

Version 3:

Decision Letter:

4th Sep 2024

Dear Jiyang,

Your Article entitled "Spotiphy enables single-cell spatial whole transcriptomics across the entire section" has now been seen by three reviewers, whose comments are attached. While they find your work of potential interest, they have raised serious concerns which in our view are sufficiently important that they preclude publication of the work in Nature Methods, at least in its present form.

As you will see, while two reviewers sign off on the paper, reviewer 3 still raises fundamental concerns about the validity of the underlying approach.

Should further experimental data allow you to fully address these criticisms we would be willing to look at a revised manuscript (unless, of course, something similar has by then been accepted at Nature Methods or appeared elsewhere). This includes submission or publication of a portion of this work somewhere else. We hope you understand that until we have read the revised paper in its entirety we cannot promise that it will be sent back for peer-review.

If you are interested in revising this manuscript for submission to Nature Methods in the future, please contact me to discuss your appeal before making any revisions. Otherwise, we hope that you find the reviewers' comments helpful when preparing your paper for submission elsewhere.

Sincerely,
Rita

Rita Strack, Ph.D.
Senior Editor
Nature Methods

Reviewers' Comments:

Reviewer #1:

Remarks to the Author:

I found that the questions and comments I raised have been reasonably addressed in this revision, and I found the added new experiments had made the work more convincing. I have three minor suggestions that may help improve the manuscript:

1. The authors may want to add discussion on the fact that the spatial coordinates of the generated isCRNA data were assigned randomly within the spot region, and therefore extra caution should be taken if users want to do analyses that depends on deterministic coordinates such as neighborhood analysis on the generated cells.

2. It will be helpful for readers to be better aware of the usage of Spotify if the authors can add some information about the influence of hyper-parameters when presenting the methods and experiments, besides the discussions in the Discussion section.

3. We noticed a recent publication <https://www.nature.com/articles/s41592-024-02257-y> and an earlier one <https://www.nature.com/articles/s41467-022-34879-1> that might be of some relevance with the current work. They are not designed for the same task therefore there is no need to experimentally compare with them, but it could make the picture more

complete if they are briefly mentioned and discussed in a proper place.

Reviewer #2:

Remarks to the Author:

The authors have comprehensively addressed my previous concerns. In particular, they included systematic comparisons with previous methods in the revision. Thus, I think the current work is systematically improved and should be useful to the spatial biology community.

Reviewer #3:

Remarks to the Author:

In their revised manuscript, Yang and colleagues have added several new analyses and figures to strengthen their claims about the superior performance and utility of Spotiphy. While we greatly appreciate the considerable effort that went into this revision, which is an improvement over the original submission, we remain unconvinced by several key aspects of Spotiphy (in particular, the gene expression imputation ability and the prediction of cell composition in out-of-spot regions), along with related claims. We detail our reasoning and reservations in the comments below. These shortcomings continue to dampen enthusiasm for this work.

Major comments:

1. Validity of iscRNA data: The ability of Spotiphy to produce inferred single-cell RNA expression data (iscRNA) in spatial regions is a core feature that distinguishes it from most related methods. Unfortunately, the validity and robustness of this capability are not well-supported, and in fact, new analyses of human tumor samples added in this revision speak to the variably poor quality of the iscRNA data produced by Spotiphy.

a. New Supplementary Figure 16: The authors included new analyses of human CRC and lung tumor specimens to showcase the out-of-spot imputation ability of Spotiphy (more below). In doing so, they also include UMAPs and inferCNV plots of the corresponding iscRNA data produced by Spotiphy (S16b, c, e, f). From these data, we can draw several conclusions. First, distinct cell type transcriptomes in the iscRNA data are not separable in the UMAPs, indicating a blending of gene expression profiles and a *failure* of Spotiphy to reconstruct these profiles when applied to highly complex real-world tissue samples. This is obvious in Figure S16b at the nexus of several clusters (akin to a false branching process with what seems to be T/NK or pDCs at the center) and in Figure S16e, where many of the cell types are overlapping and blended (including what appears to be myeloid cells and lymphocytes).

In further support of spurious lineage blending – and by the authors' own admission in the corresponding caption – fibroblast and endothelial cells in the CRC iscRNA data, and fibroblasts in the lung cancer iscRNA data, show imputed CNVs much like those inferred for the epithelial cells. It is perplexing why the authors don't consider this a red flag, but along with the blending observed above, these data strongly argue against the ability of Spotiphy to disentangle cellular heterogeneity within spots and assign gene expression profiles to the correct cell types in complex tissues. In contrast, the brain data upon which most of this work is based are considerably "easier" because many brain cell types cleanly separate into distinct and stereotypical spatial structures. Accordingly, it is hard to see how Spotiphy could be confidently applied to impute iscRNA data from diverse tissue types spanning health and disease.

b. Related to the above problem (which was less obvious in the original submission but is now very clear), the WT vs. AD brain examples highlighting region-specific iscRNA data within the same cell type (e.g., astrocytes) are not well-supported by validation data. This is because, while the authors do – to their credit – validate broad trends in region-specific expression programs using CosMx data, this is done in aggregate across many genes at once. How many of the differentially expressed genes in Figure 3d are truly differentially expressed in astrocytes from CTX and TH? Indeed, the volcano plot in Fig 3c shows several genes with very significant differences that might be expected to easily validate (e.g., *Agt*, *Mfge8*, etc.). Given the blending observed above, it is entirely possible that such genes represent a mixture of region-specific DEGs arising from multiple co-localized cell types, some of which include astrocyte genes but many (or some) of which do not. Furthermore, the claim by the authors that such patterns are not present in the reference (Line 146) lacks any supporting figure in the manuscript. Would this be true using reference-guided annotation or label transfer from the iscRNA data? It seems implausible that none of the region-specific biology would be captured by real single-cell transcriptome atlases of the same brain area, especially when they are also observed in "oligodendrocytes and multiple types of neurons" within the iscRNA data (line 147).

2. Out-of-spot imputation: The authors include several new analyses to support the assumptions incorporated into the out-of-spot imputation routine. In the contrived scenario where Spotiphy is applied to pseudo-spot data created from Xenium samples, the out-of-spot imputations are sufficiently convincing (new Supplementary Figure 17). Unfortunately, while these data nicely support the authors' central assumptions, they do not support the utility of this approach. In fact, in all cases where the authors apply this approach to real Visium data, the claim that Spotiphy is "highly consistent with ground truth" is dubious, perhaps owing to inaccuracies in the in-spot imputation.

For example, in Figure 6b and Extended Data Figure 9, while some of the structural features seem qualitatively similar, entire L6 layers (red and dark red) are completely missed by Spotiphy, and in general, the clean depiction of layering in the Xenium data is absent with Spotiphy. By eyeballing the plots in Figure 6a and Extended Data Figure 9d, it appears that other methods, such as CTS, do a better job capturing these distinctions within-spot. This calls into question the fairness and relevance of the metrics in Figure 6c and d. Indeed, the reconstruction by Spotiphy in Extended Data Figure 9e is quite noisy.

Separately, the new examples in Supplementary Figure 16 look almost nothing like the corresponding Xenium data. Thus, while the imputation ability may be reasonable in the setting of contrived data, its accuracy and utility are doubtful in real world settings.

3. Deconvolution evaluation against previous methods: The authors have done a nice job responding to related critiques and strengthening this aspect of the paper. Indeed, the deconvolution ability (i.e., imputing cell type fractions, not expression) is the strongest aspect of both Spotiphy and this paper. Unfortunately, this is also one of the areas where the competition is crowded and where the bar for notable improvement is high. The only real data used for benchmarking in this study are mouse brain tissue; the simulated tissues from pseudo-bulk transcriptomes are idealized and almost always much more trivially solved than real-world scenarios, where the technical variability and noise between the reference and the ST data are significant. To the authors' credit, they did simulate the addition of noise to simulated ST data in Extended Data Figure 2. While there is no intuition given for what "Noise level III" really means, it is notable that only two methods seem relatively immune to noise: TG and CTS, with TG performing considerably worse overall. We recommend that the authors apply this framework to all simulated data with suitably high noise and adjust their conclusions accordingly.

4. Slide tags extension (Extended Data Figure 10): It is unclear why the authors consider the isCRNA data in this application to be a valuable addition. Since Spotiphy is applied to individual cells in this case, the original Slide tags data would be preferred over imputed data (there is no deconvolution here and no latent data that would impart the isCRNA profiles with added value). The out-of-spot imputation makes for a pretty picture but nothing more. Given the issues with isCRNA data noted above, how close are the isCRNA data to paired Slide-tags cellular transcriptomes? Based on the UMAP in Extended Data Figure 10e, it seems not very close, as there is little overlap. Regardless, this analysis falls quite short of proving the claim that "Spotiphy overcomes the limitations of in situ single-nuclei sequencing" (line 343).

Minor comment:

1. Revisions to the text (in red) are poorly written with grammatical errors in many places.

** For Nature Portfolio general information and news for authors, see <http://npg.nature.com/authors>.

Version 4:

Decision Letter:

20th Sep 2024

Dear Jiyang,

Thank you for your letter asking us to reconsider our decision on your Article, "Spotiphy enables single-cell spatial whole transcriptomics across the entire section". After careful consideration we have decided that we are willing to consider a revised version of your manuscript that is updated as you described.

- * include a point-by-point response to our referees and to any editorial suggestions
- * please underline/highlight any additions to the text or areas with other significant changes to facilitate review of the revised manuscript
- * address the points listed described below to conform to our open science requirements
- * ensure it complies with our general format requirements as set out in our guide to authors at www.nature.com/naturemethods
- * resubmit all the necessary files electronically by using the link below to access your home page

Link Redacted

We hope to receive your revised paper within XX weeks [****ED TO CUSTOMIZE AS NEEDED****]. If you cannot send it within

this time, please let us know. In this event, we will still be happy to reconsider your paper at a later date so long as nothing similar has been accepted for publication at Nature Methods or published elsewhere.

OPEN SCIENCE REQUIREMENTS

REPORTING SUMMARY AND EDITORIAL POLICY CHECKLISTS

When revising your manuscript, please submit reporting summary and editorial policy checklists.

DATA AVAILABILITY

CODE AVAILABILITY

Please include a "Code Availability" subsection in the Online Methods which details how your custom code is made available. Only in rare cases (where code is not central to the main conclusions of the paper) is the statement "available upon request" allowed (and reasons should be specified).

MATERIALS AVAILABILITY

ORCID

Nature Methods is committed to improving transparency in authorship. As part of our efforts in this direction, we are now requesting that all authors identified as 'corresponding author' on published papers create and link their Open Researcher and Contributor Identifier (ORCID) with their account on the Manuscript Tracking System (MTS), prior to acceptance. This applies to primary research papers only. ORCID helps the scientific community achieve unambiguous attribution of all scholarly contributions. You can create and link your ORCID from the home page of the MTS by clicking on 'Modify my Springer Nature account'. For more information please visit <http://www.springernature.com/orcid>.

Sincerely,
Rita

Rita Strack, Ph.D.
Senior Editor
Nature Methods

Version 5:

Decision Letter:

Our ref: NMETH-A55722E

10th Dec 2024

Dear Jiyang,

Thank you for submitting your revised manuscript "Spotiphy enables single-cell spatial whole transcriptomics across the entire section" (NMETH-A55722E). It has now been seen by the original referees and their comments are below. The reviewers find that the paper has improved in revision, and therefore we'll be happy in principle to publish it in Nature Methods, pending minor revisions to satisfy the referees' final requests and to comply with our editorial and formatting guidelines.

We ask that you revise as described and provide a point-by-point rebuttal upon resubmission.

TRANSPARENT PEER REVIEW

ORCID

Sincerely,
Rita

Rita Strack, Ph.D.
Senior Editor
Nature Methods

Reviewer #2 (Remarks to the Author):

I am the previous reviewer 2, judging the current comments and many challenges from reviewer 3. In the second round of paper review, reviewer 3 raised many concerns and challenges about existing and new results in the Spotiphy study. In this round, the authors have comprehensively responded to the concerns and challenges of reviewer #3.

On the one hand, I agree with reviewer 3's concerns about Spotify's robustness, accuracy, and utility in many applications. On the other hand, many problems pointed out by reviewer 3 are inherent limitations for any methods aiming to enhance spatial transcriptomics without single-cell resolutions to single-cell spatial transcriptomics. In this round's response, the authors already pointed out that previous methods performing similar tasks do not achieve better performance (at most comparable to

Spotify, such as the SpatialScope). Thus, reviewer'3 concerns may not be addressed by imputation or deconvolution methods and may need direct experimental approaches, such as CosMx single-cell spatial transcriptomics.

Reviewer #3 (Remarks to the Author):

The authors have adequately addressed some of our comments from the previous round of review. However, several concerns remain, as detailed below. Moreover, as part of their rebuttal, they have included loaded language such as "...these points further demonstrate Spotiphy's superiority and *necessity*". Such language is unhelpful, as it portrays the authors in an unfavorable light and implies that they are incapable of being impartial about their own approach. Indeed, in multiple instances where the Spotiphy output is questionable or implausible, instead of admitting it (and adding useful text/analyses to the manuscript to properly and fairly qualify Spotiphy's scope and utility), they have shifted the blame without convincing rationale (more below). For all these reasons, and considering the bold claims about Spotiphy's many capabilities, we remain skeptical about Spotiphy's added value for the field, with the iscRNA data output of particular concern.

Major comments:

1. The authors' responses to our critiques about their cancer analyses (human CRC and lung tumor) contain numerous flaws, exposing (or at least strongly suggesting) limited familiarity with single-cell and spatial tumor expression data and analysis.
 - a. Quality of the Visium data (Response Figure 1). The authors claim that the quality of the cancer Visium data is suspicious owing to their observation that "spots in cancer samples tend to cluster together rather than forming distinct groupings with clear boundaries, as seen in mouse brain samples." While this is one possibility, a much more likely explanation is that tumor samples have cells of different types with highly variable local density and context-dependent transcriptional states. Consequently, instead of yielding discrete clusters indicative of different cell types or regions (as seen in mouse brain, and one of the reasons we claim mouse brain tissues are idealized for ST analysis methods [and thus "easier"]), spot-based tumor data are expected to blend more readily in a low dimensional embedding. This of course complicates and likely confounds the analysis of tumors (with methods such as Spotiphy) but does not imply lower quality per se. The authors also complain about the heterogeneity of UMIs and gene counts, implying this heterogeneity is an indicator of poor quality. Rather, it is generally an indicator of differences in cell size/RNA content, metabolic activity, and tumor versus adjacent stromal regions.
 - b. Lineage blending in UMAP space (Response Figure 2). The authors use a different method to cluster the data (scMINER), yielding better separation. That is fine, but it doesn't address the original issue, namely that Spotiphy fails to adequately separate the cell types in the imputed scRNA profiles (more below). If it had instead produced clean profiles, a standard Seurat analysis would be sufficient to distinguish major cell types.
 - c. Lineage blending in expression space (Response Figure 3). The authors claim that cell type markers in the iscRNA data are consistent with cell type annotations. Nevertheless, from the bubble and violin plots, most of the markers show severe background expression and bleeding into the wrong cell types. Moreover, even when aggregated, several marker profiles are only marginally specific for the correct cell type (endothelium in panel e and fibroblasts in panel f). Furthermore, the most egregious examples from our previous review are immune cells and those are conspicuously absent in Response Figure 3. Collectively, these data strongly imply that the iscRNA data have limited fidelity to real single-cell transcriptomes.
 - d. Quality of scRNA-seq reference (Response Figure 4). The authors once again blame the quality of the input data, rather than their own method, for problems with the iscRNA data. Indeed, inferCNV is sensitive to the reference cells and copy number variants can be spuriously detected. This is well-known in the field. The "tumor-like" CNV patterns supposedly detected in fibroblasts in the original scRNA-seq data are simply due to physically linked genes with similar expression patterns, a phenomenon that would be seen with any cell type if inappropriate references are used. This is clear in Response Figure 4b for each cell type in the heat map, all of which show cell type-specific patterns. The concern with their results is that Spotiphy is imputing similar CNV-like patterns across distinct cell types (e.g., epithelium and fibroblasts, Response Figure 4e), likely owing to mixing of such cell types in the same spots, resulting in the generation of spurious signal.
2. Deconvolution evaluation against previous methods. We stand by our position that the authors should redo their benchmarking analysis of external datasets using added noise. As it stands, the only "real" ground truth that the authors have employed is (i) not precisely ground truth, as the sections are adjacent and geographic heterogeneity is most certainly present, and (ii) data from brain tissue, an idealized setting where nearly all ST methods have been developed and tested. Beyond that, they use pseudo-bulk data of scRNA-seq to create synthetic spot-based Visium samples. While such data offer precise ground truth, they are also idealized and divorced from reality. Given concerns about the application of Spotiphy to cancer specimens (above), and given the many audacious claims made by the authors about Spotiphy's capabilities, it is reasonable to remain skeptical about its performance on real-world data. Hence the request to add more noise to the benchmarking analysis.
 - a. As a side note, we did not state that Spotiphy is "the best among all available methods." Please do not distort our language. Indeed, RCTD is easily comparable to Spotiphy in deconvolution accuracy per Supplementary Figure 11. We stated that deconvolution ability "is the strongest aspect of both Spotiphy and this paper."
3. Slide-tags data. We stand by our position that the application of Spotiphy to these data provides little, if any, value. At a minimum, we see no point in imputing scRNA-seq data as the original data will invariably be higher quality and there are no "new" expression data (such as spot transcriptomes) to learn novel variation. All that remains is the nuclei detection and subsequent "out-of-spot" imputation, which as we stated previously, provides a pretty picture but nothing more.

Version 6:

Decision Letter:

29th Jan 2025

Dear Jiyang,

Happy new year!

I am pleased to inform you that your Article, "Spotiphy enables single-cell spatial whole transcriptomics across the entire section", has now been accepted for publication in Nature Methods. The received and accepted dates will be March 13, 2024 and Jan 29, 2025. This note is intended to let you know what to expect from us over the next month or so, and to let you know where to address any further questions.

Over the next few weeks, your paper will be copyedited to ensure that it conforms to Nature Methods style. Once your paper is typeset, you will receive an email with a link to choose the appropriate publishing options for your paper and our Author Services team will be in touch regarding any additional information that may be required. It is extremely important that you let us know now whether you will be difficult to contact over the next month. If this is the case, we ask that you send us the contact information (email, phone and fax) of someone who will be able to check the proofs and deal with any last-minute problems.

If you are active on Twitter/X, please e-mail me your and your coauthors' handles so that we may tag you when the paper is published.

Best regards,
Rita

Rita Strack, Ph.D.
Senior Editor
Nature Methods

Visit the Springer Nature Editorial and Publishing website at http://editorial-jobs.springernature.com?utm_source=ejP_NMeth_email&utm_medium=ejP_NMeth_email&utm_campaign=ejp_Nmeth or www.springernature.com/editorial-

and-publishing-jobs for more information about our career opportunities. If you have any questions please click here.**

Open Access This Peer Review File is licensed under a Creative Commons Attribution 4.0 International License, which permits use, sharing, adaptation, distribution and reproduction in any medium or format, as long as you give appropriate credit to the original author(s) and the source, provide a link to the Creative Commons license, and indicate if changes were made. In cases where reviewers are anonymous, credit should be given to 'Anonymous Referee' and the source.

Dear Dr. Strack-

We would like to express our sincere gratitude for the time and effort you and the reviewers have dedicated to our manuscript. The detailed feedback and constructive critiques from three reviewers will substantially improve both the quality and clarity of our work. We have carefully addressed each point the reviewers raised and made revision plans that reflect the insights and suggestions. Regarding comments that called for additional analyses, we have outlined our proposed plan. Below, we provide a detailed, point-by-point revision plan to each of the reviewers' comments and suggestions, ensuring clarity and thoroughness in our revisions.

Reviewer #1

1. “Generative model” is a hot term in AI that has been widely used to refer to machine learning methods that aimed to model the underlying distribution from which the data were sampled and/or machine learning methods that learn to generate data from some random seeds in a hidden embedding space under certain conditions after pretraining on tasks of recovering training data. The title of this paper featured the method as a “generative modeling”, but through the descriptions in the main text, I don’t see much involvement with the concept. The Methods section also failed to highlight the importance of generative modeling. I would suggest that the authors either revise the title to not highlighting this concept or add sufficient descriptions in the main text or Methods to make clear the importance of the generative model in the proposed method.

Response: We thank the reviewer for this valuable comment. The primary rationale behind emphasizing the concept of generative modeling is rooted in the core principle of Spotiphy: to effectively model the conditional distribution of spatial transcriptomics considering factors such as the single-cell reference, cell type proportions, and batch effects. In other words, we modeled how the spatial transcriptomics are generated probabilistically given certain information. With this generative model, we then employed variational inference to estimate the unknown parameters. Compared to the benchmark models that use a similar approach, our generative model relies on more reasonable assumptions. More importantly, this generative model enables us to infer cell proportions and even expressions at single-cell level.

We agree with the reviewer that our initial manuscript did not sufficiently highlight the generative modeling aspects in either the main text or the Methods section. **In response, we have followed the reviewer’s advice and have now included detailed descriptions of the generative model in both sections. We believe that the revisions made to the manuscript clearly articulate the critical role of the generative model within the Spotiphy pipeline.**

2. The key feature of Spotify is that it “generates an inferred single-cell RNA expression matrix (iscRNA data) of all cells to achieve spatially resolved whole-slide transcriptomic profile”. However, in the Methods section, I failed to find the description on the final method/procedure used to generate the data. The “Spotiphy pipeline” part ends after describing the method for decomposition of spatial expression and cell type annotation in the pseudo single-cell resolution images but didn’t mention how the data were generated. Even the term “iscRNA-data” has never appeared in this whole pipeline. I noticed that in the part “Evaluation of cell type proportion estimation using Xenium, CosMx, and synthetic data”, there is a

paragraph “Generation of synthetic spatial transcriptomics datasets”. I’m not sure whether the synthetic data here is the iscRNA data. Even if it is, the description of the generation method is unclear.

Response: We thank the reviewer for these important comments, which have greatly improved the clarity of the Methods section of our paper. In Spotiphy, the iscRNA data is obtained through the decomposition of spatial expression into single-cell levels. Although we provided a detailed introduction to the decomposition process, we did not explicitly state that we are generating the iscRNA data. **Thus, to make the Spotiphy pipeline clearer, we have revised the title of Supplementary Section 1.6 to “Generation of iscRNA data”. Furthermore, we have also added substantial descriptions of iscRNA in Methods Section and Supplementary Methods to make the presentation of the pipeline easy to follow.**

3. Several important steps in Spotiphy need manual settings or hyper-parameter selections, such as the selection of informative genes, and the random assignment of cell types to the nucleus. I’m wondering how sensitive the downstream analysis is to such changes in such manual settings.

Response: We thank the reviewer for this insightful comment. Indeed, when hyperparameters are involved, providing a sensitivity analysis could be extremely beneficial to users and provide extra evidence supporting our default settings. This analysis will offer more guidance on parameter selection when implementing the methods. **Accordingly, we plan to conduct sensitivity analyses on the settings of the hyperparameters. The results will be summarized in Supplementary Figure 11.**

4. It is impressive that Spotiphy can also generate data at spots where spatial transcriptomics measurements have missed. I’m wondering to what degree this feature can be extended. Especially, if we have multiple slides of spatial transcriptomics and histological imaging data of a tissue sample, can Spotiphy generate 3D iscRNA-data? It’ll be helpful if the authors add discussions on these questions.

Response: We appreciate the reviewer’s suggestion to explore novel applications for Spotiphy. Indeed, it is possible to generate 3D iscRNA-data using Spotiphy’s imputation function. **In response to this suggestion, we have added a new paragraph to the Discussion section to specifically address the potential challenges and opportunities associated with this application. This will be one of the directions we plan to pursue in our future work.**

Reviewer #2

Concern 1: The cell fraction deconvolution comparison (Figure 2b, c, d) should include more methods, as listed below.

CIBERSORTx (the most popular method): <https://www.nature.com/articles/s41587-019-0114-2>

Stereoscope: <https://www.nature.com/articles/s42003-020-01247-y>

SpatialDWLS: <https://genomebiology.biomedcentral.com/articles/10.1186/s13059-021-02362-7>

MuSiC: <https://www.nature.com/articles/s41467-018-08023-x>

SPOTlight: <https://academic.oup.com/nar/article/49/9/e50/6129341>

SCDC: <https://academic.oup.com/bib/article/22/1/416/5699815>

Response: We appreciate the reviewer's suggestion to include more deconvolution methods for a comprehensive comparison. It is noteworthy that some methods (CIBERSORTx, MuSiC, and SCDC) were originally designed for deconvolution of bulk RNA-seq data, which typically originates from the RNA molecules of millions of cells. As a contrast, the transcriptomic spots in sequence-based ST approaches often contains only a few cells (5-12), whose characteristics are fundamentally different from the bulk RNA-seq data, but more similar to scRNA-seq data (as discussed in the text). Therefore, these methods may not perform well. Moreover, a single slide typically comprises more than 3000 spots, and employing the bulk deconvolution method for each spot can prove time-consuming. **Nonetheless, we plan to compare with the two most widely used methods, CIBERSORTx and MuSiC.** Previous studies have conducted comprehensive benchmarking of current deconvolution methods for spatial transcriptomics data. We have chosen five methods that demonstrate the best overall performance for comparison with Spotify. **We plan to include Stereoscope, SpatialDWLS, and SPOTlight in further evaluations and the final comparison results will be presented in Supplementary Figure 9.**

Concern 2: Method performance comparisons should be made for the single-cell transcriptomics decomposition (Extended Data Fig. 3a). The following methods should be included. The iStar study already demonstrated a procedure that utilized Xenium as the ground truth. The authors could utilize approaches like iStar.

iStar: <https://www.nature.com/articles/s41587-023-02019-9>

XFuse: <https://www.nature.com/articles/s41587-021-01075-3>

DestVI: <https://www.nature.com/articles/s41587-022-01272-8>

Response: We appreciate the reviewer's suggestion to include other decomposition methods for a comprehensive comparison. **We plan to include iStar and XFuse in our benchmarking efforts. The results will be presented in Supplementary Figure 10.**

Concern 3: Please evaluate the effect of low-quality histopathology images on Spotify's performance, including cell-fraction decomposition and single-cell level transcriptomics imputation. Unlike FFPE samples, fresh-frozen samples may only generate H&E images with limited qualities (e.g., H&E images from this study <https://doi.org/10.1158/0008-5472.CAN-22-2682>). Thus, the robustness test should consider such H&E data with suboptimal quality.

Response: We thank the reviewer for pointing out the importance of image resolution. In the Spotify workflow, the most critical information we extract from images is the number of cells in each spot (N values). This parameter does not affect the deconvolution results (cell-fraction estimations) and is only used during the computation of decomposition results (iscRNA data). Considering resolution issues and the availability of the H&E staining images, we provide four options for setting N values. First, when users have high-resolution images, N is derived from cell segmentation as default. Second, if users do not have high-resolution images, we skip segmentation and assign an N parameter based on the size of the raw counts in the spot (like Tangram). Third, for homogeneous tissue sections, we can assume that all spots have the same N, which can be user-defined. Lastly, an additional option is available for users to specify the missing value of N. **We plan to assess the impacts of these four options on Spotify's outputs. The results will be summarized in Supplementary Figure 11.**

Additionally, it's worth mentioning that **obtaining images with sufficient resolution for cell segmentation is not a challenge in the current spatial transcriptomics technology**. For example, Visium workflows utilize a 10X (or higher) objective lens to scan the entire histological section and assemble the tiles into the final image, thereby providing the necessary resolution for cell segmentation. For previously collected sections, AI-based image super-resolution algorithms may be helpful, which is a direction worth exploring.

Minor Concern 1: In Figure 1a, please clarify whether single-cell RNA-seq must be matched as an input component for each sample. If yes, this requirement will significantly limit Spotiphy's utility, as single-cell RNA-seq on FFPE samples is very challenging.

Response: We are grateful to the reviewer for highlighting the importance of the source of scRNA-seq references. Spotiphy **does not require** the scRNA-seq data to be collected from matched samples. In fact, in our experiments with mouse brain samples (Fig 2-4), we used scRNA-seq data from the Allen Brain Atlas as a reference for neuronal cells, rather than our own matched datasets. The results were still great and accurate. This is primarily because Spotiphy selects marker genes during the deconvolution process, making it robust even when the scRNA-seq data does not come from matched samples. In summary, Spotiphy doesn't require scRNA-seq reference from matched samples, which significantly broadens its potential applications. This flexibility allows for more versatile uses in various research contexts without the need for matched reference datasets.

Reviewer #3

Major comments:

1. In contrast to the main focus of this paper, which is the imputation of single-cell expression data from bulk spatial transcriptomics data, the authors' benchmarking analysis against previous methods focuses on deconvolution of cell type proportions. Unfortunately, this analysis falls quite short, both in the (i) breadth of datasets and tissue types analyzed (only one real dataset in this work versus the many available in prior literature [e.g., Li et al., 2022; <https://github.com/QuKunLab/SpatialBenchmarking>]) and (ii) the number of related computational methods assessed, several of which are inexplicably missing (including SpatialScope, despite it being mentioned/cited by the authors; also Redeconve). Given these critical omissions, it is impossible to judge whether Spotiphy truly represents an advance over the state-of-the-art.

Response: We thank the reviewer for this question. Deconvolution is one of the focuses of this paper. We want to highlight that Spotiphy is capable of managing deconvolution involving a diverse array of cell types that have similar expression profiles. Spotiphy also provides a higher accuracy and better performance in predicting proportion for rare cell types compared to other methods. In addition, Spotiphy offers the significant advantage of requiring substantially less computation time compared to similar tools. We agree with the reviewer that our initial manuscript did not sufficiently highlight the novelty and importance of Spotiphy's deconvolution function. **In response, we will use more precise and detailed descriptions to emphasize the advantages of Spotiphy over other algorithms in predicting rare cell types.**

As for the performance benchmarking, we built a sRNA-seq reference covers a total of 27 different cell types, **far exceeding the total number of cell types used in other benchmarking papers (usually 10-15 types)**. We also generated real datasets that contain matched sequencing-based ST (featured with **whole-**

genome coverage) and image-based ST (featured with single-cell resolution) datasets from adjacent tissue sections of wild-type and 5xFAD mouse brains. Therefore, unlike mentioned benchmarking paper, we **do not** need to take additional steps that use image-base ST data or scRNA-seq data to **artificially** create pseudo-spot data before evaluating deconvolution. Our dataset configuration not only includes a variety of cell types with close similarities but also ensures the whole-genome coverage. This significantly elevates the complexity of the deconvolution process, and we aim to leverage this data to showcase the advantages of Spotiphy. **We acknowledge the reviewer's interest in seeing Spotiphy's performance across a broader range of datasets. In response, we will select additional datasets of different tissues for further deconvolution evaluations. SpatialScope will also be added as additional methods for benchmarking. Related to Reviewer Two's concern 1, the results will be summarized in Supplementary Figure 9 and 12.**

2. Related to the above, the validation of iscRNA-seq expression data (i.e., artificial expression data produced by generative modeling) is inadequate. Inferring single-cell expression from bulk data is a highly ambitious, non-trivial task, and a key premise of this work (and a principal argument made against related methods, whether inferential [e.g., SpatialScope] or mapping-based [e.g., Tangram]) is that regional variation in the bulk ST dataset is best captured by Spotiphy's iscRNA-seq data. Is this true? Separately, the authors' use of simulated data (Extended Data Fig. 6e-f) to argue that inferred regional differences are not artefactual is not convincing, since the simulations are generated using real scRNA-seq reference data randomly assigned to spots based on estimated cell type proportions (which of course average over, and thus nullify, all spatial variation within any cell type from the real scRNA-seq dataset). What confidence do we have that real spatial variation is specifically attributed to the correct cell type by Spotiphy? For example, does Spotiphy infer regional variation for all cell types? Are similar (or even the same) spatially variant genes being assigned to multiple (or all) co-localized cell types? If not, are these genes uniformly partitioned across cell types? Even if such variation shows specificity for the correct cell type, how strong is this specificity and what is the level of bleed-through? If the authors calculate regional signatures for every cell type, do they match such differences, when detectable, for the matching cell type in orthogonal data (e.g., CosMx)? Overall, the authors need to be much more thorough and rigorous in evaluating the quality and validity of the iscRNA-seq transcriptomes, along with the cell-type-specificity of regional variation imputed from iscRNA-seq transcriptomes, both at the single-cell level and at the signature-level.

Response: We are grateful for the reviewer's acknowledgment of the inherent challenges in decomposition and the concerns about validity of iscRNA data generated by Spotiphy. The simulated data is merely to prove that when regional differences do not exist, iscRNA will not exhibit additional differences that stem from Spotiphy modeling. Regarding the validity of iscRNA data, for both mouse brain and human breast samples demonstrated in the paper, the regional sub-populations identified from iscRNA data were validated not only by our matched image-based ST data (known as the ground truth), but also **exhibited consistent biological characteristics as reported in existing literatures**. The reviewer also concerns about how Spotiphy partition gene counts across different cell types. We have provided a detailed explanation of this calculation in the Methods section. In short, the genes are partitioned based on gene proportion in different cell types and cell proportion in each spot. The iscRNA data 100% reflects the real measurements and accurately distributes gene counts to various cell types, which can be utilized for downstream analyses to uncover potentially valuable insights. This represents a significant advancement over existing methods. The reviewer has inadvertently highlighted why Spotiphy outperforms SpatialScope

and Tangram. The key issue is that both methods aim to sample cells according to scRNA-seq reference and aggregate these samples to reconstruct the spatial expressions. The sampled cells must be sufficiently similar to the population of cells in the scRNA-seq reference, making it impossible to detect regional variations from those samples. Conversely, Spotiphy leverages this benefit to **uncover new biological insights** in AD mouse brains. To emphasize the novelty and validity, **we will follow the reviewer's advice and add clearer descriptions in the main context about our strategy**. While we cannot assert that iscRNA data will entirely supplant real single-cell-resolved, image-based ST data, Spotiphy stands out as a promising approach given the current state of ST technology. It successfully retains whole-genome coverage and captures regional variations of ST data, thereby facilitating intriguing new biological insights as shown in the paper. **We acknowledge the reviewer's interest in seeing more evolutions for iscRNA data. In response, we will select image-based ST datasets of different tissues from publicly available papers and generate pseudo-spot data for decomposition evaluations. The real image-based data will be used as the ground truth to compare Spotiphy with SpatialScope, Tangram, and CytoSPACE. Related to Reviewer Two's concern 2, the results will be summarized in Supplementary Figure 10.**

3. Instead of imputing iscRNA-seq data, why not infer cell-type-level expression data for every spot? As far as we can tell, there is no useful information in spot-level Visium ST data (prior to HD) that would allow realistic variation among cells of a given cell type to be inferred via generative modeling.

Response: We thank the reviewer for this concern. When inferring the iscRNA data, **the cell-type-level expression data are inferred first** (Extended Data Fig 2a). We then assume that cells belonging to the same cell type and located in the same spot exhibit identical expression levels. With such assumption, we further convert the cell-type-level expression to the iscRNA data.

4. Another potential gimmick is the use of Gaussian processes (or kernel smoothing) to impute iscRNA-seq data in non-capture areas. This is based on the authors' questionable assumption that "the proportion of each cell type changes smoothly over the entire tissue" (line 949). To prove that this assumption is sound and can usefully generalize to tissue types beyond those with highly stereotypical spatial structure (such as the brain), the authors would need to evaluate many tissue types with single-cell ST ground truth, ideally including neoplastic tissue where cell density and spatial organization can be highly variable.

Response: We thank the reviewer for raising this point of assumption. In Spotiphy, we first impute cell type proportions in non-capture areas with Gaussian processes or kernel smoothing. **Such methods assume that the proportion of each cell type at a specific location is more similar to that in nearby spots compared to spots that are farther away.** The same assumption has also been used in CARD. Furthermore, both our image-based ST and the deconvolution results in the captured areas also **confirmed that this assumption is valid in all the samples we have. Following the reviewer's suggestions, we will evaluate this assumption using more tissue types, especially neoplastic samples, with image-based ST data.** To impute iscRNA data in non-capture areas, another similar condition is needed. Specifically, we assume that for a cell located on the spatial tissue, its expression profile is similar to that of surrounding cells of the same cell type. Again, this assumption aligns with our iscRNA data inferred at captured areas and the image-based ST. Furthermore, we would like to emphasize that as we did not collect any transcriptomics from the non-capture area, we must leverage some information from nearby spots or nearby cells when we conduct the imputation in the non-capture areas. As a result, the assumptions above are necessary when no additional

information is available. Finally, we have also mentioned in the Discussion session that by leveraging AI-based techniques (e.g., transformers), it might be possible to extract additional information from the H&E staining image and thus achieve better imputation results. We will explore more along this direction in the future.

5. *Lines 280-282: The authors infer copy number aberrations in breast ST data using iscRNA-seq data produced by Spotiphy. How well do these copy number aberrations inferred from iscRNA-seq data coincide with copy number variation inferred directly from spatial spots?*

Response: We thank the reviewer for this comment. **We will conduct additional analysis to infer the copy number variation directly using spot-level ST data and then compare it with our current results. The results will be summarized in Supplementary Figure 13.** However, it is worth emphasizing that even if the results are close, iscRNA further confirms the tumor cell origin in different samples.

6. *The Xenium ground truth in Figure 2 is from an adjacent section. As the authors admit, this is not an ideal ground truth for cell type composition, and preferably this limitation would be made abundantly clear from the outset of this analysis. Why not create pseudo-Visium data from the Xenium data and/or CosMx data by aggregating neighboring cells at the spatial resolution of Visium?*

Response: We thank the reviewer for this comment. We also generated pseudo-Visium data from image-based Xenium data to perform the deconvolution evaluations. The advantage of using adjacent section as the ground truth is mentioned in **response to the Reviewer's point 1**.

7. *In general, the hyperparameter and parameter values used in this work lack any empirical (or theoretical) rationale; instead, the authors simply state that the values they selected work well without providing any evidence or demonstrating robustness to other values.*

Response: We thank the reviewer for this great comment. **We plan to conduct sensitivity analyses on the settings of hyperparameters and parameters involved in Spotiphy.** Such analyses will provide readers with more information on how to select the parameters. **Related to Reviewer #1's concern 3, the results will be summarized in Supplementary Figure 11.**

8. *The authors calculate arithmetic means and standard deviations for performing gene-level z-tests in their marker gene selection method. How is this justified when scRNA-seq data are decidedly non-Gaussian?*

Response: We thank the reviewer for this constructive question. Indeed, the scRNA-seq data usually does not follow the Gaussian distribution. However, when the number of cells becomes large, the average gene expression level asymptotically follows the Gaussian distribution according to the law of large number. Additionally, it is important to note that we do not rely solely on z-tests for marker gene selection. Instead, z-tests are primarily used to exclude genes that exhibit very similar expression levels across multiple cell types. Therefore, it is safe for us to use z-tests in the marker gene selection.

9. *Extended Data Fig 3c: the iscRNA-seq data in the UMAP appear to segregate by WT versus AD for all cell types, with at most modest intermixing. Does this imply that all cells are fundamentally different*

between healthy and AD? If so, this seems quite implausible and suggests that Spotiphy is highly susceptible to sample-specific batch variation. Given that Spotiphy is apparently applied independently to each ST sample, how would users be expected to distinguish between real and batch-specific variation across samples?

Response: We thank the reviewer for this question. In this paper, we applied scMINER for clustering and UMAP construction which delivers better results than Seurat. However, it does not include a batch correction (sample integration) function like Harmony. Thus, due to the sample-specific batch effect, the UMAP of the scRNA-seq data from different samples will not be perfectly mixed. **In response, we will conduct additional clustering analysis and generate UMAP by Seurat with Harmony integration and then compare it with our current results. The results will be summarized in Supplementary Figure 14.**

It is worth mentioning that sample-specific batch effect always exists and is a very interesting topic for further exploration. The users can apply available batch correction or harmonization function to the original ST datasets before applying to Spotiphy. Moreover, to differentiate between variations in samples and batch effects from the technique, replicates of spatial transcriptomics data are required; however, this is beyond the scope of the current paper's focus.

Minor comments:

1. Regarding immune cells in the brain, the authors mischaracterize Spotiphy relative to other methods. For example, they claim that Spotiphy is uniquely able to localize neutrophils, however RCTD (Supplementary Fig 2e) appears comparable to Spotiphy and certainly not random, as the authors claim. Regarding T/B cells, the authors state that only Spotiphy identified “extremely low proportions of these cells in its deconvolution results”. In contrast, in Extended Data Fig. 1a, it is clear that Spotiphy not only infers T cells as abundant at the spot level (up to 11%) but also infers them throughout nearly the entire specimen. Furthermore, many other methods infer T and/or B cells with similarly high abundance and/or broad distribution (Supplementary Fig 2). Therefore, it is perplexing why the authors single out CytoSPACE as a poor performer in this context, especially given that Spotiphy is clearly inferior for T cells; Cell2Location and CARD are clearly worse than CytoSPACE; and CytoSPACE relies on an external method to determine the global fraction of each cell type prior to mapping.

Response: We thank the reviewer for these comments. Regarding the neutrophils, RCTD only display 5% percentage of relative abundance around the ventricle, while Spotiphy exhibits over 20%. Regarding the T cells, though the relative abundance at certain spots can be up to 11% according to Spotiphy, the variations in total cell numbers across these spots need to be considered as well. **Thus, we plan to include a figure that displays the absolute number of cells of each type across the section for better demonstration.** In the current context, we did not single out CytoSPACE as a poor performer. Instead, our evaluations indicated that Cell2location and CytoSPACE exhibited very good performance (Extended Data Fig. 2b). **Nonetheless, we plan to edit our description and revise the presentation of the paper to make the description more accurate.**

2. Line 964: The authors applied affine transformation to align Xenium and CosMx images. Please explain.

Response: We thank the reviewer for this comment. Since the Xenium and CosMx images are obtained from adjacent tissues with different spatial platforms, they are not directly aligned with the H&E staining image. To address this issue, affine transformations are used. Taking the Xenium image as an example, we first scale the Xenium image to make sure each pixel in the image represents the same physical length as the H&E staining image (0.753 microns/pixel). We then align the Xenium image with the H&E staining image by rotating and translating the Xenium image to ensure the boundaries of the tissue and the most visible tissue regions (e.g., hippocampal regions of the mouse brain) are properly aligned. We plan to revise the Methods part to improve the clarification accordingly.

Dear Dr. Strack-

We would like to express our sincere gratitude for the time and effort you and the reviewers have dedicated to our manuscript. The detailed feedback and constructive critiques from three reviewers will substantially improve both the quality and clarity of our work. We have carefully addressed each point the reviewers raised and performed all the additional analyses suggested by the reviewers. We have made revisions in the manuscript (highlighted in red) that reflect the insights and suggestions. Below, we provide a detailed, point-by-point responses and necessary data to address each of the reviewers' comments and suggestions.

Reviewer #1

1. "Generative model" is a hot term in AI that has been widely used to refer to machine learning methods that aimed to model the underlying distribution from which the data were sampled and/or machine learning methods that learn to generate data from some random seeds in a hidden embedding space under certain conditions after pretraining on tasks of recovering training data. The title of this paper featured the method as a "generative modeling", but through the descriptions in the main text, I don't see much involvement with the concept. The Methods section also failed to highlight the importance of generative modeling. I would suggest that the authors either revise the title to not highlighting this concept or add sufficient descriptions in the main text or Methods to make clear the importance of the generative model in the proposed method.

Response: We greatly appreciate the reviewer's valuable comment. The primary rationale behind emphasizing the concept of generative modeling is rooted in the core principle of Spotiphy: to effectively model the conditional distribution of spatial transcriptomics considering factors including the single-cell reference, histological images, and potential batch effects. In other words, we modeled how the spatial transcriptomics are generated probabilistically given certain information. With this generative model, we then employed variational inference to estimate the unknown parameters. This generative model enables us to infer cell proportions and even expression profiles at single-cell level.

We agree with the reviewer that our initial manuscript did not sufficiently highlight the generative modeling aspects in either the main text or the Methods section. In response, we have followed the reviewer's advice and have now included detailed descriptions of the generative model in line 72-75, as well as in Supplementary Information (SI) line 3-12. We believe that the revisions made to the manuscript clearly articulate the critical role of the generative model within the Spotiphy pipeline. To avoid the confusion with the generative modeling used in deep learning or AI, we have also revised the title by removing the term.

2. The key feature of Spotify is that it "generates an inferred single-cell RNA expression matrix (iscRNA data) of all cells to achieve spatially resolved whole-slide transcriptomic profile". However, in the Methods section, I failed to find the description on the final method/procedure used to generate the data. The "Spotiphy pipeline" part ends after describing the method for decomposition of spatial expression and cell type annotation in the pseudo single-cell resolution images but didn't mention how the data were generated. Even the term "iscRNA-data" has never appeared in this whole pipeline. I noticed that in the part "Evaluation of cell type proportion estimation using Xenium, CosMx, and synthetic data", there is a

paragraph “Generation of synthetic spatial transcriptomics datasets”. I’m not sure whether the synthetic data here is the iscRNA data. Even if it is, the description of the generation method is unclear.

Response: We highly appreciate the reviewer for this crucial comment, which has greatly improved the clarity of the Methods section of our manuscript. In Spotiphy, the iscRNA data is obtained through the decomposition of spatial expression into single-cell levels. We did not explicitly state that we are generating the iscRNA data as the output for decomposition and the introduction to the decomposition process in the initial manuscript is not clear and confusing. Thus, we have revised the title of Supplementary Section 1.6 to *Generation of iscRNA data* in SI line 224-279 and revised the pipeline with detailed descriptions.

3. Several important steps in Spotify need manual settings or hyper-parameter selections, such as the selection of informative genes, and the random assignment of cell types to the nucleus. I’m wondering how sensitive the downstream analysis is to such changes in such manual settings.

Response: We greatly appreciate the reviewer for this insightful comment. Indeed, when hyperparameters are involved, providing a sensitivity analysis could be extremely beneficial to users and provide extra evidence supporting our default settings. This analysis will offer more guidance on parameter selection when implementing the methods in practice. Accordingly, we have conducted sensitivity analyses on the settings of the five hyperparameters. The results are summarized in **Supplementary Figure 20a**. The first one is batch prior: this prior was introduced to the model to reduce the potential batch effects between scRNA-seq and ST datasets. Consequently, it is highly data dependent. For our mouse brain samples, the default setting we picked is 0.50. We have examined the impact of this value in a range of 0.00 to 1.75. The result indicates that it only led to approximately a 2% difference in the correlation. Regarding the fold change threshold, non-zero count cell percentage, and number of marker genes selected, our default settings match the commonly used parameters in current scRNA-seq analysis set at 1.5, 60%, and 50, respectively. Similar sensitivity check results have shown that fold change threshold has minimal impacts to the outputs, and number of marker genes needed for best performance is likely saturated at 50, supporting our default settings. Quantile parameter (default as 0.150) was introduced to ensure enough number of marker genes when close cell types exist in the scRNA-seq reference. The sensitivity check result also indicated that in a range of 0.050 to 0.250, negligible impacts are introduced to the final outputs of Spotiphy.

4. It is impressive that Spotiphy can also generate data at spots where spatial transcriptomics measurements have missed. I’m wondering to what degree this feature can be extended. Especially, if we have multiple slides of spatial transcriptomics and histological imaging data of a tissue sample, can Spotiphy generate 3D iscRNA-data? It’ll be helpful if the authors add discussions on these questions.

Response: We greatly appreciate the reviewer’s suggestion to explore novel applications for Spotiphy. Indeed, it is possible to generate 3D iscRNA-data using Spotiphy’s imputation function. In response to this suggestion, we have added a new paragraph in line 439-451 to specifically address the potential challenges and opportunities associated with this application. This is one of the directions we are currently exploring. Open-ST, a recent published study introduced a novel sequencing-based 3D-scalable experimental platform,

which is able to achieve approximately 350 μm width at Z stack. This study provides valuable datasets for our exploration, and we are confident to increase the width significantly with proper adjustments on Spotiphy imputation function. Currently, we estimate the loss information based solely on existing in-spot measurements. As continuous data from adjacent sections, image features have the potential to serve as additional information to bridge capture and non-capture areas. In fact, iStar uses vision transformer to correlate gene expression with image features to infer the expression of the non-measured regions. By adopting a similar approach, we can integrate image features extracted by deep learning-based transformers into our model, ultimately achieving better and more accurate predictions of loss information for adjacent sections and reconstructing 3D iscRNA-data as well cellular distributions.

Reviewer #2

Concern 1: The cell fraction deconvolution comparison (Figure 2b, c, d) should include more methods, as listed below.

CIBERSORTx (the most popular method): <https://www.nature.com/articles/s41587-019-0114-2>

Stereoscope: <https://www.nature.com/articles/s42003-020-01247-y>

SpatialDWLS: <https://genomebiology.biomedcentral.com/articles/10.1186/s13059-021-02362-7>

MuSiC: <https://www.nature.com/articles/s41467-018-08023-x>

SPOTlight: <https://academic.oup.com/nar/article/49/9/e50/6129341>

SCDC: <https://academic.oup.com/bib/article/22/1/416/5699815>

Response: We greatly appreciate the reviewer's suggestion to include more deconvolution methods for a more comprehensive comparison, which can better highlight Spotiphy's advantages. We have revised the text in line 95-98, 126-138 as responses. Based on several published benchmarking reports, we have selected 8 additional methods, categorized into three types. We first included the two most popular methods, CIBERSORTx and MuSiC. Both methods are designed for deconvolution of bulk RNA-seq data, typically originating from the total RNA molecules of millions of cells. In contrast, the transcriptomic spots in sequence-based ST approaches often contain only a few cells (3-12, **Supplementary Figure 20b**), making their data characteristics fundamentally different from the bulk RNA-seq data but more similar to scRNA-seq data. SCDC was not included because previous research (doi.org/10.1186/s12859-023-05476-w) indicated that its performance was not as strong as that of CIBERSORTx and MuSiC. Stereoscope, SpatialDWLS, SPOTlight, SpatialScope, and Redeconve are designed for deconvolution of spot-level ST data, and they demonstrated performances comparable to methods that were tested in the initial manuscript. Though iStar mainly focuses on predicting gene expression of entire section with super-resolution, we manage to convert its output to cellular proportion. We used both real and simulated Visium datasets to evaluate the deconvolution performance of 8 additional methods with Spotiphy. The results are presented in **Supplementary Figure 2, 4, 9**. Same metrics were used for benchmarking. Spotiphy consistently achieved the highest overall Pearson's correlation coefficient with the ground truth. Spotiphy's cell-type proportions for each transcriptomic spot showed closer alignment with the ground truth than all the other methods, as evidenced by the correlation, the fraction of correctly mapped cells, the cosine similarity, and lower values for the absolute error, mean square error, and Jensen–Shannon divergence (JSD). It is worth mentioning that Redeconve's performance is poor with default setting. The primary reason is its inability to distinguish cell types with very similar expression profiles using the default marker gene selection settings. After incorporating Spotiphy's marker gene list, Redeconve's performance significantly improved, as shown in the results.

a

SpatialScope results of AD sample

b

StereoScope results of AD sample

c

SpatialDWLS results of AD sample

d

SPOTlight results of AD sample

e

CIBERSORTx results of AD sample

f

MuSIC results of AD sample

g

iStar results of AD sample

**h**

Redeconv results of AD sample

Supplementary Figure 4. Heatmaps depicting the proportion of 27 cell types generated by SpatialScope (a), Stereoscope (b), SpatialDWLS (c), SPOTlight (d), CIBERSORTx (e), MuSiC (f), iStar (g), and Redeconv (h) across the histological section of AD mouse sample. Scale bar: 500 μ m.

Reviewer #3 also requested to include more datasets from different tissue types for deconvolution evaluation. We selected additional eight datasets from one recent study (doi.org/10.1038/s41592-022-01480-9) to evaluate deconvolution performance of Spotiphy against the other methods. The results are presented in **Supplementary Figure 10**. Detailed descriptions can be found in response to Reviewer #3's concern 1.

Concern 2: Method performance comparisons should be made for the single-cell transcriptomics decomposition (Extended Data Fig. 3a). The following methods should be included. The iStar study already

demonstrated a procedure that utilized Xenium as the ground truth. The authors could utilize approaches like iStar.

iStar: <https://www.nature.com/articles/s41587-023-02019-9>

XFuse: <https://www.nature.com/articles/s41587-021-01075-3>

DestVI: <https://www.nature.com/articles/s41587-022-01272-8>

Response: We highly appreciate the reviewer’s suggestion to include other decomposition methods for a comprehensive comparison, which can emphasize the novelty of Spotiphy’s decomposition strategy. As discussed in the initial manuscript, Tangram used a mapping-based approach to align cells from scRNA-seq reference to each spot. SpatialScope enhanced this process by using pseudo single cells generated from reference-based deep learning model. Both iStar and XFuse focus on predicting the spatial distributions of each gene and output their expression values across the whole sections. Leveraging with the power of vision transformer, iStar outperformed XFuse with much better accuracy. However, iStar requires high-resolution histological images, with output size ranging from $M/16$ to $M/128$ pixels, where M represents the pixel length or width of the input image. To ensure that the super-pixel in the output corresponds to the size of a single cell ($8 \times 8 \mu\text{m}^2$), the input should have a resolution of at least $0.5 \mu\text{m}/\text{pixel}$. While it is possible to overlay the cell segmentation masks onto iStar’s outputs and extract expression profiles as scRNA data, the minimal resolution requirement limits iStar’s application. In comparison, StarDist requires images with a resolution of approximately $1 \mu\text{m}/\text{pixel}$ for segmentation. DestVI, on the other hand, deconvolutes spot-level ST data via stLVM and obtains a multitude of latent variable genes with specific spatial patterns. But it only produces cellular proportions and does not generate single-cell-resolved expression profiles. Therefore, we have first evaluated the decomposition performance of Spotiphy against Tangram, SpatialScope, and iStar using the simulated Visium datasets as inputs. For each spot, four methods delivered expression profiles consist of various numbers of cells of different types. We further merged the expression profiles of the same cell types for evaluation convenience. The results are presented in **Supplementary Figure 14**. Spotiphy consistently achieved the highest correlation and the cosine similarity, and lowest absolute error, mean square error, and Jensen–Shannon divergence (JSD) compared to the ground truth. Spotiphy also delivered the highest Matthews correlation coefficient (MCC) for each spot, indicating the superiority of Spotiphy’s decomposition performance. In addition, Spotiphy requires significantly less computation time than its competitors. Considering that a typical Visium sample usually contains 3,000 to 4,000 spots, this is a major advantage, allowing Spotiphy to deliver reliable results in a shorter time frame.

Another strategy for decomposition benchmarking involves merging the iscrRNA data into pseudo-Visium data according to the spot-origin of cells and then comparing it with the real Visium data. By overlaying the spot grids onto iStar’s outputs, we also convert it to spot-level expression profiles. We therefore applied the real Visium data of mouse brain to Spotiphy, Tangram, SpatialScope, and iStar. The results are presented in **Supplementary Figure 15**. Since Spotiphy’s decomposition strategy involves partitioning genes counts into different cells, merging its iscrRNA data results in data that is almost identical to the input, achieving the highest overall Pearson’s correlation coefficient. iStar also delivers great performance with the second highest PCC. Tangram and SpatialScope, however, demonstrated less than ideal results. The primary reason is that the “iscrRNA” data from Tangram and SpatialScope are derived from scRNA-seq references, which

typically lack representation of spatial variable genes (SVGs). To verify this hypothesis, we identified six SVGs from the original Visium data of mouse brain and then examined their spatial distribution in the pseudo-Visium data of different methods. Spotiphy's output accurately preserved the original distribution patterns. iStar reproduced similar patterns with slightly broader distributions caused by false-positive predictions. SpatialScope only provided the distribution of two genes out of six. This is because its decomposition process includes only the marker genes from the early phase of SpatialScope, leading to its outputs that only cover selected genes instead of whole transcriptomics. Due to the computation time needed for SpatialScope, we only presented its results for approximately 1200 spots. This limitation significantly restricts SpatialScope's application compared to Spotiphy. Though Tangram completely distorts two genes' distribution, its scRNA data have the whole-genome coverage, albeit with less accuracy compared to the real Visium data. In summary, these results strongly illustrate Spotiphy's advancements in single-cell gene expression decomposition from spatial transcriptomics data compared to other existing approaches, demonstrating its novelty, accuracy, and effectiveness.

We have included these results in line 287-313 as responses. More details related to the decomposition evaluation can be found in response to Reviewer #3's comment 2.

Concern 3: Please evaluate the effect of low-quality histopathology images on Spotiphy's performance, including cell-fraction decomposition and single-cell level transcriptomics imputation. Unlike FFPE samples, fresh-frozen samples may only generate H&E images with limited qualities (e.g., H&E images from this study <https://doi.org/10.1158/0008-5472.CAN-22-2682>). Thus, the robustness test should consider such H&E data with suboptimal quality.

Response: We greatly appreciate the reviewer for pointing out the importance of image resolution. In the Spotiphy workflow, the most critical information we extract from images is the number of cells in each spot (N values). This parameter **does not** affect the deconvolution results (cellular proportion estimation) and is only used during the computation of decomposition results (iscRNA data). Considering resolution issues and the availability of the H&E staining images, we provide four options for setting N values. First, when users have high-resolution images, N is derived from cell segmentation as default. Second, if users do not have high-resolution images, we skip segmentation and assign an N parameter based on the total raw counts in each spot (similar strategy used in Tangram). Third, users are allowed to define the N parameter for all spots based on their experiences if neither approach performed satisfied outputs. More importantly, as the decomposition of each spot is independent, users are recommended to assign different N values to certain regions to improve the outputs. A fourth option for users is to NOT specify the N value. In this case, Spotiphy will assume that only cell types with an estimated proportion greater than 10% in the deconvolution actually exist in a spot. Users also have the flexibility to modify this threshold as needed. We have assessed the impacts of these four options to Spotiphy's decomposition outputs. The results are presented in **Supplementary Figure 20c**. The results demonstrate that regardless of the chosen option, Spotiphy consistently delivers reliable decomposition results (iscRNA data).

Additionally, it's worth mentioning that obtaining images with a resolution of approximately 1 $\mu\text{m}/\text{pixel}$ is not a challenge in the current spatial transcriptomics technology. For example, Visium workflows utilize a 10X objective lens to scan the entire histological section and assemble the tiles into the final image, thereby providing the necessary resolution for cell segmentation. DBiT-seq, Slide-seq, and StereoSeq all required entire scanning images from 10X objective lens. For previously collected sections, AI-based image super-resolution algorithms may be helpful, which is a direction worth exploring.

Minor Concern 1: In Figure 1a, please clarify whether single-cell RNA-seq must be matched as an input component for each sample. If yes, this requirement will significantly limit Spotiphy's utility, as single-cell RNA-seq on FFPE samples is very challenging.

Response: We are grateful to the reviewer for highlighting the importance of the source of scRNA-seq references. Though the matched scRNA-seq datasets would be ideal, Spotiphy **does not require** that as discussed in line 411-420. In fact, in our evaluations with mouse brains (Fig 2-4), we used scRNA-seq data from the Allen Brain Atlas as a reference for all the neurons, astrocytes, and oligodendrocytes, rather than the matched scRNA-seq data. Spotiphy's deconvolution and decomposition results were still accurate and consistently outperformed other methods. This is primarily because Spotiphy selects marker genes during the deconvolution process, making it robust even when the scRNA-seq data does not come from matched samples. Spotiphy also assigns variables (batch prior) to each gene to model the differences between ST and scRNA-seq profiles, effectively mitigating batch effects on deconvolution and decomposition. In summary, as one of Spotiphy's strengths, it doesn't require scRNA-seq reference from matched samples, significantly broadening its potential applications.

Reviewer #3

Major comments:

1. In contrast to the main focus of this paper, which is the imputation of single-cell expression data from bulk spatial transcriptomics data, the authors' benchmarking analysis against previous methods focuses on deconvolution of cell type proportions. Unfortunately, this analysis falls quite short, both in the (i) breadth of datasets and tissue types analyzed (only one real dataset in this work versus the many available in prior literature [e.g., Li et al., 2022; <https://github.com/QuKunLab/SpatialBenchmarking>]) and (ii) the number of related computational methods assessed, several of which are inexplicably missing (including SpatialScope, despite it being mentioned/cited by the authors; also Redeconve). Given these critical omissions, it is impossible to judge whether Spotiphy truly represents an advance over the state-of-the-art.

Response: We greatly appreciate the reviewer's concerns regarding the insufficiency of datasets, tissue types, and benchmarking methods used for deconvolution evaluations in the initial manuscript. Reviewer #2 has also raised the same concern. Deconvolution is one of the focuses of Spotiphy, yet the initial manuscript did not sufficiently showcase the novelty and importance of Spotiphy's deconvolution function. We agree with both reviewers that including additional deconvolution methods for a comprehensive comparison can highlight the superiority of Spotiphy. In response, we have added eight additional methods for deconvolution benchmarking. More detailed descriptions can be found in response to Reviewer #2's concern 1.

We agree with the reviewer that including additional datasets from different tissue types can emphasize Spotiphy's advancements and generalizability. We appreciate the reviewer for sharing a recent benchmarking study and suggesting the use of their datasets for evaluations. This article utilized simulated pseudo-Visium data generated from paired scRNA-seq data to test the accuracy of deconvolution by comparing predicted data with the ground truth. We adopted this approach and applied eight datasets from human liver, lung, kidney, heart, and pancreas, as well as mouse kidney, pancreas, and trachea to evaluate deconvolution performance of Spotiphy against the other 12 methods (iStar not included due to the lack of images). The results are presented in **Supplementary Figure 10**. Spotiphy consistently delivers the best performance across all tissue types. These results highlight Spotiphy's capability to handle deconvolution across a diverse array of tissue types while maintaining its superiority. We also rephrased the descriptions in line 132-138 to emphasize the advantages of Spotiphy.

We would like to emphasize that the multi-platform datasets from mouse brains used in our initial manuscript were well-designed and have been useful not only for deconvolution benchmarking, but also for biological validations. (1) We built a scRNA-seq reference that covers a total of **27 different cell types** which have closely similar transcriptomic profiles at whole-genome coverage, far exceeding the total number of cell types used in other benchmarking studies (usually 5-15 types). This significantly elevates the complexity and difficulty of the deconvolution process. (2) We generated datasets that contain matched sequencing-based ST (featured with **whole-genome coverage**) and image-based ST (featured with single-cell resolution) datasets from adjacent tissue sections of wild-type and 5xFAD mouse brains. Therefore, we do not need to artificially generate simulated Visium data for deconvolution evaluation. In fact, the similarity between the Spotiphy’s results and the image-based ST data further supported that Spotiphy’s

outcomes not only exhibit high statistical accuracy but also reflect biological reality. The biological validity of Spotiphy's results is further validated by previous studies discussed in the initial manuscript.

2. Related to the above, the validation of *iscRNA-seq* expression data (i.e., artificial expression data produced by generative modeling) is inadequate. Inferring single-cell expression from bulk data is a highly ambitious, non-trivial task, and a key premise of this work (and a principal argument made against related methods, whether inferential [e.g., SpatialScope] or mapping-based [e.g., Tangram]) is that regional variation in the bulk ST dataset is best captured by Spotiphy's *iscRNA-seq* data. Is this true? Separately, the authors' use of simulated data (Extended Data Fig. 6e-f) to argue that inferred regional differences are not artefactual is not convincing, since the simulations are generated using real *scRNA-seq* reference data randomly assigned to spots based on estimated cell type proportions (which of course average over, and thus nullify, all spatial variation within any cell type from the real *scRNA-seq* dataset). What confidence do we have that real spatial variation is specifically attributed to the correct cell type by Spotiphy? For example, does Spotiphy infer regional variation for all cell types? Are similar (or even the same) spatially variant genes being assigned to multiple (or all) co-localized cell types? If not, are these genes uniformly partitioned across cell types? Even if such variation shows specificity for the correct cell type, how strong is this specificity and what is the level of bleed-through? If the authors calculate regional signatures for every cell type, do they match such differences, when detectable, for the matching cell type in orthogonal data (e.g., CosMx)? Overall, the authors need to be much more thorough and rigorous in evaluating the quality and validity of the *iscRNA-seq* transcriptomes, along with the cell-type-specificity of regional variation imputed from *iscRNA-seq* transcriptomes, both at the single-cell level and at the signature-level.

Response: We greatly appreciate the reviewer's acknowledgment of the inherent challenges in decomposition and the concerns regarding accuracy of *iscRNA* data generated by Spotiphy, especially in comparison to existing methods. Reviewer #2 has also raised the same concern. We agree with both reviewers that including decomposition benchmarking can further highlight the superiority of Spotiphy. In response, we have evaluated Spotiphy's decomposition performance against SpatialScope (DL-based), Tangram (mapping-based), and iStar (VT-based). The results are presented in **Supplementary Figure 14-15**. More detailed descriptions can be found in response to Reviewer #2's concern 2.

The reviewer raised concerns for the simulated data mentioned in Extended Data Fig. 4d-e. We agree with the reviewer that the simulation *randomly assigned* single cells to spot, which will average out the presence of SVGs. The simulated data is merely to support that when no SVGs exist (with simulated data), Spotiphy will not introduce any variation that stems from its modeling. In response, we have rephrased the text in line 184-193 for clarification.

The reviewer also raised concerns about the validity of regional differences identified by *iscRNA* data. We agree that verifying the validity of Spotiphy is a core aspect of our study. Therefore, we provided matched image-based ST datasets and performed additional experiments to provide independent evidence supporting the findings from *iscRNA* analysis. We also cited relative studies with widely accepted evidence to support our findings. For both mouse brain and human breast samples demonstrated in the initial manuscript, the regional sub-populations identified from *iscRNA* data were validated by matched image-based ST datasets (as the ground truth), they also exhibited consistent biological characteristics as reported in published studies discussed in line 175-182, 202-219, 227-233, 256-263, and 279-285. Our ultimate goal is to

systematically analyze and utilize iscRNA data generated by Spotiphy to generate new hypotheses, which can be further used to guide subsequent experimental validations, ultimately advancing the study of biological processes.

In addition, the reviewer raised concerns on the accuracy of Spotiphy in partitioning gene counts among different cell types. In response, we have revised the text and have provided detailed explanation of this process in SI line 224-286 to emphasize the novelty of Spotiphy's decomposition. In short, for each spot, the average expression of each gene across these cell types is calculated using scRNA-seq reference, and the counts of spot will then be partitioned based on the gene proportions of these cell types. Therefore, the iscRNA data reflects the real measurements of ST data and can be utilized for scRNA-seq analysis pipelines to uncover potentially valuable insights. This unique decomposition process represents a significant advancement of Spotiphy over SpatialScope and Tangram. Both methods aim to sample cells based on scRNA-seq reference (Tangram uses real cells from reference, SpatialScope generates pseudo cells from reference) and aggregate these samples to reconstruct the expressions of spatial spots. The sampled cells have to be sufficiently similar to the population of cells in the scRNA-seq reference, which significantly confines the gene expression range of each cell type. The limitations make it challenging to reflect the regional variations detected by ST data in their decomposition outputs. This is confirmed by the results shown in **Supplementary Figure 15**. While there is no promise that iscRNA data will 100% reflect the ground truth (real single-cell-resolved ST data), Spotiphy stands out as a promising approach given the current state of ST technologies and analysis pipelines. It maintains the whole-genome coverage and captures regional variations of ST data, thereby facilitating intriguing new biological insights as shown in the manuscript.

3. Instead of imputing iscRNA-seq data, why not infer cell-type-level expression data for every spot? As far as we can tell, there is no useful information in spot-level Visium ST data (prior to HD) that would allow realistic variation among cells of a given cell type to be inferred via generative modeling.

Response: We are grateful to the reviewer for this valuable point. We agree with the reviewer that inferring variation between cells of a given cell type in the same spot is challenging. Therefore, in our current modeling for decomposition (Extended Data Fig. 2a), we do not account for this difference and simply consider that the gene expression profile of cells of the same cell type from the same spot is identical. Specifically, we first partition the gene counts to each cell type and generate cell-type-level expression inferences for each spot. Based on the cell numbers obtained through segmentation and the cellular proportions obtained through deconvolution, we calculate the number of cells for each cell type. The cell-type-level expression profile is then divided by the numbers of this type obtained from previous segmentation and deconvolution steps, and all single-cell level expression profiles are normalized by CPM to obtain the iscRNA data output. In response, we have rephrased and provided a detailed description of this process in SI line 224-286 for clarification. We acknowledge that the current solution is not perfect. However, we would like to emphasize that it does not undermine the ability of iscRNA data to preserve and reflect SVG differences in the downstream analyses. Additionally, we are exploring new strategies to enhance our ability to predict differences between cells of the same type within the same spot by incorporating additional information, such as image features derived from deep learning-based vision transformers, as discussed in line 439-451.

4. Another potential gimmick is the use of Gaussian processes (or kernel smoothing) to impute iscRNA-seq data in non-capture areas. This is based on the authors' questionable assumption that "the proportion of each cell type changes smoothly over the entire tissue" (line 949). To prove that this assumption is sound and can usefully generalize to tissue types beyond those with highly stereotypical spatial structure (such as the brain), the authors would need to evaluate many tissue types with single-cell ST ground truth, ideally including neoplastic tissue where cell density and spatial organization can be highly variable.

Response: We greatly appreciate the reviewer's concerns regarding the authenticity and reliability of the assumption. We agree with the reviewer that more tissue types, especially neoplastic tissues are needed for a comprehensive testing for Spotiphy's imputation performance. 10x Genomics recently released Xenium datasets for multiple tumor types that also have Visium datasets publicly available. Although these datasets are not derived from adjacent tissue sections, they still provide valuable information. We selected lung and colorectal cancers samples and applied Spotiphy to the Visium datasets. The results are presented in **Supplementary Figure 16**. Despite the sections not being adjacent, we still find regions with similar histological structures in both Xenium and Visium datasets. Coloring by cell types, Spotiphy yields pseudo-single-cell-resolution images that are closely similar to Xenium data, indicating that Spotiphy's imputation function accurately restored the spatial distribution of various cell types for non-capture areas. These results demonstrated that even in highly heterogeneous tumor samples, Spotiphy could accurately impute the spatial distribution of different cell types. Additionally, InferCNV analysis using iscRNA data generated from lung and colorectal cancer samples successfully distinguished tumor cells from normal cells, further demonstrating the superiority of Spotiphy.

We agree that it is illogical to assume that the assumption holds true without establishing reasonable premises. It is worth noting that our initial description was not precise. In fact, the key assumption used in our current imputation model is that cellular proportions at nearby locations have a higher correlation, while those at distant points have a lower correlation, which is expected to hold in general. To support this assumption, we used three Xenium datasets of human lung cancer, colorectal cancer, and self-generated mouse brain for verification. Specifically, we take a random point on the Xenium image and draw a circle with a diameter of 50 μ m and calculate the cellular proportion as the starting point. We then move the point randomly in any direction and draw the circle again to calculate the cellular proportion and its similarity to

the starting data via correlation, and finally generate the result in **Supplementary Figure 17**. This observation demonstrated that within a reasonably small scale (below 100 μm), the correlation maintains at high level. It explained why the imputed cellular proportions of mouse brains (Figure 6) aligned well with Xenium data. To further evaluate the robustness of Spotiphy's imputation function, we overlay spot grids onto the Xenium image and remove non-capture areas. We then use Spotiphy to predict the distribution of these regions and generate the Correlation-Distance curves (yellow lines). The Spotiphy's prediction line aligns well with the real observed data line, indicating that the imputation function could reasonably reflect reality in both stereotypical homogeneous tissues, as well as highly heterogeneous neoplastic tissues.

Supplementary Figure 17. Imputation evaluations using Xenium datasets. a-c, Correlation between distance and cellular proportion of 50- μm spot in mouse brain (a), human lung cancer (b), human colorectal (c). d-f, Comparison between Xenium (the ground truth, left panel) and Spotiphy's imputation results (right panel) based on in-spot data of mouse brain (d), human lung cancer (e), human colorectal (f). Shadows in right panels represent spot location. Scale bar: 500 μm .

Furthermore, we cited multiple published articles, ranging from early histological studies to recent single-cell ST platforms (MERFISH), which support the commonly accepted consensus that cells of the same type typically exhibit a gradually uniform distribution within a tissue to maintain tissue function, which also support the assumption mentioned above. Moreover, research has shown that most genes mirror the steady and consistent spatial distribution, as do cells of the same type, particularly those involved in maintaining basic cellular functions and tissue architecture. Other popular algorithms such as CARD also employs the same assumption for its gene expression imputation. In sum, these evidences indicate that at least within small scale (100-200 μm), cellular proportion changes are minimal enough to neglect and therefore Gaussian processes or kernel smoothing could still be used for imputation. Though it is worth noting some genes may exhibit more variable spatial expression patterns, reflecting the tissue heterogeneity, and we cannot guarantee that this function is able to provide 100% accurate prediction in all circumstance, Spotiphy is the first promising approach to increase the information intensity of ST data. In response, we have carefully revised the text in line 320-328. Additionally, as mentioned in the Discussion, new strategies that leverage deep learning-based vision transformers to extract additional information from histological images may potentially achieve better imputation results.

5. Lines 280-282: The authors infer copy number aberrations in breast ST data using iscRNA-seq data produced by Spotiphy. How well do these copy number aberrations inferred from iscRNA-seq data coincide with copy number variation inferred directly from spatial spots?

Response: We greatly appreciate the reviewer's suggestion on comparing CNV aberrations derived from iscRNA-seq data with from the spot-level ST data. In response, we have conducted the inferCNV analysis using the original ST data in **Supplementary Figure 13** and revised the text in line 241-251. In short, the inferCNV results from spot-level ST data also managed to distinguish the tumor and normal samples, consistent with the results from iscRNA data, and the original studies. However, it is worth emphasizing that intra-heterogeneity of the CNV patterns is observed in the spot-level results, and there is no effective way to determine the source of these variations. In contrast, the inferCNV results from iscRNA data displayed that these variations came from sub-clusters of LumSec or LumHR cells. This result demonstrates once again that the iscRNA data from Spotiphy not only accurately reflects the reality from the original data but also provides additional valuable information. This highlights its potential for further mining insight from the ST data to uncover new biological discoveries.

6. The Xenium ground truth in Figure 2 is from an adjacent section. As the authors admit, this is not an ideal ground truth for cell type composition, and preferably this limitation would be made abundantly clear from the outset of this analysis. Why not create pseudo-Visium data from the Xenium data and/or CosMx data by aggregating neighboring cells at the spatial resolution of Visium?

Response: We greatly appreciate the reviewer's concerns regarding the limitations of using adjacent sections as the ground truth. We agree that these limitations should be clearly stated. In response, we have revised the text in line 126-129. We have generated simulated Visium data for additional evaluations as the reviewer suggested in the initial manuscript, but with some adaptations. Instead of aggregating Xenium or CosMx data, we used single-cell profiles to generate the simulated Visium datasets. The primary reason for this modification is that the Xenium data and CosMx data has pre-designed gene panels and limited number of genes in each panel (249 targets in Xenium and 951 targets in CosMx), which do not provide whole-genome coverage as the core of this paper. The use of simulated datasets derived from single-cell profiles also elevates the complexity and difficulty of the deconvolution process due to the higher number of genes involved, which better demonstrate the advances of Spotiphy. We would like to emphasize that the matched ST datasets from adjacent sections are not only used to obtain single-cell-resolved ST data that is mathematically close to the ground truth for performance evaluation, but also to cross-validate the new biological findings revealed by isCRNA data discussed in the initial manuscript.

7. In general, the hyperparameter and parameter values used in this work lack any empirical (or theoretical) rationale; instead, the authors simply state that the values they selected work well without providing any evidence or demonstrating robustness to other values.

Response: We greatly appreciate the reviewer for this insightful comment regarding the importance of the sensitivity analysis for Spotiphy. Reviewer #1 has also raised the same concern. In response, we have conducted sensitivity analyses on the settings of the hyperparameters. The results were summarized in **Supplementary Figure 20**. More detailed descriptions can be found in response to Reviewer #1's concern 3.

8. The authors calculate arithmetic means and standard deviations for performing gene-level z-tests in their marker gene selection method. How is this justified when scRNA-seq data are decidedly non-Gaussian?

Response: We thank the reviewer for this constructive question. We agree with the reviewer that Gaussian is not a popular assumption in scRNA-seq differential expression analysis compared with negative binomial or zero-inflated negative binomial or non-parametric approaches such as rank-sum tests (the default in Seurat, Scanpy, and other scRNA-seq analysis tools). We reason that this approach is still valid in our marker gene selection pipeline for the following reasons. (1) Large sample size: According to the central limit theorem, the sampling distribution of the mean will always be approximately normally distributed, as long as the sample size is large enough. In our scRNA-seq reference, each cell type contains at least 5,000 cells, considered quite large. For common scRNA-seq datasets, the number of cells per type is usually over 300 (~ 6,000 sequencing cells of ~20 cell types in one dataset), which meets the requirements. (2) Pairwise comparison instead of one-vs-rest in our marker gene selection: In one-vs-rest test, Gaussian is clearly invalid to model the expression in the rest population which is a mixture of different cell types but seems reasonable in our pairwise comparison (one vs. another cell type) to model the expression in each of the two cell types. (3) High precision: Our "z-test" is essentially a *Welch's t-test* allowing different variance and sample size in the two cell types. *Welch's t-test* has been demonstrated to have a high precision compared to other approaches in a previous benchmarking study of scRNA-seq differential expression (doi.org/10.1186/s12859-019-2599-6) when the departure from the normality assumption is not severe. In our marker gene selection for deconvolution, precision matters. (4) Computation speed: Rank-sum test is our alternative but suffers from more computational cost especially when the cell type number is high. In contrast, z-test is extremely fast, requiring less than two minutes to complete all necessary z-tests for >130,000 cells across 27 cell types. We have also revised the text in SI line 50-60 and added necessary description for clarification.

9. Extended Data Fig 3c: the scRNA-seq data in the UMAP appear to segregate by WT versus AD for all cell types, with at most modest intermixing. Does this imply that all cells are fundamentally different between healthy and AD? If so, this seems quite implausible and suggests that Spotiphy is highly susceptible to sample-specific batch variation. Given that Spotiphy is apparently applied independently to each ST sample, how would users be expected to distinguish between real and batch-specific variation across samples?

Response: We greatly appreciate the reviewer's concerns regarding the batch effects across different samples. We agree with the reviewer that it is crucial to shield users from batch effects. The modest

intermixing between WT and AD samples represents sample-derived batches, as cells of the same type are clustered together. In the initial manuscript, we used scMINER (doi.org/10.1101/2023.01.26.523391, <https://github.com/jyyulab/scMINER>), an in-house clustering algorithm, for constructing UMAPs of iscRNA data in Extended Data Fig 3c. Unfortunately, scMINER currently does not include a batch correction (sample integration) function. To test if current integration methods like Harmony can fix this issue, we have conducted clustering and UMAP plots using Seurat package with and without applying Harmony in **Supplementary Figure 11**. It is clear that before Harmony integration, the cells from WT and AD also showed a certain degree of separation in Seurat UMAP. After integration, the cells were perfectly intermixed in each cluster. Interestingly, the cells were segregated by cell sub-type, with intra-heterogeneity particularly observed in neurons, glia, and immune cells. This perfectly aligned with the results of scMINER. These findings indicate that the separation of cells in Extended Data Fig 3c is due to potential sample-derived batch effects, and users can apply current integration methods or harmonization functions like Harmony to effectively minimize this issue. More importantly, the regional specifications of cells that iscRNA revealed remain unaffected, demonstrating the validity of the discovery.

Removing the sample-derived batch effect is a very interesting topic for further exploration. Differentiating between sample-derived and technique-derived batches requires data replicates; however, this is beyond the scope of this study and can be a stand-alone topic.

Minor comments:

1. Regarding immune cells in the brain, the authors mischaracterize Spotiphy relative to other methods. For example, they claim that Spotiphy is uniquely able to localize neutrophils, however RCTD (Supplementary Fig 2e) appears comparable to Spotiphy and certainly not random, as the authors claim.

Regarding T/B cells, the authors state that only Spotiphy identified “extremely low proportions of these cells in its deconvolution results”. In contrast, in Extended Data Fig. 1a, it is clear that Spotiphy not only infers T cells as abundant at the spot level (up to 11%) but also infers them throughout nearly the entire specimen. Furthermore, many other methods infer T and/or B cells with similarly high abundance and/or broad distribution (Supplementary Fig 2). Therefore, it is perplexing why the authors single out CytoSPACE as a poor performer in this context, especially given that Spotiphy is clearly inferior for T cells; Cell2Location and CARD are clearly worse than CytoSPACE; and CytoSPACE relies on an external method to determine the global fraction of each cell type prior to mapping.

Response: We greatly appreciate the reviewer’s concerns regarding the description about Spotiphy’s deconvolution results compared to other methods. We agree that the text in the initial manuscript is not accurate and somewhat misleading. In response, we have revised the text in line 106-119. For neutrophil distribution, both RCTD and CIBERSORTx appeared comparable to Spotiphy. However, they incorrectly classified partial neutrophils as macrophages, resulting in an apparent increase in macrophages around the ventricle and a lower relative abundance of neutrophils (5%), compared to Spotiphy’s over 20%. Due to the low numbers of immune cells, and varying cell counts per spot, using proportions for rare cells can further lead to inaccuracies. Thus, we translated these proportions into absolute cell numbers (by multiplying with the total cell number per spot) for comparison. The results are presented in **Supplementary Figure 6**. For B cells, most methods performed well except CytoSPACE, StereoScope, iStar, and Cell2location. Spotiphy exhibited a low false-positive detection of T cells at the striatum (upper right region), its prediction for macrophage and microglia aligned well with the ground truth. In summary, with a total 13 methods included for deconvolution benchmarking, our evaluations indicate that Spotiphy demonstrated the best overall performance.

2. Line 964: The authors applied affine transformation to align Xenium and CosMx images. Please explain.

Response: We thank the reviewer for this valuable comment. Due to significant differences in the material processing and data generation processes of various ST platforms, even we used adjacent sections, the final images look different and cannot be properly aligned. To address this issue and make the ST data comparable, we used affine transformations to manually align the Xenium and CosMx images to the H&E staining image of Visium data. Taking Xenium as an example, we first scale the Xenium image to make sure each pixel in the image represents the same physical length as the H&E staining image of Visium data (0.753 microns/pixel). We then align the Xenium image with the H&E staining image by rotating and translating the Xenium image to ensure the regions of brain tissue (e.g., cerebral cortex, hippocampus, thalamus, hypothalamus, and striatum) are properly aligned. We have revised the text in line 919-924 to clarify this process.

Dear Dr. Strack-

We would like to express our sincere gratitude for the time and effort you and the reviewers have dedicated to our manuscript. In this round of revision, we have carefully addressed each point raised by Reviewers #1 and #3 and performed all the additional analyses they suggested. Revisions have been made to the manuscript (highlighted in red) to reflect these insights and suggestions. Below, we provide detailed, point-by-point responses and the necessary data to address each of the reviewers' comments and suggestions.

Reviewer #1

1. The authors may want to add discussion on the fact that the spatial coordinates of the generated iscRNA data were assigned randomly within the spot region, and therefore extra caution should be taken if users want to do analyses that depends on deterministic coordinates such as neighborhood analysis on the generated cells.

Response: We greatly appreciate the reviewer's valuable comment on the limitations of Spotiphy's cell type assignment. Indeed, this randomness poses constraints for users studying cell interactions and neighborhood analysis at a fine scale. In response, we have added a new paragraph in line 445-453 to specifically discuss the limitations of current approach, and the potential improvements and opportunities with the introduction of vision transformer.

2. It will be helpful for readers to be better aware of the usage of Spotify if the authors can add some information about the influence of hyper-parameters when presenting the methods and experiments, besides the discussions in the Discussion section.

Response: We greatly appreciate the reviewer's insightful suggestion. Indeed, we should have introduced the role of the hyper-parameters and their impacts on the results at the beginning of the paper when presenting our model. In response, we have added descriptions in line 71-72 to introduce the four hyper-parameters and cited the supplementary result (Supplementary Fig. 20) showing their impacts. We also included a note in the beginning of our online tutorial to help the users in tuning the hyper-parameters for better results.

3. We noticed a recent publication <https://www.nature.com/articles/s41592-024-02257-y> and an earlier one <https://www.nature.com/articles/s41467-022-34879-1> that might be of some relevance with the current work. They are not designed for the same task therefore there is no need to experimentally compare with them, but it could make the picture more complete if they are briefly mentioned and discussed in a proper place.

Response: We greatly appreciate the reviewer's insightful suggestion. In response, we have added a new paragraph in lines 480-486 to specifically discuss the advantages and potential of properly combining Spotiphy with these analytical methods in detail.

Reviewer #3

1. Validity of iscRNA data: The ability of Spotiphy to produce inferred single-cell RNA expression data (iscRNA) in spatial regions is a core feature that distinguishes it from most related methods. Unfortunately, the validity and robustness of this capability are not well-supported, and in fact, new analyses of human tumor samples added in this revision speak to the variably poor quality of the iscRNA data produced by Spotiphy.

Response: We appreciate the reviewer's recognition that the generation of iscRNA data is one of Spotiphy's core innovations compared to existing methods. We also acknowledge the reviewer's concerns about the validity of iscRNA data. As mentioned in our previous response, Spotiphy, like any other algorithm, cannot perfectly restore the ground truth 100% of the time. The accuracy of the output is dependent on multiple factors including the quality of the input data. If the input data is of poor quality, the algorithm cannot accurately infer and predict the missing data.

The reviewer also noted that predicting tumor samples is inherently more challenging than predicting ordinary samples, such as mouse brains. This difficulty arises partly from the poor quality of the data itself and partly from the limitations of existing methods in effectively and accurately performing deconvolution. As a new method, Spotiphy has repeatedly demonstrated superior results compared to all existing methods at the algorithmic level. As detailed in our previous responses to both Reviewer #2 and #3, we have conducted comprehensive validation of iscRNA data and benchmarking of Spotiphy against Tangram, SpatialScope, and iStar using **both simulated and real Visium datasets**, where Spotiphy has shown superior performance (**Supplementary Figures 14-15**). Furthermore, we have attempted to decompose spot expression to the single-cell level, which not only increases the density of expression data but also allows spatial data to seamlessly integrate with downstream scRNA-seq analysis tools. This enables researchers to extract more useful information from the original ST data. Based on this logic, we will address the reviewer's concerns one by one.

*a. New Supplementary Figure 16: The authors included new analyses of human CRC and lung tumor specimens to showcase the out-of-spot imputation ability of Spotiphy (more below). In doing so, they also include UMAPs and inferCNV plots of the corresponding iscRNA data produced by Spotiphy (S16b, c, e, f). From these data, we can draw several conclusions. First, distinct cell type transcriptomes in the iscRNA data are not separable in the UMAPs, indicating a blending of gene expression profiles and a *failure* of Spotiphy to reconstruct these profiles when applied to highly complex real-world tissue samples. This is obvious in Figure S16b at the nexus of several clusters (akin to a false branching process with what seems to be T/NK or pDCs at the center) and in Figure S16e, where many of the cell types are overlapping and blended (including what appears to be myeloid cells and lymphocytes).*

Response: The reviewer's first concern pertains to the UMAP results of human colorectal cancer and lung cancer samples we added in **Supplementary Figure 16**. We acknowledge that the current UMAP plots are not ideal, as some cells of different types are overlapping, and the separation is not clear. Reviewer #3 believes that the issue is due to Spotiphy's decomposition function not being accurate, making the generated iscRNA data untrustworthy. However, several other factors could be contributing to the problem. In response, we have analyzed these additional possibilities.

- 1) **Quality of Visium data.** As mentioned earlier, Spotiphy's results depend on the quality of the input data. We have conducted quality control (QC) and clustering analyses of Visium datasets for colorectal cancer, lung cancer, and mouse brain samples (Response Figure 1). Indeed, we found that the quality of these cancer samples was not as ideal as those of mouse brain tissue, as we observed heterogeneity in the original counts. Consequently, when using the spot-level expression data for clustering analysis, **a large number of spots tend to cluster together rather than forming distinct groupings with clear boundaries.**

- 2) **Clustering algorithm.** As noted by Reviewer #3, the mixed cell types are mostly immune cells (T/NK or pDCs in colorectal cancer, and myeloid cells and lymphocytes in the lung cancer) with few cell numbers and similar expression profiles. Our group has developed scMINER, a mutual information-based tool for single-cell clustering, which has been shown to outperform existing single-cell clustering algorithms. We have used scMINER to perform the clustering analysis of these scRNA data (Response Figure 2). Indeed, the results showed a significant improvement over the Seurat results, and therefore, we have replaced the clustering results in **Supplementary Figure 16**.

In further support of spurious lineage blending – and by the authors' own admission in the corresponding caption – fibroblast and endothelial cells in the CRC iscRNA data, and fibroblasts in the lung cancer iscRNA data, show imputed CNVs much like those inferred for the epithelial cells. It is perplexing why the authors don't consider this a red flag, but along with the blending observed above, these data strongly argue against the ability of Spotiphy to disentangle cellular heterogeneity within spots and assign gene expression profiles to the correct cell types in complex tissues. In contrast, the brain data upon which most of this work is based are considerably "easier" because many brain cell types cleanly separate into distinct and stereotypical spatial structures. Accordingly, it is hard to see how Spotiphy could be confidently applied to impute iscRNA data from diverse tissue types spanning health and disease.

Response: The reviewer's second concern pertains to the inferCNV results. We acknowledge that some normal cells (fibroblasts) displayed CNV patterns similar to tumor cells in the current inferCNV plots, which is illogical and has led Reviewer #3 to question the validity of the iscRNA data. In response, we have analyzed additional possibilities to address these concerns.

1) **Cell identity in iscrNA data.** We first generated feature plots of well-known marker genes of epithelial, endothelial, and fibroblast cells to ensure the cell identity in both iscrNA datasets are correct (Response Figure 3). Indeed, the clusters identified by the marker genes are consistent with the cell annotation generated by iscrNA data.

- 2) **Quality of scRNA-seq reference.** As mentioned earlier, Spotiphy's results are also dependent on the quality of the scRNA-seq reference. For colorectal cancer sample, we used scRNA-seq data from Ignasius et al., (*Nat. Gen.* 2022) as reference for decomposition. To determine the potential impact of scRNA-seq reference, we performed inferCNV analysis directly using the scRNA-seq data, and surprisingly, **the fibroblasts also displayed “tumor-like” CNV patterns** when using B, Plasma, and T/NK cells as normal cell references (Response Figure 4, the same setting used in Supplementary Figure 16). It is well-known that the output of inferCNV is highly sensitive to the choice of normal cell reference. The same cell type can exhibit different CNV patterns depending on the reference cells used. Therefore, we re-ran inferCNV on both the scRNA-seq reference and isCRNA data for the colorectal cancer sample, using fibroblasts

as the normal cell reference. Both the scRNA-seq reference and iscRNA data produced reasonable results.

scRNA-seq data from Peng et al., (*Cell* 2022) was used as reference for lung cancer sample decomposition. Similarly, when using B, NK, T & ILC, and other myeloid as normal cell references (Response Figure 5, the same setting used in Supplementary Figure 16), the **fibroblasts also displayed “tumor-like” CNV patterns**. When fibroblasts were used as the reference and the inferCNV analysis was re-run, both the scRNA-seq reference and iscRNA data showed reasonable results. These results strongly suggest that the “incorrect” CNV patterns from iscRNA data is attributed to the scRNA-seq reference, **instead of a *failure* of Spotify**. We therefore have decided to replace the panels in **Supplementary Figure 16**.

- 3) **Limitations of inferCNV and improvement from Spotiphy.** As mentioned in the original inferCNV article, distinguishing tumor cells from normal cells is **challenging**. By increasing the resolution of input data from spot level to single-cell level, Spotiphy enhance the accuracy and significantly improve the likelihood of correctly distinguishing between tumor and normal cells. In the case of the colorectal cancer, we successfully identified two groups of epithelial cells exhibiting normal and tumor CNV patterns.

b. Related to the above problem (which was less obvious in the original submission but is now very clear), the WT vs. AD brain examples highlighting region-specific iscRNA data within the same cell type (e.g., astrocytes) are not well-supported by validation data. This is because, while the authors do – to their credit – validate broad trends in region-specific expression programs using CosMx data, this is done in aggregate across many genes at once. How many of the differentially expressed genes in Figure 3d are truly

differentially expressed in astrocytes from CTX and TH? Indeed, the volcano plot in Fig 3c shows several genes with very significant differences that might be expected to easily validate (e.g., Agt, Mfge8, etc.). Given the blending observed above, it is entirely possible that such genes represent a mixture of region-specific DEGs arising from multiple co-localized cell types, some of which include astrocyte genes but many (or some) of which do not. Furthermore, the claim by the authors that such patterns are not present in the reference (Line 146) lacks any supporting figure in the manuscript. Would this be true using reference-guided annotation or label transfer from the iscRNA data? It seems implausible that none of the region-specific biology would be captured by real single-cell transcriptome atlases of the same brain area, especially when they are also observed in “oligodendrocytes and multiple types of neurons” within the iscRNA data (line 147).

Response: We fully understand that skepticism about the human cancer results has led Reviewer #3 to question our previous findings regarding mouse brains. We acknowledge that there were disorders in the arrangement of current figure panels, which may have caused confusion. In response, we have revised the current figure arrangements and generated additional plots to better support our conclusions.

- 1) **Supplementary information.** We have listed all relative information (including gene symbols and expression values) of DEGs in the **Supplementary Table S5 and S7**. We have edited the current text and supplementary tables to make it easier to access the information supporting our validations with CosMx data.
- 2) **Cell identity in iscRNA data.** We first generated feature plots of well-known marker genes of astrocyte and microglia to ensure the cell identity in iscRNA dataset are correct (Response Figure 6). Indeed, the clusters identified by the marker genes are consistent with the cell annotation generated by iscRNA data.
- 3) **DEGs expression patterns for astrocyte and microglia.** We have generated bar plots for each DEG used in calculating scores in **Figures 3-4** to clearly display their expression patterns in both the iscRNA data and CosMx data (Response Figure 7-8). It is worth noting that the CosMx data contains only 1000 genes in total, so we selected the available DEGs that are present in both datasets. We have also included other cell types located in the same spots as astrocytes and examined the expression of astrocyte DEGs across these cells (Response Figure 9a). Similar plots were generated for microglia (Response Figure 9b). To further determine if these DEGs also display unique expression patterns in the **scRNA-seq reference**, we also checked the DEGs expression patterns in the scRNA-seq reference. **These results ruled out the possibility that these DEGs are expressed by other cell types in the same location, strongly supporting the validity of the iscRNA data generated by Spotiphy**
- 4) Both astrocytes and microglia are primarily clustered together in the scRNA-seq reference (**Figure 1c**), whereas these cells form multiple distinct sub-clusters in the iscRNA data, as depicted in **Extended Data Figure 3b**. This supports our conclusion that “such patterns are not present in the reference (Line 146)”. We have cited the proper plots in the text and made the results clearer and easier to interpret.

Token together, we hope these results could address Reviewer #3’s concerns regarding reference-guided annotation or label transfer from the iscRNA data.

Response Figure 8: Expression patterns of each microglia DEG used in calculating scores for both the isCRNA data (left) and CosMx data (right).

It is completely understandable that Reviewer #3 has concerns that the region-specific biology described in the iscRNA data is not captured by other real single-cell transcriptome atlases. In response, we have included additional explanations in the Discussion section. First, we have provided evidence that the region-specific astrocytes identified by iscRNA data have been validated by published scRNA-seq data, in which different brain regions were dissected and scRNA-seq data were generated separately (**Extended Data Figure 4a-c**). Additionally, the DAM population identified by iscRNA data has also been well-characterized by other studies using scRNA-seq technology from multiple published papers. These results strongly support **the validity of iscRNA data**. Secondly, we believe that capturing region-specific biology through conventional single-cell transcriptome atlases is **challenging** for several key reasons: i) The single-cell sampling may not cover **enough regions of tissue**, where ST data usually have full coverage of the whole tissue block. ii) The population of cells, particularly rare cells, is often too small to detect these intra-variations. In comparison, Spotiphy can generate an average of about 10,000 cell expression profiles from a single tissue section (whereas single scRNA-seq sample typically has around 5,000 cells on average). This **high data volume-to-cost ratio** significantly enhances the potential for researchers to extract biologically meaningful information from ST data. iii) More importantly, even if sub-clusters of certain cell types are observed in single-cell transcriptome atlases, it is **impossible** to determine if they are regional variations without the spatial information unique to iscRNA data generated by Spotiphy. Together, these points further demonstrate **Spotiphy's superiority and necessity**.

2. *Out-of-spot imputation: The authors include several new analyses to support the assumptions incorporated into the out-of-spot imputation routine. In the contrived scenario where Spotiphy is applied to pseudo-spot data created from Xenium samples, the out-of-spot imputations are sufficiently convincing (new Supplementary Figure 17). Unfortunately, while these data nicely support the authors' central assumptions, they do not support the utility of this approach. In fact, in all cases where the authors apply this approach to real Visium data, the claim that Spotiphy is "highly consistent with ground truth" is dubious, perhaps owing to inaccuracies in the in-spot imputation.*

Response: We would first like to thank the reviewer for acknowledging that our imputation assumption is both reasonable and fully provable. This recognition indicates that Spotiphy's **unique** imputation function, along with its deconvolution and decomposition functions, can enhance data density for ST researchers compared to any existing method. This is highly significant in biological research, as even a small amount of extra information can greatly guide further wet lab validation. In this regard, **Spotiphy represents a substantial advancement for spatially resolved biological research.**

For example, in Figure 6b and Extended Data Figure 9, while some of the structural features seem qualitatively similar, entire L6 layers (red and dark red) are completely missed by Spotiphy, and in general, the clean depiction of layering in the Xenium data is absent with Spotiphy. By eyeballing the plots in Figure 6a and Extended Data Figure 9d, it appears that other methods, such as CTS, do a better job capturing these distinctions within-spot. This calls into question the fairness and relevance of the metrics in Figure 6c and d. Indeed, the reconstruction by Spotiphy in Extended Data Figure 9e is quite noisy. Separately, the new examples in Supplementary Figure 16 look almost nothing like the corresponding Xenium data. Thus, while the imputation ability may be reasonable in the setting of contrived data, its accuracy and utility are doubtful in real world settings.

Response: The reviewer's concerns about our imputation data centered on the distribution of L6 layers (including three cell types: L6 IT CTX, L6 CT CTX, and L6b CTX) not aligning closely with the ground truth in **Figure 6** and **Extended Data Figure 9**, leading the reviewer to question the validity of the imputation data. In response, we have provided the following detailed explanations and additional results:

- 1) **Highlight the pseudo image generation process.** As shown in **Extended Data Figure 9b**, we perform deconvolution and decomposition of the input ST data before the out-of-spot imputation (Response Figure 10). From the figure, we can see that Spotiphy's deconvolution performance remains strong, with the exception of L6 IT CTX. We then identify the locations of the nuclei from H&E images and convert the cell-type proportion absolute numbers. Finally, we assign cell type information to the detected nuclei and color-code them to generate the pseudo gap-free image. This process indicates that the **total number of nuclei/cells** detected from Visium image directly impacts the pseudo image results. As mentioned in the Discussion section, due to the sensitivity differences between DAPI and H&E staining, the absolute number of nuclei on Visium (48,219) is significantly smaller compared to Xenium (104,831) when we convert proportions into cell numbers. Consequently, the cell density for each cell type in the pseudo image is lower compared to the ground truth, resulting in a **lower number of L6 IT cells**, which raised the concerns. However, the distribution of L6 CT CTX and L6b CTX in the pseudo image remains very consistent with the ground truth, demonstrating the effectiveness of Spotiphy. Admittedly, as pointed out by Reviewer #3, we acknowledge that Spotiphy is not guaranteed to be superior to existing methods in every aspect, and indeed, no method can be. On this basis, we would like to highlight some factors that may affect the imputation performance.

- 2) **High difficulty of task.** As emphasized in the manuscript, we have assembled a comprehensive mouse brain scRNA-seq reference with **27 cell types**, including **10 cell types** have similar expression profiles to L6 layer cells. The deconvolution and decomposition across that many cell types goes far beyond the cases provided by other methods, presenting us with a very **challenging** problem. Even though the tissue used are what R3 refers to as “easier” mouse brains. We truly appreciate R3's acknowledgment of the difficulty of this task. Therefore, it is reasonable to expect some degree of underestimation or overestimation of **one or two cell types** within a **fine-scale region** of the entire section. In fact, when evaluating all 27 cell types together across the entire section, our accuracy remains significantly higher than that of the other two methods. We have provided all relevant information for calculating metrics in **Figure 6 and Extended Data Figure 9** in **Supplementary Table S17**.

- 3) **Randomness of assignment.** Although Spotiphy can accurately predict the number of various cell types in each spot, the lack of additional information prevents us from determining the specific cell type for each cell within a spot. As a result, we currently assign cell types to these locations randomly. In fact, none of the existing methods can solve this issue, and **Tangram also adopts a similar random scheme.** The introduction of randomness means that our pseudo image can guarantee high accuracy over a broad area (e.g., the whole section). However, when zooming into a fine-scale region, the image may appear somewhat noisy compared to the ground truth. We acknowledge that this is not the ideal solution. Therefore, using Vision Transformer to obtain additional information would be a good approach to make our assignments more accurate. However, that is not the focus of this article. As the **first** method to use an imputation scheme to increase ST data density (recovering information from non-capture areas), Spotiphy's imputation capability offers the community a significant new perspective and advances the entire field of research and progress.

3. Deconvolution evaluation against previous methods: The authors have done a nice job responding to related critiques and strengthening this aspect of the paper. Indeed, the deconvolution ability (i.e., imputing cell type fractions, not expression) is the strongest aspect of both Spotiphy and this paper. Unfortunately, this is also one of the areas where the competition is crowded and where the bar for notable improvement is high. The only real data used for benchmarking in this study are mouse brain tissue; the simulated tissues from pseudo-bulk transcriptomes are idealized and almost always much more trivially solved than real-world scenarios, where the technical variability and noise between the reference and the ST data are significant. To the authors' credit, they did simulate the addition of noise to simulated ST data in Extended Data Figure 2. While there is no intuition given for what "Noise level III" really means, it is notable that only two methods seem relatively immune to noise: TG and CTS, with TG performing considerably worse overall. We recommend that the authors apply this framework to all simulated data with suitably high noise and adjust their conclusions accordingly.

Response: We are grateful to the reviewer for recognizing Spotiphy's deconvolution capabilities as the best among all available methods. The reviewer expressed concern that we used only one real dataset in the benchmarking process. It should be noted that to perform benchmarking with real data, matched image-based ST data (serving as the ground truth) is required, preferably from the same or adjacent tissue slices. This is **one of the novelties** we would like to highlight in this study: **We are the first** to provide such a matched dataset to the ST community, which can be used to improve algorithm development. Due to the uniqueness of this matched dataset, we used a significant amount of simulated data for additional benchmarking, which is the most common approach used in other studies. In our last response, we added additional methods and datasets for comparison. The results showed that Spotiphy outperformed all other methods in every scenario.

Reviewer #3 expressed concerns about the noise levels. In our supplementary methods, we had detailed how to add noise to simulated data and provided the parameter values corresponding to the three levels of noise. Reviewer #3 suggested adding noise to the new data as well. We believe this is **unnecessary** for the following reasons.

- 1) **The purpose of adding noise** is to determine whether Spotiphy's output would be significantly affected by data quality. This has been thoroughly demonstrated in a comparison using mouse brains with three noise levels, as shown in **Extended Data Figure 2** and **Supplementary Figure 7-9**.
- 2) The additional simulated datasets come from a benchmarking article recommended by Reviewer #3, where all the simulated datasets have already been subjected to additional noises. Additionally,

samples from different tissues were used to test Spotiphy's stability and accuracy across different tissues, which was fully demonstrated in the **Supplementary Figure 10**. Therefore, we believe that the reviewer's suggestion of adding noise to all data does not provide additional validation of Spotiphy's performance superiority, and thus we respectfully do not consider this proposal reasonable or necessary.

We hope the reviewer will agree.

4. Slide tags extension (Extended Data Figure 10): It is unclear why the authors consider the iscRNA data in this application to be a valuable addition. Since Spotiphy is applied to individual cells in this case, the original Slide tags data would be preferred over imputed data (there is no deconvolution here and no latent data that would impart the iscRNA profiles with added value). The out-of-spot imputation makes for a pretty picture but nothing more. Given the issues with iscRNA data noted above, how close are the iscRNA data to paired Slide-tags cellular transcriptomes? Based on the UMAP in Extended Data Figure 10e, it seems not very close, as there is little overlap. Regardless, this analysis falls quite short of proving the claim that "Spotiphy overcomes the limitations of in situ single-nuclei sequencing" (line 343).

Response: We appreciate the reviewer's concerns about the extension of Slide tags. As we mentioned in the discussion, Slide tags represent an excellent technological breakthrough that truly obtains ST data with single-cell resolution and high gene coverage comparable to scRNA-seq data. However, its authors also acknowledged that the biggest limitation of Slide tags is that the current platform **cannot recognize all the nuclei/cells on a slice**, resulting in a significant loss of information. This was the specific limitation we referred to. We have made UMAP plot from Slide tags' original data and iscRNA data (Extended Data Figure 10d). It is clear that for CA3, Cortical, Endothelial, Inhibitory, and Microglia, there are less than 50 cells present in the dataset, making it very challenging to draw any meaningful conclusions.

We have demonstrated that with a few subtle adjustments to the Spotiphy imputation function, we could infer the expression data of nearby cells using Slide tags' snRNA-seq data as input. This greatly compensates for the weakness of Slide tags discussed above. Reviewer #3 expressed concerns about the similarity of iscRNA data to the original data. In response, we have added the following additional explanations and modified Extended Data Figure 10 accordingly.

- 1) To more clearly illustrate the origin of the cells in the UMAP plot, we generated a pairwise UMAPs to separately display the cells from the Slide tags' original data and Spotiphy-imputed data (Extended Data Figure 10d). Additionally, we included a table showing the number of each cell type in both datasets. This clearly demonstrates that Spotiphy can increase the number of each cell type, thereby enhancing the likelihood that researchers can obtain useful information from the data.
- 2) Extended Data Fig. 10f-g have shown that all the expression profiles of iscRNA data are very close to the original data. By increasing the number of these cells, Spotiphy has significantly enhanced the total amount of data from Slide tags. We acknowledge that these predictions are not always 100% accurate and may impact downstream analysis. We would like to emphasize that the primary role of Spotiphy here is to ensure the user can have enough data to analyze. The user will need to make their judgments on the results of any downstream analysis.

Minor comment:

1. Revisions to the text (in red) are poorly written with grammatical errors in many places.

Response: We thank the reviewer for carefully reviewing our revised manuscript. We will revise the additional text to correct and improve all the language errors mentioned by the reviewer.

Point-by-point responses to reviewers

Reviewer #1

1. The authors may want to add discussion on the fact that the spatial coordinates of the generated isCRNA data were assigned randomly within the spot region, and therefore extra caution should be taken if users want to do analyses that depends on deterministic coordinates such as neighborhood analysis on the generated cells.

Response: We greatly appreciate the reviewer's valuable comment on the limitations of Spotiphy's cell type assignment. Indeed, this randomness poses constraints for users studying cell interactions and neighborhood analysis at a fine scale. In response, we have added a new paragraph in line 448-460 to specifically discuss the limitations of current approach, and the potential improvements and opportunities with the introduction of vision transformer.

2. It will be helpful for readers to be better aware of the usage of Spotify if the authors can add some information about the influence of hyper-parameters when presenting the methods and experiments, besides the discussions in the Discussion section.

Response: We greatly appreciate the reviewer's insightful suggestion. Indeed, we should have introduced the role of the hyper-parameters and their impacts on the results at the beginning of the paper when presenting our model. In response, we have added descriptions in line 71-72 to introduce the four hyper-parameters and cited the supplementary result (**Supplementary Fig. 1**) showing their impacts. We also included a note in the beginning of our online tutorial to help the users in tuning the hyper-parameters for better results.

3. We noticed a recent publication <https://www.nature.com/articles/s41592-024-02257-y> and an earlier one <https://www.nature.com/articles/s41467-022-34879-1> that might be of some relevance with the current work. They are not designed for the same task therefore there is no need to experimentally compare with them, but it could make the picture more complete if they are briefly mentioned and discussed in a proper place.

Response: We greatly appreciate the reviewer's insightful suggestion. In response, we have added a new paragraph in lines 481-486 to specifically discuss the advantages and potential of properly combining Spotiphy with these analytical methods in detail.

Reviewer #3

1. Validity of iscRNA data: The ability of Spotiphy to produce inferred single-cell RNA expression data (iscRNA) in spatial regions is a core feature that distinguishes it from most related methods. Unfortunately, the validity and robustness of this capability are not well-supported, and in fact, new analyses of human tumor samples added in this revision speak to the variably poor quality of the iscRNA data produced by Spotiphy.

Response: We appreciate the reviewer's recognition that the generation of iscRNA data is one of Spotiphy's core innovations compared to existing methods. We also acknowledge the reviewer's concerns about the validity of iscRNA data. As mentioned in our previous response, Spotiphy, like any other algorithm, cannot perfectly restore the ground truth 100% of the time. The accuracy of the output is dependent on multiple factors including the quality of the input data. If the input data is of poor quality, the algorithm cannot accurately infer and predict the missing data.

The reviewer also noted that predicting tumor samples is inherently more challenging than predicting ordinary samples, such as mouse brains. This difficulty arises partly from the poor quality of the data itself and partly from the limitations of existing methods in effectively and accurately performing deconvolution. As a new method, Spotiphy has repeatedly demonstrated superior results compared to all existing methods at the algorithmic level. As detailed in our previous responses to both Reviewer #2 and #3, we have conducted comprehensive validation of iscRNA data and benchmarking of Spotiphy against Tangram, SpatialScope, and iStar using **both simulated and real Visium datasets**, where Spotiphy has shown superior performance (**Supplementary Figures 15-16**). Furthermore, we have attempted to decompose spot expression to the single-cell level, which not only increases the density of expression data but also allows spatial data to seamlessly integrate with downstream scRNA-seq analysis tools. This enables researchers to extract more useful information from the original ST data. Based on this logic, we will address the reviewer's concerns one by one.

*a. New Supplementary Figure 16: The authors included new analyses of human CRC and lung tumor specimens to showcase the out-of-spot imputation ability of Spotiphy (more below). In doing so, they also include UMAPs and inferCNV plots of the corresponding iscRNA data produced by Spotiphy (S16b, c, e, f). From these data, we can draw several conclusions. First, distinct cell type transcriptomes in the iscRNA data are not separable in the UMAPs, indicating a blending of gene expression profiles and a *failure* of Spotiphy to reconstruct these profiles when applied to highly complex real-world tissue samples. This is obvious in Figure S16b at the nexus of several clusters (akin to a false branching process with what seems to be T/NK or pDCs at the center) and in Figure S16e, where many of the cell types are overlapping and blended (including what appears to be myeloid cells and lymphocytes).*

Response: The reviewer's first concern pertains to the UMAP results of human colorectal cancer and lung cancer samples we added in **old Supplementary Figure 16**. We acknowledge that the current UMAP plots are not ideal, as some cells of different types are overlapping, and the separation is not clear. The reviewer believes that the issue is due to Spotiphy's decomposition function not being accurate, making the generated iscRNA data untrustworthy. However, several other factors could be contributing to the problem. In response, we have analyzed these additional possibilities.

- 1) **Quality of Visium data.** As mentioned earlier, Spotiphy's results depend on the quality of the input data. We have conducted quality control (QC) and clustering analyses of Visium datasets for colorectal cancer, lung cancer, and mouse brain samples (Response Figure 1). Indeed, we found that the quality of these cancer samples was not as ideal as those of mouse brain tissues, as we observed heterogeneity in the original UMI (nCount) and gene counts (nFeature). Consequently, when using the spot-level expression data for clustering analysis, spots in cancer samples tend to **cluster together** rather than forming distinct groupings with clear boundaries,

as seen in mouse brain samples. The poor quality of cancer samples is one of the factors that make deconvolution on cancer sample challenging.

2) **Clustering algorithm.** As noted by the reviewer, the mixed cell types are mostly immune cells (T/NK or pDCs in the colorectal cancer dataset, and myeloid cells and lymphocytes in the lung cancer dataset) with few cell numbers and similar expression profiles. Our group has developed **scMINER** (<https://github.com/jyyulab/scMINER>), a mutual information-based tool for single-cell clustering, which has been shown to outperform existing single-cell clustering algorithms.

We have used scMINER to perform the clustering analysis of these iscRNA data (Response Figure 2). Indeed, the results showed a significant improvement over the Seurat results, and therefore, we have replaced the clustering results in new Supplementary Figure 17.

In further support of spurious lineage blending – and by the authors’ own admission in the corresponding caption – fibroblast and endothelial cells in the CRC iscRNA data, and fibroblasts in the lung cancer iscRNA data, show imputed CNVs much like those inferred for the epithelial cells. It is perplexing why the authors don’t consider this a red flag, but along with the blending observed above, these data strongly argue against the ability of Spotiphy to disentangle cellular heterogeneity within spots and assign gene expression profiles to the correct cell types in complex tissues. In contrast, the brain data upon which most of this work is based are considerably “easier” because many brain cell types cleanly separate into distinct and stereotypical spatial structures. Accordingly, it is hard to see how Spotiphy could be confidently applied to impute iscRNA data from diverse tissue types spanning health and disease.

Response: The reviewer's second concern pertains to the inferCNV results. We acknowledge that some normal cells (e.g., fibroblasts, endothelial cells) displayed CNV patterns similar to tumor cells in the current

inferCNV plots, which is illogical and has led the reviewer to question the validity of the iscRNA data. In response, we have analyzed additional possibilities to address these concerns.

- 1) **Cell identity in iscRNA data.** We first generated feature plots of well-known marker genes of epithelial, endothelial, and fibroblast cells to ensure the cell identity in both iscRNA datasets are correct (Response Figure 3). As demonstrated, the clusters expressing marker genes are **consistent** with the annotations generated by iscRNA data.

2) **Quality of scRNA-seq reference.** As mentioned earlier, Spotiphy's results are also dependent on the quality of the scRNA-seq reference. For colorectal cancer sample, we used scRNA-seq data from Ignasius et al., (*Nat. Gen.* 2022) as reference for decomposition. Since the original paper **only** displayed inferCNV results of epithelial cells, to determine the potential impact of scRNA-seq reference, we performed inferCNV analysis directly using the scRNA-seq reference, and surprisingly, **both fibroblasts and endothelial cells displayed “tumor-like”**

CNV patterns when using **B, Plasma B, and T/NK cells** as normal cell references (Response Figure 4b). It is well-known that the output of inferCNV is highly **sensitive** to the choice of normal cell reference. The same cell type can exhibit different CNV patterns depending on the reference cells used. Therefore, we re-ran inferCNV on both the scRNA-seq reference and iscRNA data for the colorectal cancer dataset, including **fibroblasts and endothelial cells** as the normal cell reference. Both the scRNA-seq reference and iscRNA data produced reasonable results (Response Figure 4c, f).

The scRNA-seq data from Peng et al., (*Cell* 2022) was used as reference for lung cancer sample decomposition. Similarly, when using **B, NK, T & ILC, and other myeloid** as normal cell references (Response Figure 5b), the **fibroblasts also displayed “tumor-like” CNV patterns**. When fibroblasts were included in the normal reference and the inferCNV analysis was re-run,

both the scRNA-seq reference and isCRNA data showed reasonable results (Response Figure 5c, f).

These results strongly suggest that the “incorrect” CNV patterns from isCRNA data is attributed to the scRNA-seq reference and inferCNV algorithm, **instead of a *failure* of Spotiphy**. We therefore have decided to replace the inferCNV panels in **Supplementary Figure 17**.

3) **Limitations of inferCNV and improvement from Spotiphy**. As mentioned in the original inferCNV article, distinguishing tumor cells from normal cells is **challenging**. By increasing the resolution of input data from spot level to single-cell level, Spotiphy enhances the accuracy and significantly improves the likelihood of correctly distinguishing between tumor and normal cells. To prevent users from encountering similar issues and obtaining confusing results during

downstream analysis of iscRNA data generated by Spotiphy, we have added relevant notes in both **line 798-800 in the revised manuscript and the online tutorial of Spotiphy**. These additions aim to remind users of potential influences on downstream analysis caused by the scRNA reference used.

b. Related to the above problem (which was less obvious in the original submission but is now very clear), the WT vs. AD brain examples highlighting region-specific iscRNA data within the same cell type (e.g., astrocytes) are not well-supported by validation data. This is because, while the authors do – to their credit – validate broad trends in region-specific expression programs using CosMx data, this is done in aggregate across many genes at once. How many of the differentially expressed genes in Figure 3d are truly differentially expressed in astrocytes from CTX and TH? Indeed, the volcano plot in Fig 3c shows several genes with very significant differences that might be expected to easily validate (e.g., Agt, Mfge8, etc.). Given the blending observed above, it is entirely possible that such genes represent a mixture of region-specific DEGs arising from multiple co-localized cell types, some of which include astrocyte genes but many (or some) of which do not. Furthermore, the claim by the authors that such patterns are not present in the reference (Line 146) lacks any supporting figure in the manuscript. Would this be true using reference-guided annotation or label transfer from the iscRNA data? It seems implausible that none of the region-specific biology would be captured by real single-cell transcriptome atlases of the same brain area, especially when they are also observed in “oligodendrocytes and multiple types of neurons” within the iscRNA data (line 147).

Response: We fully understand that skepticism about the human cancer results has led the reviewer to question our previous findings regarding mouse brains. We acknowledge that there were disorders in the arrangement of current figure panels, which may have caused confusions. In response, we have generated additional plots to better support our conclusions and revised the figure panel citations in the text.

- 1) **Cell identity in iscRNA data.** We first generated feature plots of well-known marker genes of astrocyte and microglia to ensure the cell identity in iscRNA dataset are correct (Response Figure 6). Indeed, the clusters identified by the marker genes are consistent with the cell annotation generated by iscRNA data.
- 2) **Region-specific DEGs expression patterns in astrocytes and microglia.** We have listed all relative information (including gene symbols and expression values) of DEGs for astrocyte and microglia in the **Supplementary Table S5, S7, and S9**, respectively. We have also edited the current text and supplementary tables to make it easier to access the information supporting our validations with CosMx data. We then generated violin and bar plots for each DEG used in calculating scores in **Figures 3-4** to clearly display their expression patterns in both the iscRNA data and CosMx data (Response Figure 7-8). Since the CosMx data contains only 1000 genes in total, we selected the available DEGs that are present in both datasets. It is worth mentioning that only the AD sample showed two genes with opposite patterns in the CosMx data (*Slc4a4* and *Plcb4*), while the WT sample displayed the expected results. Since the WT and AD samples can be considered replicates in the case of astrocytes, with the proportions of each astrocyte subtype not differing substantially between the two samples, these results provide a more accurate reflection of reality.
- 3) **Region-specific DEGs expression patterns in co-localized cell types.** We have also included other cell types co-localized in the same spots as astrocytes (labeled as non-astrocyte) and generated box plots to examine the expression of astrocyte DEGs. Similar plots were generated using other cell types co-localized with microglia (labeled as non-microglia, Response Figure 9). As demonstrated, region-specific patterns similar to those observed in astrocytes and microglia **have not been detected** in other cell types.

- 4) **scRNA-seq reference re-annotation using label transfer from isCRNA data.** Following the reviewer's suggestion, we used the annotation of isCRNA data as the standard to re-annotate the scRNA-seq reference (Response Figure 10). For astrocyte, most of the cells (7,046 out of 7,106 cells) were labeled as “CTX”, with only 60 cells were labeled as “HPF” or “STR”, indicating that the astrocytes in the scRNA-seq reference were **homogeneous and lacked regionally specific subtypes**. The same was true for microglia, where cells were automatically labeled as either the “CTX&HPF” (5,120 cells) or “FT” (1,709 cells) subtypes and were mixed together in a relatively uniform manner. These results support our conclusion that “such patterns are not

present in the reference” in **line 146-148**. Additionally, we have cited the relative plots in the text and made the results clearer and easier to interpret. Both astrocytes and microglia are primarily clustered together in the scRNA-seq reference (Figure 1c), whereas these cells form multiple distinct sub-clusters in the iscRNA data (Extended Data Figure 3b). These results strongly indicate that the region-specific subtypes identified in iscRNA data are not detectable in the scRNA-seq reference.

Token together, these results ruled out the possibility that **these DEGs are expressed by other co-localized cell types**, strongly supporting the validity of iscRNA data generated by Spotiphy. We hope these results could address the reviewer’s concerns.

It is completely understandable that the reviewer has concerns that the region-specific biology described in the iscRNA data is not captured by other real single-cell transcriptome atlases. In response, we have included additional discussion in **line 391-399**. First, we have provided evidence that the region-specific astrocytes identified by iscRNA data have been validated by published scRNA-seq data, in which different brain regions were dissected and scRNA-seq data were generated separately (**Extended Data Figure 4a-c**). Additionally, the DAM population identified by iscRNA data has also been well-characterized by other studies using scRNA-seq technology from multiple publications. These results strongly support **the validity of iscRNA data**. Secondly, we believe that capturing region-specific biology through conventional single-cell transcriptome atlases is **challenging** for several key reasons: i) The single-cell sampling may not cover **enough regions of tissue**, where ST data usually has full coverage of the whole tissue block. ii) The population of cells, particularly rare types of cells, is often too small to detect these intra-variations. In comparison, Spotiphy can generate an average of about 10,000 cell expression profiles from a single tissue section (whereas one scRNA-seq sample typically has around 5,000 cells on average). This **high data volume-to-cost ratio** significantly enhances the potential for researchers to extract biologically meaningful information from ST data. iii) More importantly, even if sub-clusters of certain cell types are observed in single-cell transcriptome atlases, it is **impossible** to determine if they are regional variations without the spatial information unique to iscRNA data generated by Spotiphy. Together, these points further demonstrate **Spotiphy's superiority and necessity**.

2. *Out-of-spot imputation: The authors include several new analyses to support the assumptions incorporated into the out-of-spot imputation routine. In the contrived scenario where Spotiphy is applied to pseudo-spot data created from Xenium samples, the out-of-spot imputations are sufficiently convincing (new Supplementary Figure 17). Unfortunately, while these data nicely support the authors' central assumptions, they do not support the utility of this approach. In fact, in all cases where the authors apply this approach to real Visium data, the claim that Spotiphy is "highly consistent with ground truth" is dubious, perhaps owing to inaccuracies in the in-spot imputation.*

Response: We would first like to thank the reviewer for acknowledging that our imputation assumption is both reasonable and fully provable. This recognition indicates that Spotiphy's **unique** imputation function, along with its deconvolution and decomposition functions, can enhance data density for ST researchers compared to any existing method. This is highly significant in biological research, as even a small amount of extra information can greatly guide further wet lab validation. In this regard, **Spotiphy represents a substantial advancement for spatially resolved biological research.**

For example, in Figure 6b and Extended Data Figure 9, while some of the structural features seem qualitatively similar, entire L6 layers (red and dark red) are completely missed by Spotiphy, and in general, the clean depiction of layering in the Xenium data is absent with Spotiphy. By eyeballing the plots in Figure 6a and Extended Data Figure 9d, it appears that other methods, such as CTS, do a better job capturing these distinctions within-spot. This calls into question the fairness and relevance of the metrics in Figure 6c and d. Indeed, the reconstruction by Spotiphy in Extended Data Figure 9e is quite noisy. Separately, the new examples in Supplementary Figure 16 look almost nothing like the corresponding Xenium data. Thus, while the imputation ability may be reasonable in the setting of contrived data, its accuracy and utility are doubtful in real world settings.

Response: The reviewer's concerns about our imputation data centered on the distribution of L6 layers (including three cell types: L6 IT CTX, L6 CT CTX, and L6b CTX) not aligning closely with the ground truth in **Figure 6** and **Extended Data Figure 9**, leading the reviewer to question the validity of the imputation data. In response, we have provided the following detailed explanations and necessary results to address factors that may affect the imputation performance:

- 1) **High difficulty of task.** As emphasized in the manuscript, we have assembled a comprehensive mouse brain scRNA-seq reference with **27 cell types**, including **10 cell types** with similar expression profiles to L6 layer cells. The deconvolution across such a large number of cell types presents a very **challenging** problem, far beyond the cases provided by other methods, even though the tissue used is what the reviewer refers to as "easier" mouse brains. We truly appreciate the reviewer's acknowledgment of the difficulty of this task. Admittedly, as pointed out by the reviewer, we acknowledge that Spotiphy is not guaranteed to be superior to existing methods in every aspect, and indeed, no method can be. Therefore, it is reasonable to expect some degree of underestimation or overestimation of **one or two cell types** within a **fine-scale region** of the entire section (Response Figure 11). As demonstrated, Spotiphy underestimates L6 IT CTX proportion, but remains accurate for L6 CT CTX and L6b CTX proportions. In fact, when evaluating all 27 cell types together, whether across a fine-scale region or the entire section (Figure 6c-d), Spotiphy's accuracy consistently remains significantly higher than that of the other two methods. We have provided all relevant information for calculating metrics in **Supplementary Table S17**.
- 2) **Pseudo image generation process.** As shown in **Extended Data Figure 9b**, out-of-spot cell-type proportion imputation relies on in-spot cell-type deconvolution outputs. Based on these proportions, we then identified the locations of the nuclei from H&E images and converted the cell-type proportion to absolute numbers (Response Figure 11). Finally, we assigned cell type information to the detected nuclei and color-coded them to generate the pseudo gap-free image. This process

indicates that the **total number of nuclei/cells** detected from Visium image directly impacts the pseudo image results. As mentioned in line 955-957, due to the sensitivity differences between DAPI and H&E staining, the absolute number of nuclei on Visium (48,219) is significantly smaller compared to Xenium (104,831) when we convert proportions into cell numbers. Consequently, the cell density for each cell type in the pseudo image is lower compared to the ground truth, resulting in **a lower number of L6 IT cells**, which raised the concerns. By contrast, both distribution and cell number of L6 CT CTX and L6b CTX in the pseudo image remains very consistent with the ground truth, demonstrating the effectiveness of Spotiphy.

- 3) **Randomness of assignment.** Although Spotiphy can accurately predict the number of each cell type in each spot, the lack of additional information prevents us from aligning each detected nucleus/cell to the specific cell type within a spot. Therefore, we currently assign annotations to these nuclei/cells randomly (Extended Data Figure 9b). In fact, **Tangram** also adopts a similar random approach for its mapping function, and none of the existing methods can solve this issue. The introduction of randomness means that our pseudo image can guarantee high accuracy over a broad area (e.g., the whole section). However, when zooming into a fine-scale region, the image may appear somewhat noisy compared to the ground truth. We acknowledge that this is not the ideal solution. Therefore, using deep learning-based vision transformer (VT) to obtain additional information from images would be a good approach to make our assignments more accurate. However, that is not the focus of this article. It is worth mentioning that it is generally accepted that the domain within which a solitary cell can effectively communicate is approximately 250 μm in size (doi: 10.1073/pnas.94.23.12258). The Visium spot size is 50 μm , which falls within this cell communication range. This minimizes the impact of random distribution within the spot on the downstream analysis of cell-cell communication. Taken together, as the **first** method to use an imputation approach to increase ST data density (recovering information from non-capture areas), Spotiphy's imputation capability offers the community a significant new perspective and advances the entire field of research and progress.
- 4) **Unmatched human cancer datasets.** As we mentioned in the last response, both Visium and Xenium datasets for the human colorectal cancer and lung cancer samples were from the 10x Genomics public database. Unfortunately, they **do not** come from the same tissue blocks, therefore the overall structures of the sections are inherently different (Response Figure 12). We identified and picked the regions with similar pathological structures based on the H&E staining images and compared the Xenium results with Spotiphy-derived pseudo images (**old Supplementary Figure 16a, d**). The main purpose of these panels was to demonstrate that the cell-type proportions and distribution patterns in Spotiphy-derived pseudo images are comparable to the ground truth (Xenium). It became clear to us that these panels were not achieving the desired effect and were causing unnecessary confusion. Therefore, we have decided to **remove these panels** in the **new supplementary Figure 17**.

3. *Deconvolution evaluation against previous methods: The authors have done a nice job responding to related critiques and strengthening this aspect of the paper. Indeed, the deconvolution ability (i.e., imputing cell type fractions, not expression) is the strongest aspect of both Spotiphy and this paper. Unfortunately, this is also one of the areas where the competition is crowded and where the bar for notable improvement is high. The only real data used for benchmarking in this study are mouse brain tissue; the simulated tissues from pseudo-bulk transcriptomes are idealized and almost always much more trivially solved than real-world scenarios, where the technical variability and noise between the reference and the ST data are significant. To the authors' credit, they did simulate the addition of noise to simulated ST data in Extended Data Figure 2. While there is no intuition given for what "Noise level III" really means, it is notable that only two methods seem relatively immune to noise: TG and CTS, with TG performing considerably worse overall. We recommend that the authors apply this framework to all simulated data with suitably high noise and adjust their conclusions accordingly.*

Response: We are grateful to the reviewer for recognizing Spotiphy's deconvolution capabilities as **the best** among all available methods. The reviewer expressed concern that we used only one real dataset in the benchmarking process. It should be noted that to perform benchmarking with real data, matched image-based ST data (serving as the ground truth) is required, preferably from the same or adjacent tissue slices. This is **one of the novelties** we would like to highlight in this study: **we are the first** to provide such a matched dataset to the ST community, which can be used to improve algorithm development. Due to the uniqueness of this matched dataset, we used a significant amount of simulated data for additional benchmarking, which is the most common approach used in other studies. In our last response, we added additional methods and datasets for comparison. The results showed that Spotiphy outperformed all other methods in every scenario.

The reviewer expressed concerns about the noise levels. In our supplementary methods, we had detailed how to add noise to simulated data and provided the parameter values corresponding to the three levels of noise. The reviewer suggested adding noise to the new data as well. We believe this is **unnecessary** for the following reasons.

- 1) **The purpose of adding noise** is to determine whether Spotiphy's output would be significantly affected by data quality. This has been thoroughly demonstrated in a comparison using mouse brains with three noise levels, as shown in **Extended Data Figure 2** and **Supplementary Figure 7-9**.
- 2) The additional simulated datasets come from a benchmarking article recommended by the reviewer, where all the simulated datasets have already been subjected to additional noises. Additionally, samples from different tissues were used to test Spotiphy's stability and accuracy across different tissues, which was fully demonstrated in the **Supplementary Figure 10**. Therefore, we believe that the reviewer's suggestion of adding noise to all data does not provide additional validation of Spotiphy's performance superiority, and thus we respectfully do not consider this proposal reasonable or necessary.

We hope the reviewer will agree.

4. Slide tags extension (Extended Data Figure 10): It is unclear why the authors consider the iscRNA data in this application to be a valuable addition. Since Spotiphy is applied to individual cells in this case, the original Slide tags data would be preferred over imputed data (there is no deconvolution here and no latent data that would impart the iscRNA profiles with added value). The out-of-spot imputation makes for a pretty picture but nothing more. Given the issues with iscRNA data noted above, how close are the iscRNA data to paired Slide-tags cellular transcriptomes? Based on the UMAP in Extended Data Figure 10e, it seems not very close, as there is little overlap. Regardless, this analysis falls quite short of proving the claim that "Spotiphy overcomes the limitations of in situ single-nuclei sequencing" (line 343).

Response: We appreciate the reviewer's concerns about the extension of Slide-tags. As we mentioned in line 345-348, Slide-tags represent an excellent technological breakthrough that truly obtains ST data with single-cell resolution and high gene coverage comparable to scRNA-seq data. However, its authors also acknowledged that the biggest limitation of Slide tags is that the current platform **cannot recognize all the nuclei/cells on a slice**, resulting in a significant loss of information. This was the specific limitation we referred to. Specifically, it is clear that for CA3, Cortical, Endothelial, Inhibitory, and Microglia, there are less than 50 cells present in the original Slide-tags dataset, making it very **challenging** to draw any meaningful conclusions. Spotiphy's imputation function enable us (in SI line 281-286) to predict the expression profiles (iscRNA data) of the non-targeted nuclei nearby the nuclei directly using Slide-tags' snRNA-seq data as input. Therefore, this process **does not** involve any deconvolution or decomposition

from Spotiphy. We have re-generated UMAP plot from Slide-tags' original data and iscRNA data using scMINER to have a clearer clustering visualization (Extended Data Figure 10d). The additional cell numbers greatly compensate for the weakness of Slide tags data discussed above. The reviewer expressed concerns about the similarity of iscRNA data to the original data. In response, we have added the following additional explanations and modified Extended Data Figure 10 accordingly in line 711-713.

- 1) To more clearly illustrate the origin of the cells in the UMAP plot, we generated a pairwise UMAPs to separately display the cells from the Slide-tags' original data and Spotiphy-inferred data (Extended Data Figure 10d). It is worth noting that the shape of the UMAP is somewhat unusual. By comparing it with the UMAP from the original article (Response Figure 13), we found that both UMAPs exhibit similarly unusual shapes for the same cell types. Therefore, we believe these visualization patterns were primarily due to the quality of the Slide-tags' data, rather than any artifact introduced by Spotiphy imputation. Additionally, we included a table showing the number of each cell type in both datasets. This clearly demonstrates that Spotiphy can increase the number of each cell type, thereby enhancing the likelihood that researchers can obtain useful information from the data.
- 2) To better demonstrate that the Spotiphy-inferred data closely matches the Slide-tags' original data, we used heatmaps to display the expression patterns of the marker gene sets from the original article in both datasets (Extended Data Figure 10f). The consistent pattern strongly indicates that the expression profiles of iscRNA data are very close to the original data.
- 3) We further examined the proportion of cells in both datasets (Extended Data Figure 10g). The similar cell ratios suggest that imputation does not alter the cell composition in the hippocampus of mouse brain and **does not** introduce any artifact. Additionally, by increasing the number of each cell type, Spotiphy has significantly enhanced the data density from Slide-tags. We acknowledge that these predictions are not always 100% accurate and may impact downstream analysis. We would like to emphasize that the primary role of Spotiphy here is to ensure the user can have enough data to analyze. The user will need to make their judgments on the results of any downstream analysis.

Extended Data Figure 10

Extended Data Fig. 10. Spotiphy's imputation to loss nuclei increases information density of Slide-tags data.

Minor comment:

1. Revisions to the text (in red) are poorly written with grammatical errors in many places.

Response: We thank the reviewer for carefully reviewing our manuscript. We have revised the additional texts and corrected all the language errors mentioned by the reviewer.

Point by Point Response to Reviewers (Round 3)

Reviewer #3:

*The authors have adequately addressed some of our comments from the previous round of review. However, several concerns remain, as detailed below. Moreover, as part of their rebuttal, they have included loaded language such as “...these points further demonstrate Spotiphy’s superiority and *necessity*”. Such language is unhelpful, as it portrays the authors in an unfavorable light and implies that they are incapable of being impartial about their own approach. Indeed, in multiple instances where the Spotiphy output is questionable or implausible, instead of admitting it (and adding useful text/analyses to the manuscript to properly and fairly qualify Spotiphy’s scope and utility), they have shifted the blame without convincing rationale (more below). For all these reasons, and considering the bold claims about Spotiphy’s many capabilities, we remain skeptical about Spotiphy’s added value for the field, with the iscRNA data output of particular concern.*

Response: We appreciate the reviewer for acknowledging our responses to some of the comments from previous round of review. Yet, some concerns remain, including follow-up questions to our previous responses and misunderstandings due to our unclear explanations. Below, we summarize the reviewer’s remaining comments and address them in more detail.

1. We agree that the response should concentrate on detailing the supporting evidence rather than merely highlighting Spotiphy's superiority. Our intention was not to overemphasize Spotiphy's strengths, but simply to summarize our responses. This does not prevent us from maintaining an unbiased perspective when evaluating Spotiphy vs. all the other approaches. We have removed these line as per the reviewer’s suggestion.
2. The reviewer’s primary concern clearly centers on the quality of iscRNA data. The reviewer stated: *“The ability of Spotiphy to produce inferred single-cell RNA expression data (iscRNA) in spatial regions is a core feature that distinguishes it from most related methods”*. The iscRNA data indeed represents one of the novelties of our approach. The reviewer believes that our previous response did not adequately address Spotiphy's issues and instead offered various untenable excuses. We agree that it is inappropriate to make excuses without directly addressing the problem. It is unfortunate that our previous response gave the reviewer the wrong impression, as that was not our intention. Although we are unsure of the exact reason that biased the reviewer's interpretation, it may have resulted from our improper writing habits, style, or an inaccurate choice of words or terms. Regardless, we want to emphasize that we deeply regret any misunderstanding we may have caused. Our intention was simply to provide all possible explanations for the reviewer’s previous comments. We summarize all iscRNA data related issues raised by the reviewer over the three rounds and attempt to address them comprehensively in the following response to avoid any further confusion. Additionally, we have never shied away from acknowledging the shortcomings and issues of the current Spotiphy function. We thoroughly discussed the current drawbacks of Spotiphy and the exploration of future directions for improvement in the updated manuscript, aiming to enhance our approach and gradually achieve the ideal condition of approximating real scRNA-seq data as envisioned by the reviewer.
3. The reviewer’s second concern is deconvolution. We apologize for the misunderstanding; the reviewer did not explicitly state that Spotiphy is the best but rather noted that *“the deconvolution ability is the strongest aspect of both Spotiphy and this paper”*. Nonetheless, we are primarily grateful that our work and efforts have been recognized by the reviewer to the extent that we are considered at least among the top, if not the best. We have provided additional benchmarking analysis of external datasets with added noise as suggested by the reviewer. Additionally, we would

like to highlight that Spotiphy has a slight advantage over RCTD in terms of clearer and more accurate identification of rare cell populations, such as neutrophils and macrophages (Fig. 2 and Supplementary Fig. 4).

4. The reviewer's final concern is that Spotiphy only brings little improvements to Slide-tags. We would like to clarify that in our previous response, we explicitly stated at the beginning that Spotiphy does not offer any technical advancements for this approach, and the "improvement" we mentioned is merely a minor attempt at utilizing Spotiphy's imputation function, as it is not the main novelty and feature of Spotiphy that we have highlighted such as iscRNA data and deconvolution. The "improvement" we intended to convey is that while Slide-tags data is valuable in our view, the small number of nuclei makes it challenging for unsupervised downstream analyses such as differential expression (DE) analysis and cellular composition of a given region due to insufficient statistical powers. Spotiphy's imputation function can be used to generate more pseudo single-nuclei profiles for analysis. From this perspective, what the reviewer stated, "*there are no 'new' expression data (such as spot transcriptomes) to learn novel variation*" is actually in line with our expectations. Admittedly, as the reviewer points out, these additional profiles are inferred and may raise issues like unreliable downstream analysis results, which we have acknowledged and discussed as a focus for future research in the updated manuscript. By improving the accuracy of the inferred profiles in the future, we could enhance the potential value of this application and provide extra insights to the whole ST community.

We hope this summary helps the reviewer understand our arguments more clearly and reasonably.

Major comments:

1. The authors' responses to our critiques about their cancer analyses (human CRC and lung tumor) contain numerous flaws, exposing (or at least strongly suggesting) limited familiarity with single-cell and spatial tumor expression data and analysis.

Response: We noticed that the reviewer has consistently expressed doubts about the iscRNA data generated by Spotiphy in three rounds of comments. In the first round, the reviewer overlooked the new findings from our iscRNA data in mouse brain tissues, which were not reported by traditional scRNA-seq techniques and were further validated by other wet lab-based experiments (Fig. 3-4). The reviewer diminished the value of our work by emphasizing "*the brain data upon which most of this work is based are considerably 'easier' because many brain cell types cleanly separate into distinct and stereotypical spatial structures*". Nevertheless, our method is, if not the first, currently the best approach for offering iscRNA data, especially compared to the unsatisfactory results of Tangram and SpatialScope (Supplementary Fig. 15-16).

To properly and fairly qualify Spotiphy's scope and utility, we provided performance evaluations of Spotiphy using additional cancer tissues as suggested by the reviewer. We included two datasets from human lung and CRC tumors, primarily because they have scRNA-seq, Visium, and Xenium data publicly available, which helps address other comments from the reviewer. These two datasets successfully addressed the reviewer's concerns about our imputation arguments for non-capture areas.

Regarding the evaluations of iscRNA data, we indeed provided "incorrect" plots in our first response, which raised the reviewer's key concerns in the second round: "*new analyses of human tumor samples added in this revision speak to the variably poor quality of the iscRNA data produced by Spotiphy*". Specifically, the reviewer noted, "*First, distinct cell type transcriptomes in the iscRNA data are not separable in the UMAPs*". Indeed, as the reviewer has pointed out, cells in close proximity on UMAP tend to represent similar profiles, and the inability of Spotiphy to accurately disassembly the gene expression to single-cell level is one of the possibilities. In our last response, we have acknowledged this possibility as drawbacks

of Spotiphy. We have never claimed that current version of Spotiphy is perfect for any tissue or that it provides flawless scRNA data. We have consistently emphasized its limitations and have extensively discussed how we plan to address these issues in our future work to make it more accurate and effective. Specifically, our intention was merely to list other potential factors that may contribute to this phenomenon for a comprehensive discussion and to address these concerns in detail. Unfortunately, the reviewer perceived these efforts as “*shifted the blame without convincing rationale*”. While we once again express our sincere apologies for any misunderstanding, we address each point comprehensively in the following to ensure that our meaning is correctly conveyed.

a. Quality of the Visium data (Response Figure 1). The authors claim that the quality of the cancer Visium data is suspicious owing to their observation that “spots in cancer samples tend to cluster together rather than forming distinct groupings with clear boundaries, as seen in mouse brain samples.” While this is one possibility, a much more likely explanation is that tumor samples have cells of different types with highly variable local density and context-dependent transcriptional states. Consequently, instead of yielding discrete clusters indicative of different cell types or regions (as seen in mouse brain, and one of the reasons we claim mouse brain tissues are idealized for ST analysis methods [and thus “easier”]), spot-based tumor data are expected to blend more readily in a low dimensional embedding. This of course complicates and likely confounds the analysis of tumors (with methods such as Spotiphy) but does not imply lower quality per se. The authors also complain about the heterogeneity of UMIs and gene counts, implying this heterogeneity is an indicator of poor quality. Rather, it is generally an indicator of differences in cell size/RNA content, metabolic activity, and tumor versus adjacent stromal regions.

Response: Similar to the clustering results of scRNA-seq data, nearby spots in UMAP plot are also considered to have similar expression profiles at spot-level. As the reviewer pointed out, the number of cells in each spot, the proportion of different cell types, and the expression profile of each cell type are highly context-dependent, leading to such outcomes. This is the main reason we proposed the concept of region-specific cell subtypes identified by scRNA data analysis (Fig. 3-5). We generally agree with the reviewer that “*instead of yielding discrete clusters indicative of different cell types or regions, spot-based tumor data are expected to blend more readily in a low dimensional embedding*”, as this phenomenon is apparent in many tumor Visium datasets. However, data exhibiting this phenomenon do not necessarily display heterogeneity or significant variations in QC metrics such as UMIs and gene counts. We randomly selected additional six Visium datasets (S1 to S6) of colorectal cancer from a recent paper by Alberto Valdeolivas et al. (<https://doi.org/10.1038/s41698-023-00488-4>) and conducted the basic QC and clustering analysis using Seurat. Although the spots in each slide tended to cluster together in UMAP plot, the QC metrics did not reveal the same degrees of variation observed in former lung and CRC datasets (**Response Figure 1a-b**). Similar patterns were observed in three breast cancer samples we used in the manuscript: Spots from same sample tended to cluster together in UMAP plot, but the heterogeneity and variations of UMIs and gene counts in tumor samples (BC1160920F and BCCID4535) showed no distinct differences compared to the normal sample (BCCID44971) within a relatively small range (**Response Figure 1c-d**). More importantly, Spotiphy produced high-quality scRNA data for these three breast cancer samples. The novel tumor subtypes identified by these scRNA data were further validated by independent Resolve smFISH data (image-based ST data), revealing intriguing cell communication patterns that were consistent with previous research (Fig. 5).

These results suggest that the variations in spot-based data described as “*cells of different types with highly variable local density and context-dependent transcriptional states*” were detected by Spotiphy, thereby producing reliable scRNA data in at least some tumor samples when an accurate and comprehensive scRNA reference was provided. Based on these observations, we believe that the scRNA data of former lung and CRC datasets are also reliable, at least to some extent (Supplementary Fig. 17). We have listed additional factors contributing to the heterogeneity and variations in QC metrics to provide a more comprehensive discussion of what affects the accuracy of Spotiphy's results. One such factor is the

suboptimal quality of Visium data, which results in high noise levels that obscure the true differences in the data. As a computational biology research lab specializing in the generation and analysis of scRNA-seq and ST data, our group has profiled and analyzed numerous tumor and non-tumor datasets from hundreds of tissues and species using different protocols and platforms. With our “*limited familiarity with single-cell and spatial tumor expression data and analysis*”, we have observed that the QC metrics for lung and CRC datasets were not as ideal compared to other tumor data we have encountered (**Response Figure 1**). This prompted us to highlight data quality in our previous response. Again, we are not making excuses for the shift issues but hope that this additional data and explanation would help the reviewer accurately understand our thought process. Additionally, we are open to collect more tumor Visium data to investigate the relationship between QC metrics of spot-based profiles and the quality of scRNA data, if the reviewer requires.

b. Lineage blending in UMAP space (Response Figure 2). The authors use a different method to cluster the data (scMINER), yielding better separation. That is fine, but it doesn't address the original issue, namely that Spotify fails to adequately separate the cell types in the imputed scRNA profiles (more below). If it had instead produced clean profiles, a standard Seurat analysis would be sufficient to distinguish major cell types.

Response: As the reviewer noted, Seurat (R) or Scanpy (Python) is excellent entry-level tool for scRNA-seq cluster analysis that can handle most tasks with “*discrete clusters indicative of different cell types*”. However, “*tumor samples have cells of different types with highly variable local density and context-dependent transcriptional states*”. These complicated, context-dependent cell subtypes should be present in both spot-based ST and scRNA-seq data. Therefore, it is particularly necessary to adopt more sensitive and accurate clustering approach to effectively distinguish these cell subtypes, which are inherently very similar. This is one of the advantages of scMINER, as we aimed to illustrate in our previous Response Figure 2. In fact, the reviewer's argument that “*If it had instead produced clean profiles, a standard Seurat*

analysis would be sufficient to distinguish major cell types” is not reasonable. We randomly selected a scRNA-seq data of tumor-associated myeloid-derived cells (MDCs) from a recently published study by Gabriela Rapozo Guimarães et al. (<https://doi.org/10.1038/s41467-024-49916-4>), where they used Scanpy for clustering analysis (**Response Figure 2**). It is evident that different immune cell types are not well dispersed, and some markers show “background expression and bleeding” between different cell subtypes. We do not want to delve deeply into whether the annotations in this paper are correct. Our intention is merely to illustrate that it is very challenging to properly cluster and annotate cell types with similar expression profiles using Seurat or Scanpy, even with the real scRNA-seq data. In contrast, scMINER managed to effectively separate all cell subtypes in the scRNA data of lung and CRC datasets. Seurat also displayed decent results, indicating that Spotipy is actually doing a good job generating scRNA data (previous Response Figure 2). Rather than denying, we have acknowledged and discussed the potential

drawbacks of Spotiphy in the updated manuscript. Currently, Spotiphy is unable to produce ideal iscRNA data for input with any conditions, as the reviewer envisions. If there is similar expression “leakage” in the scRNA-seq reference used, the iscRNA data would exhibit similar patterns. We also discussed that introducing DL-based modeling could be a solution to make Spotiphy smarter and more capable of accurately distinguishing gene expression from different cell types, but it’s not the main focus of this paper.

c. Lineage blending in expression space (Response Figure 3). The authors claim that cell type markers in the iscRNA data are consistent with cell type annotations. Nevertheless, from the bubble and violin plots, most of the markers show severe background expression and bleeding into the wrong cell types. Moreover, even when aggregated, several marker profiles are only marginally specific for the correct cell type (endothelium in panel e and fibroblasts in panel f). Furthermore, the most egregious examples from our previous review are immune cells and those are conspicuously absent in Response Figure 3. Collectively, these data strongly imply that the iscRNA data have limited fidelity to real single-cell transcriptomes.

Response: As demonstrated above, “expression leakage” of certain marker genes among similar cell types can occur even in some real scRNA-seq data. We would like to emphasize that since iscRNA data is derived from scRNA-seq reference and Visium data, current version of Spotiphy cannot “magically” generate iscRNA data with quality equivalent to or better than real scRNA-seq data. Acknowledging the shortcomings of such iscRNA data, we would like to discuss the extent to which each of these inputs affects its accuracy.

To address reviewer's concern regarding epithelial, endothelial and fibroblast cells in the iscRNA data of lung cancer, we used feature plots to illustrate the distribution of these marker genes in the scRNA reference for lung cancer (**Response Figure 3a-c**). The results showed that in the real scRNA-seq reference, the epithelial marker genes were primarily distributed in the corresponding clusters, with a small amount of leakage into the endothelial clusters, and the expression level was significantly lower. This is consistent with what was observed in the iscRNA data (previous Response Figure 3d). The endothelial marker genes were mainly distributed in the corresponding clusters in the scRNA-seq reference. However, in the iscRNA data, aside from the corresponding cluster, there was clear leakage of endothelial marker genes, primarily concentrated in other myeloid and some epithelial cells (previous Response Figure 3e). For fibroblast marker genes, leakage occurred in both scRNA-seq reference and iscRNA data, mainly concentrated in other myeloid (previous Response Figure 3f). To ensure the correct identification of other myeloid, we examined the distribution of its marker genes in both the iscRNA and scRNA-seq references (**Response Figure 3d-e**). The results showed that, although there was a certain degree of leakage of marker genes expression patterns, the cell identity was clearly consistent with the iscRNA data’s annotation, which indicated that for these four cell types, clusters can be correctly annotated using the expression pattern of marker genes, even without the pre-labeled annotation from iscRNA data.

Based on these results, we conclude that **when there is some leakage in the scRNA-seq reference, iscRNA data cannot avoid retaining this leakage**. As we emphasized earlier, the current version of Spotiphy also suffers from “inappropriate” gene expression assignment, which may increase the leakage level in iscRNA data. This phenomenon appears to be more severe in cell types with smaller populations (such as immune cells), which is somewhat expected given Spotiphy's modeling. Therefore, we have emphasized in the Discussion section the importance of collecting sufficient cell numbers for each cell type in the scRNA-seq reference to achieve high accuracy in the outputs, and potential limitations. But this defect **does not** affect the unsupervised cell type annotation with marker genes.

Response Figure 3: Markers of epithelial (a), endothelial (b), fibroblast (c), and other myeloid (d) confirmed the cell identities in the scRNA-seq reference of lung cancer. Markers of other myeloid (e) in isCRNA data of lung cancer.

d. *Quality of scRNA-seq reference (Response Figure 4).* The authors once again blame the quality of the input data, rather than their own method, for problems with the isCRNA data. Indeed, inferCNV is sensitive to the reference cells and copy number variants can be spuriously detected. This is well-known in the field. The “tumor-like” CNV patterns supposedly detected in fibroblasts in the original scRNA-seq data are simply due to physically linked genes with similar expression patterns, a phenomenon that would be seen with any cell type if inappropriate references are used. This is clear in Response Figure 4b for each cell type in the heat map, all of which show cell type-specific patterns. The concern with their results is that Spotiphy is imputing similar CNV-like patterns across distinct cell types (e.g., epithelium and fibroblasts, Response Figure 4e), likely owing to mixing of such cell types in the same spots, resulting in the generation of spurious signal.

Response: The reviewer's concern is that the epithelial and fibroblast cells from isCRNA data of colorectal cancer exhibited similar CNV patterns (previous Response Figure 4e). But the real scRNA-seq data showed relatively distinct CNV patterns under the same settings (previous Response Figure 4b). We extracted these two panels and highlighted the differences in CNV patterns between epithelial and fibroblast cells in **panel e** with a red box (**Response Figure 4**). It is clear that while some similarities exist, there are also significant differences (even opposite patterns) of more than 50% of the heatmaps. As mentioned earlier, considering that isCRNA data's quality is lower than the scRNA-seq reference, this could explain the extra similarities. Nonetheless, it **does not** prevent us from distinguishing different cell subtypes by their CNV patterns, which in fact supports that Spotiphy's isCRNA data has great application potential for downstream analysis.

Response Figure 4: InferCNV analysis of colorectal cancer sample using different normal cell references.

2. *Deconvolution evaluation against previous methods.* We stand by our position that the authors should redo their benchmarking analysis of external datasets using added noise. As it stands, the only “real” ground truth that the authors have employed is (i) not precisely ground truth, as the sections are adjacent and geographic heterogeneity is most certainly present, and (ii) data from brain tissue, an idealized setting where nearly all ST methods have been developed and tested. Beyond that, they use pseudo-bulk data of scRNA-seq to create synthetic spot-based Visium samples. While such data offer precise ground truth, they are also idealized and divorced from reality. Given concerns about the application of Spotiphy to cancer specimens (above), and given the many audacious claims made by the authors about Spotiphy's capabilities, it is reasonable to remain skeptical about its performance on real-world data. Hence the request to add more noise to the benchmarking analysis.

a. As a side note, we did not state that Spotiphy is “the best among all available methods.” Please do not distort our language. Indeed, RCTD is easily comparable to Spotiphy in deconvolution accuracy per

Supplementary Figure 11. We stated that deconvolution ability “is the strongest aspect of both Spotipy and this paper.”

Response: As suggested by the reviewers, we have included benchmarking analysis of external datasets with additional noise. The detailed methodology for introducing the noise is provided in the Methods section and Supplementary Methods. Based on the previous results (Supplementary Figure 11), we selected 6 datasets and performed this analysis on the top five best-performing methods, including Spotipy (**Response Figure 5**).

As shown in the figure, for each data set, the five boxes on the left represent the original data without added noise, while the five boxes on the right include the additional noise. Although the added noise reduces the overall accuracy of all methods, the general trend remains consistent: Spotify's performance continues to be robust and highly accurate.

The reviewer suggests using additional “real” datasets to evaluate Spotify’s performance, but the reviewer did not realize that the benchmarking cannot be completed when there is no corresponding ground truth (**Response Figure 6**). The benchmarking paper (<https://doi.org/10.1038/s41592-022-01480-9>) collected 45 paired real datasets (ST and scRNA-seq) and 32 simulated datasets. Among the 45 pairs of real datasets, 17 are image-based ST, which are not applicable for Spotify. The remaining 28 pairs are sequence-based ST data, but due to the absence of ground truth (cell-type proportions), deconvolution evaluation cannot be performed with ST and scRNA-seq provided. In fact, the evaluation of the image-based data pairs in that paper relies solely on the similarity in the distribution of certain genes (paper Fig. 2). For cellular proportion and deconvolution evaluations, 32 simulated datasets were used, as only these simulated datasets have a known ground truth for benchmarking (paper Fig. 4). Followed the logic of the original paper, the additional datasets we used are all from these 32 simulated datasets.

Spatial type	Spatial tech.	Data no.	Size/ radius	Spots	Genes	Sparsity of examined data
Image-based	seqFISH	5	≤1 cell	2,952	4,129	0.46
	MERFISH	4	≤1 cell	3,746	3,395	0.35
	STARmap	2	≤1 cell	1,465	593	0.79
	ISS	2	≤1 cell	6,000	120	0.93
	FISH	1	≤1 cell	3,039	84	0.72
	osmFISH	1	≤1 cell	3,405	33	0.30
	BARISTAseq	1	≤1 cell	11,426	80	0.83
	EXseq	1	≤1 cell	1,154	42	0.41
Seq-based	10x visium	21	55 μm	2,002	28,737	0.32
	Slide-seq	3	10 μm	1,9175	23,682	0.89
	ST	2	100 μm	875	17,916	0.45
	HDST	1	≤1 cell	6,000	14,199	0.36
	Seq-scope	1	≤1 cell	2,177	19,532	0.86
Simulated	Simulated	32	5–15 cell	10,000	25,224	0.32

Response Figure 6: Information for the external datasets. From the original paper figure 1.

In fact, this "compromise" benchmarking approach in this article highlights the significant biological importance of our matched dataset. This is why we invested significant efforts into generating **the matched datasets (scRNA-seq, Visium, Xenium, and CosMx) for mouse brains**. Overall, we invested over \$200,000 to generate these datasets. As we emphasized in the updated manuscript, we believe these matched datasets are a valuable resource for the spatial omics community, particularly for the development and evaluation of ST algorithms. As proof of their great value, the datasets have been downloaded over 200 times through Zenodo (**Response Fig. 7**), despite our work not yet being published or submitted to *bioRxiv* in the past ten months.

January 16, 2024 (v1) Dataset Open
 View Edit

Spotiphy: generative modeling in single-cell spatial whole transcriptomics

Yang, Jiyan ; Zheng, Ziqian

Spatial transcriptomics (ST) is revolutionizing tissue analysis by preserving spatial context, yet achieving spatial whole transcriptomics at single-cell resolution remains challenging. Here we present Spotiphy (Spot imager with pseudo single-cell resolution histology), a novel method and toolkit, that combines the power of generative modeling with single-cell RNA sequencing...

Uploaded on January 23, 2024 181 257

Response Figure 7: Downloads of our matched datasets on Zenodo.

3. Slide-tags data. We stand by our position that the application of Spotiphy to these data provides little, if any, value. At a minimum, we see no point in imputing scRNA-seq data as the original data will invariably be higher quality and there are no “new” expression data (such as spot transcriptomes) to learn novel variation. All that remains is the nuclei detection and subsequent “out-of-spot” imputation, which as we stated previously, provides a pretty picture but nothing more.

Response: The reviewer commented that our application of Spotiphy does not provide meaningful value to the Slide-tags data. We respectfully disagree with the reviewer. As mentioned in our previous response, the presence of fewer than 50 nuclei in the original Slide-tags dataset for CA3, Cortical, Endothelial, Inhibitory, and Microglia poses a significant challenge for performing any unsupervised downstream analysis, such as identifying differentially expressed (DE) genes. The insufficient statistical power makes it difficult to draw any meaningful conclusions.

In scRNA-seq data analysis, determining the sample size needed to detect differentially expressed (DE) genes involves hypothesis testing for differences in mean expression levels. The null hypothesis states that there is no difference in the mean values of transformed CPM for any gene between two groups, whereas the alternative hypothesis suggests that a difference does exist. The test statistic is assumed to follow a t-distribution, which forms the fundamental basis of the widely used limma test for DE analysis. To determine the required sample size, we evaluate different effect sizes and standard deviation values under a type I error rate of 0.05 and a desired statistical power of 0.8. For instance, assuming a fold change effect size of 1.5, a mean expression value of 1 for the control group, and a standard deviation of 1 for any given gene, the required sample size is **64** cells per group (**Response Table 1**). The assumed mean expression and standard deviation values are highly conservative, derived from real data to ensure robust sample size estimates for practical applications. This result suggests that the DE analysis results obtained using Slide-tag original data are statistically insignificant due to insufficient power.

Difference in mean (treatment vs. control)	0.12	0.25	0.38	0.5	0.62	0.75	0.88
Sample size for control	1092	253	110	64	42	29	22
Sample size for treatment	1092	253	110	64	42	29	22
Total sample size	2184	506	220	128	84	58	44

Response Table 1: Sample size given varying “difference in mean response”.

As the reviewer noted, “there are no “new” expression data (such as spot transcriptomes) to learn novel variation” from the pseudo snRNA data generated by Spotiphy, which aligns with our expectations. Admittedly, while Spotiphy can increase the number of nuclei for each cell type and provide sufficient

statistical power to perform downstream analysis (Extended Data Fig. 10), it also raises additional concerns about using pseudo snRNA data.

Another potential improvement could come from cellular composition analysis within a given ROI. As mentioned earlier, the original Slide-tags dataset can barely reflect the cellular composition of any ROI accurately due to the significant loss of nuclei. Although the cellular composition inferred by Spotiphy differs somewhat from the original data (Extended Data Fig. 10f), it remains generally consistent. Currently, it is difficult to draw clear conclusions about whether Spotiphy could improve the accuracy of cellular composition. We are open to exploring this direction further if more publicly available slide-tag data becomes available in the near future.

Overall, working with Slide-tags data is challenging. Although it warrants further investigation, it is not the primary focus of this paper. In fact, applying Spotiphy to Slide-tags data is merely a minor extension of Spotiphy and has never been our main focus.